# The number of cytokinesis nodes in mitotic fission yeast scales with cell size

Wasim A Sayyad[1], Thomas D Pollard[1,2,3]*

[1]Department of Molecular Cellular and Developmental Biology, Yale University, New Haven, United States; [2]Department of Molecular Biophysics and Biochemistry, Yale University, New Haven, United States; [3]Department of Cell Biology, Yale University, New Haven, United States

**Abstract** Cytokinesis nodes are assemblies of stoichiometric ratios of proteins associated with the plasma membrane, which serve as precursors for the contractile ring during cytokinesis by fission yeast. The total number of nodes is uncertain, because of the limitations of the methods used previously. Here, we used the ~140 nm resolution of Airyscan super-resolution microscopy to measure the fluorescence intensity of small, single cytokinesis nodes marked with Blt1-mEGFP in live fission yeast cells early in mitosis. The ratio of the total Blt1-mEGFP fluorescence in the broad band of cytokinesis nodes to the average fluorescence of a single node gives about 190 single cytokinesis nodes in wild-type fission yeast cells early in mitosis. Most, but not all of these nodes condense into a contractile ring. The number of cytokinesis nodes scales with cell size in four strains tested, although large diameter *rga4Δ* mutant cells form somewhat fewer cytokinesis nodes than expected from the overall trend. The Pom1 kinase restricts cytokinesis nodes from the ends of cells, but the surface density of Pom1 on the plasma membrane around the equators of cells is similar with a wide range of node numbers, so Pom1 does not control cytokinesis node number. However, when the concentrations of either kinase Pom1 or kinase Cdr2 were varied with the *nmt1* promoter, the numbers of cytokinesis nodes increased above a baseline of about ~190 with the total cellular concentration of either kinase.

*For correspondence:
thomas.pollard@yale.edu

Competing interest: The authors declare that no competing interests exist.

## Editor's evaluation

This manuscript reports important quantitative results on cytokinetic nodes using fission yeast as a model. Using an Airyscan microscopy the authors develop images with ~ 140 nm resolution and estimate ~ 190 cytokinetic nodes per cell prior to actomyosin ring assembly. The work also describes important conclusions on scaling parameters, all of which will help refine quantitative models of cytokinesis.

## Introduction

Cytokinesis nodes are stoichiometric assemblies of multiple proteins, which associate with the plasma membrane around the middle of fission yeast cells and polymerize actin filaments that form the cytokinetic contractile ring (*Vavylonis et al., 2008*). Similar assemblies containing myosin-II may form contractile rings in other cells (*Beach et al., 2014*; *Henson et al., 2017*; *Hickson and O'Farrell, 2008*; *Werner et al., 2007*). In fission yeast, cytokinesis nodes form from two distinct types of interphase nodes (*Akamatsu et al., 2014*). At the end of cytokinesis, type 1 nodes containing kinases Cdr1 and Cdr2 appear in a broad band around the middle of the daughter cells as SIN (septation initiation network) activity drops (*Pu et al., 2015*) and remain stationary throughout the interphase (*Akamatsu et al., 2014*; *Martin and Berthelot-Grosjean, 2009*; *Morrell et al., 2004*; *Moseley et al., 2009*). Type

2 nodes containing the proteins Blt1p, Klp8p, Gef2p, and Nod1p (*Jourdain et al., 2013*; *Martin and Berthelot-Grosjean, 2009*; *Morrell et al., 2004*; *Moseley et al., 2009*; *Ye et al., 2012*) originate from the constricting contractile ring from the previous cell cycle and diffuse along the plasma membrane from the new ends of the daughter cells. During the $G_2$ phase of the cell cycle, type 1 nodes capture diffusing type 2 nodes to form cytokinesis nodes (*Akamatsu et al., 2014*). During 10 min prior to spindle pole body (SPB) separation, cytokinesis nodes mature by sequentially accumulating stoichiometric ratios of myosin-II (Myo2 heavy chain and two light chains), IQGAP Rng2p, and F-BAR protein Cdc15p (*Vavylonis et al., 2008*; *Wu et al., 2006*). At the time of SPB separation, smaller numbers of formin Cdc12p join each node, where they nucleate and elongate actin filaments (*Laporte et al., 2011*; *Wu et al., 2006*). Interactions between actin filaments from one node and myosin-II in adjacent nodes pull the nodes together into a contractile ring (*Vavylonis et al., 2008*). Interestingly, 10–12 min after SPB separation, high SIN activity disperses the Cdr2p-mEGFP from the cytokinesis nodes into the cytoplasm (*Akamatsu et al., 2014*; *Martin and Berthelot-Grosjean, 2009*; *Morrell et al., 2004*; *Moseley et al., 2009*).

In addition to being the precursors for cytokinesis nodes, interphase nodes are critical for timing cell division. Interphase cells grow linearly at their tips (*Fantes and Nurse, 1977*; *Moreno et al., 1989*) until they reach a threshold size of twice the initial cell length (*Gu and Oliferenko, 2019*; *Neumann and Nurse, 2007*). During this period, the interphase node components also double in number (*Allard et al., 2018*; *Pan et al., 2014*). Interphase nodes are thought to sense the cell surface area to control cell size at the time of division (*Facchetti et al., 2019a*; *Pan et al., 2014*).

In spite of their central role in the cell cycle, quantitative analysis of nodes has been challenging, largely due to their heterogeneity observed by fluorescence microscopy. Rather than appearing as a uniform population of spots, the fluorescence from node markers ranges from tiny dim spots to much larger and brighter particles both during interphase (*Moseley et al., 2009*) and mitosis (*Vavylonis et al., 2008*). Histograms of fluorescence intensity per node show that small, dim nodes predominate, although the eye of the observer is drawn to larger, more intense spots (*Akamatsu et al., 2017*). Many studies ignored dim nodes entirely.

Single-molecule localization super-resolution microscopy (SMLM) by *Laplante et al., 2016* explained the heterogeneity of node intensities and sizes. The 35 nm resolution of this method revealed that cytokinesis nodes in live cells are much smaller (<100 nm) than their diffraction-limited images in confocal micrographs. These particles are uniform in size and contain stoichiometric ratios of proteins, each with a distinct spatial distribution within the node. Some of these tiny, single nodes are so close together that they have twice or three times the numbers of localizations of single nodes, accounting for the variation in node intensity in confocal micrographs. Blurring the super-resolution images of nodes with the point spread function of a confocal microscope reproduced the heterogeneous mixtures of sizes and intensities. SMLM showed that the two types of interphase nodes are also tiny structures that tend to form clusters with twice or three times the number of localizations of single nodes (*Akamatsu et al., 2017*). Dim nodes predominate in confocal micrographs of interphase cells. With a favorable marker (such as the abundant protein Cdr2) histograms of node intensities have higher intensity peaks corresponding to multiples of these dim spots, which Akamatsu et al. called 'unitary nodes' corresponding to Laplante's single nodes.

The difficulty in resolving individual nodes has compromised efforts to count cytokinesis nodes. *Vavylonis et al., 2008* counted 63 ± 10 cytokinesis nodes tagged with Rlc1-3GFP in the broad bands of wild-type cells in 3D reconstructions of *z*-series acquired by spinning disk confocal microscopy. Nodes with low intensities grouped in one main peak; a smaller, higher intensity peak was attributed to pairs and clusters of unresolved nodes. *Deng and Moseley, 2013* applied deconvolution to images obtained in wide-field microscopy and counted 52–106 interphase nodes tagged with Cdr2-mCherry. *Pan et al., 2014* used the Find Maxima macro function in ImageJ to find pixels with maximum intensity in maximum projection images and counted <50 interphase nodes and noted that the number scaled with cell length. However, they ignored dim nodes by choosing the only pixels in ROI with intensities greater than approximately twice the mean background intensity. Using the same scaled fire LUT plugin in ImageJ (*Willet et al., 2019*) measured ≤15 nodes marked with Mid1 in wild-type cells that had a yellow, orange, or white intensity but did not count dim purple or red intensity nodes. *Allard et al., 2018* used the superior (140 nm) resolution of Airyscan microscopy and an automated particle tracking plugin to count ~50 nodes tagged with Cdr2-mEGFP during interphase. However,

the algorithm used in the plugin was designed to find only local intensity maxima and did not count the dim nodes with low intensities.

In principle, one should be able to identify and count every single node in a cell by SMLM, but the underlying strategy to collect the data has precluded this direct approach in living cells with fluorescent proteins. Only a thin slice of the cell is imaged before all of the proteins are irreversibly photobleached. Therefore, Laplante et al., 2016 used this limited sample to estimate the number of cytokinesis nodes in fission yeast broad bands. One approach was to measure the local density of nodes in a 400-nm optical section from the top or bottom of the cell and then extrapolate this density around the equator, giving ~140 total nodes for five different tagged proteins (Mid1, Rlc1, Cdc15, Myo2, and Rng2). The second method was to blur the SMLM images with the point spread function to simulate a confocal image. This resulted in about half the single nodes being blurred together, so the 65 nodes counted by Vavylonis et al., 2008 represented roughly 130 single nodes.

Accurate measurements of node numbers are important for two reasons. First, the number is a key parameter in computer simulations to test ideas about the mechanisms of contractile ring assembly and constriction. Second, estimates of the number of molecules per node have relied on measuring the total numbers of each protein in the broad band of nodes and then dividing by the number of nodes (Laplante et al., 2016; Vavylonis et al., 2008).

We aimed to make a definitive count of the number of cytokinesis nodes and determine how this number varies with cell size. Our strategy depended on three considerations. First, we used the scaffold protein Blt1 tagged with mEGFP to mark cytokinesis nodes, since it is the only protein known to be associated with nodes throughout the cell cycle. Other node proteins join or leave nodes throughout mitosis. For example, Myo2, Rng2, and Cdc12 arrive during 10 min prior to the separation of SPBs, and Mid1 and Cdr2 depart as contractile rings form and mature. Second, we collected data during a short time interval of 2–5 min after the SPB divided, when the nodes are still in the form of a broad band, to avoid the compositional changes that occur continuously during the cell cycle. All of the nodes considered here were within minutes of condensing into contractile rings. Third, following Allard et al., 2018 we used Airyscan microscopy to image cytokinesis nodes. The higher resolution of Airyscan allowed us to detect and measure the fluorescence of dim individual cytokinesis nodes better than by conventional confocal microscopy. We calculated the total number of cytokinesis nodes from the ratio of the total intensity of Blt1-mEGFP in the broad band of nodes around the equator to the average intensity of a single node.

We report that wild-type fission yeast cells have about 190 cytokinesis nodes in a broad band at the equator. About 85% of these cytokinesis nodes condense into the contractile ring. The number of cytokinesis nodes during early mitosis scales directly with cell size in strains that are smaller or larger than wild-type cells. The kinases Pom1 and Cdr2 modulate the number of nodes set by the cell size.

## Results
### Blt1 as a cytokinesis node marker

Two challenges complicate counting cytokinesis nodes in fission yeast. First, cytokinesis nodes form over 10 min, and after the SPB divides, they condense into a densely packed contractile ring over 10–12 min. Second, the protein composition of cytokinesis nodes changes dramatically during these 22 min (Bhatia et al., 2014; Lee and Wu, 2012; Pan et al., 2014; Rincon et al., 2014). The numbers per node of formin Cdc12, F-BAR Cdc15, anillin Mid1, myosin-II Myo2, and IQGAP Rng2 change rapidly as cytokinesis nodes form from interphase nodes (10 min prior to SPB separation), condense into contractile rings (10–12 min after SPBs separate) and then mature (over the next 15 min) (Wu and Pollard, 2005).

We solved the first issue by studying live cells with SPBs marked with Sad1-RFP. We selected cells with two narrowly separated SPBs and well-dispersed nodes in a broad band around the equator.

We dealt with the second issue by marking nodes with the scaffold protein Blt1 tagged with mEGFP. Blt1 is the only protein known to persist in nodes throughout the cell cycle, so it is found in interphase nodes, cytokinesis nodes, and contractile rings (Akamatsu et al., 2014; Goss et al., 2014; Moseley et al., 2009). Therefore, Blt1 is the only reasonable marker to measure the number of cytokinesis nodes. The fluorescent proteins were integrated into the haploid genome to ensure that all copies of each protein were tagged and expressed at normal levels.

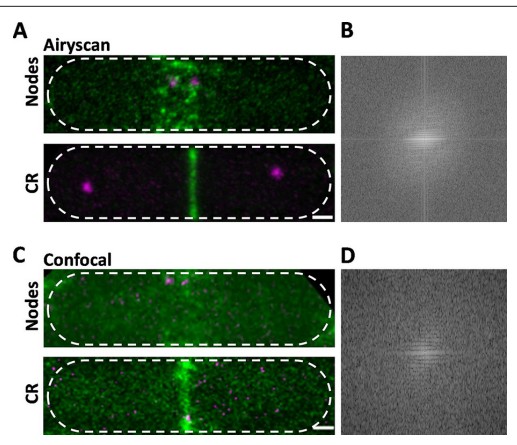

**Figure 1.** Comparison of Airyscan and confocal images of cytokinesis nodes and contractile rings in wild-type fission yeast cells expressing Blt1-mEGFP (green) and Sad1-RFP showing two spindle pole bodies (SPBs) (magenta). (**A**) Airyscan images. Max projections of 40 Z-slices. The upper cell has a broad band of cytokinesis nodes (green). The lower cell has a contractile ring. (**B**) Fast Fourier transform of the image in lower panel A, green channel. (**C**) Confocal images. Max projections of 20 Z-slices of confocal fluorescence with 0.6 AU + 8X averaging + iteratively constrained deconvolution micrographs. The upper cell has a broad band of cytokinesis nodes. The lower cell has a contractile ring. (**D**) Fast Fourier transform of the image in lower panel C, green channel. The scale bar is 1 μm.

## Airyscan fluorescence microscopy resolves dim unitary cytokinesis nodes in a broad band

We used a Zeiss Airyscan fluorescence microscope to acquire 40 Z-slices covering the entire volume of cells to measure the total and local fluorescence of Blt1-mEGFP in wild-type fission yeast cells in 2–3 s. The frame size and pixel dwell time were adjusted so that motions of cytokinesis nodes (20 nm/s; *Vavylonis et al., 2008*) did not blur the images during image acquisition. Cytokinesis nodes appeared as a broad band of fluorescence spots in the cortex around the middle of mitotic cells with two SPBs. The fluorescence intensities of these spots ranged from very bright to dim (*Figure 1A*, upper panel).

To compare the high resolution and sensitivity of Airyscan to a conventional laser scanning confocal microscope, we used the same Airyscan detector in the confocal mode with a pinhole of 0.6 AU, 8× line averaging + linear constrained iterative deconvolution, and 2× laser intensity to compensate for the lower signal-to-noise ratio as assessed previously (*Kolossov et al., 2018*). Large nodes stood out in the confocal images, but the higher background noise made it difficult to detect the dim cytokinesis nodes in broad bands. Contractile rings were brighter and narrower in Airyscan than confocal images (*Figure 1A, C*, lower panels). Fast Fourier transform analysis of images of cells with contractile rings showed that Airyscan retrieved higher frequencies and more extensively distributed information. The horizontal line in the frequency domain of the Airyscan image corresponds to the contractile ring (*Figure 1B*), which was weak in the frequency domain of the confocal image (*Figure 1D*). In addition to providing data for 3D reconstructions, Airyscan imaging can resolve small, dim cytokinesis nodes, which we interpret as single nodes, while the brighter nodes consist of two or more single nodes (*Akamatsu et al., 2017*; *Laplante et al., 2016*) located too closely to be resolved.

Time-lapse imaging showed that cytokinesis nodes of all sizes, including single nodes, condensed into bigger and brighter nodes as they coalesced into a contractile ring during 10–12 min after separation of the SPBs as described for the brighter fraction of the population of nodes by confocal microscopy (*Vavylonis et al., 2008*). Neither the Airyscan nor confocal microscopy resolved cytokinesis nodes known to be present in contractile rings (*Figure 1A, C*, lower panels) by single-molecule localization microscopy (*Laplante et al., 2016*).

## Counting cytokinesis nodes in the broad band

The methods section explains how we used the three-dimensional distribution of fluorescence of spots around the equator to identify nodes (*Figure 2A*) and measure their fluorescence. To qualify as a single node, the fluorescence had to be confined in the X–Y plane within a circle of 12-pixels (0.5 μm) in diameter and in the Z direction in more than 3 and less than 7 Z-slices of 170 nm (1.19 μm). The total fluorescence within the 12-pixel diameter circle in 7 Z-slice images was sum projected and measured as a node fluorescence (*Figure 2B*). From the same sum projected image we calculated the average fluorescence from 8 equivalent volumes of cytoplasm on both sides of the node broad band (*Figure 2B*). The fluorescence of each candidate node was its total fluorescence minus the average fluorescence of an equivalent volume (~0.9 μm³) of cytoplasm. We used 3D reconstructions of stacks

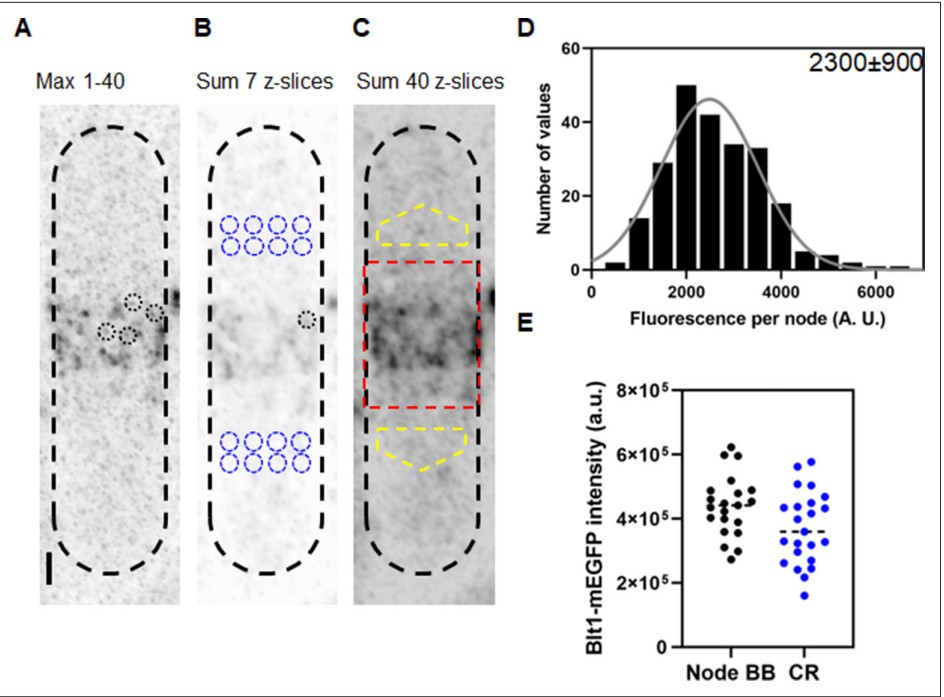

**Figure 2.** Counting cytokinesis nodes in broad bands and contractile rings of wild-type cells expressing Blt1-mEGFP. (**A**) Reverse contrast fluorescence micrograph of *Figure 1A*, upper panel. Black dotted circles are nodes selected for fluorescence measurements. The scale bar is 1 μm. (**B**) Reverse contrast fluorescence micrograph of a sum projection of seven slices of the cell in panel A. A black dotted circle marks a spot selected to measure the fluorescence of single nodes and blue dotted circles mark cytoplasmic regions selected to measure background fluorescence. (**C**) Reverse contrast fluorescence micrograph of a sum projection of 40 *Z*-slices of the cell in panel A. The red rectangle outlines the region used to measure the total node broad band fluorescence and the yellow polygonal area marks the area chosen to measure background intensity to subtract from the broad band area. (**D**) Histogram of the distribution of fluorescence intensities of a sample of 235 cytokinesis nodes from 21 cells including all the selected nodes. The continuous curve is a fit of a Gaussian distribution to the intensities of 231 single nodes excluding the 4 high-intensity nodes. The mean intensity value is 2300 AU and the standard deviation (SD) is 900 AU. (**E**) Bee–Swarm plots comparing the total Blt1-mEGFP intensity in node broad bands (BB) (black, 21 cells) and contractile rings (CR) (blue, 23 cells).

The online version of this article includes the following source data for figure 2:

**Source data 1.** Related to *Figure 2D, E*.

---

of Airyscan images to measure the total fluorescence of all the cytokinesis nodes around the equator of each cell to compare with the average fluorescence of single nodes.

We measured the fluorescence of 235 nodes in 21 mitotic, wild-type cells satisfying the aforementioned ROI conditions and plotted a histogram to determine the distribution of intensities and the average fluorescence intensity (*Figure 2D*). Given the small number of molecules in each node (*Akamatsu et al., 2017*; *Laplante et al., 2016*), the exchange of molecules between nodes and cytoplasm (*Laporte et al., 2011*) and the fact that less than 70% of mEGFP is mature and emitting photons (*Balleza et al., 2018*), a range of intensities is expected, which does not detract from the usefulness of an average node intensity for calculating the total number of nodes from the total fluorescence intensity of the broad band of nodes. An initial Gaussian fit peaked at 2486 AU with a few higher intensity nodes. We assumed that the sample included a few closely spaced pairs of nodes, so we eliminated nodes with fluorescence values more than twice the peak of the Gaussian curve (2% of the total), before calculating the mean fluorescence of the single nodes. For the sample of 231 single nodes, the Shapiro–Wilk $W$ test calculated $W = 0.9842$, which does not reject the null hypothesis that the data are from a normal distribution (*King and Eckersley, 2019*; *Royston, 1992*). A descriptive statistical test also gave a $Z$ score of 1.7, which also does not reject the null hypothesis that the data are from a normal distribution (*Ghasemi and Zahediasl, 2012*).

We estimated that wild-type fission yeast cells have on average ~190 (standard deviation [SD] = 40) single cytokinesis nodes from the ratio of total Blt1-mEGFP fluorescence intensity in the broad band (corrected for cytoplasmic background) of a given cell to the average fluorescence intensity of Blt1-mEGFP in a single node, 2300 ± 900 AU (*Figure 2D*).

To determine if all the nodes in the broad band combine to form a contractile ring in wild-type cells, we measured the total Blt1-mEGFP fluorescence of contractile rings (after background subtraction) and of the broad band of nodes (after background subtraction) (*Figure 2E*). The average total fluorescence intensities are 440,000 ± 95,000 AU in broad bands and 372,000 ± 111,000 AU in contractile rings. An unpaired *t*-test showed that these numbers are significantly different with p = 0.035. Thus, about 85% of the Blt1 in nodes is incorporated into contractile rings.

## Mutant strains with a range of sizes

We used five different mutant strains to determine the numbers of cytokinesis nodes in cells with a range of sizes. These mitotic cells had volumes from 50 to 450 μm$^3$ (*Figure 3*).

The Wee1 kinase catalyzes the phosphorylation of Cdc2/Cdk1 to prevent cells from entering mitosis (*Russell and Nurse, 1987*). At the restrictive temperature of 36°C, *wee1-50* cells enter mitosis prematurely, so they are smaller than normal (*Nurse, 1975*). After 4 hr at the restrictive temperature, the average cell length of *wee1-50* mutant cells (8.0 ± 0.9 μm) was about half the average length of wild-type cells (14.2 ± 0.8 μm) (*Figure 3C*). The diameter of *wee1-50* mutant cells (3.8 ± 0.1 μm) was the same as wild-type cells (3.7 ± 0.1 μm). When released to the permissive temperature, most cells started to form a contractile ring in 10–15 min. The broad band of nodes looked more dispersed in these cells than in other strains.

The Cdc25 phosphatase removes the inhibitory phosphates added by Wee1 from Cdc2/Cdk1 to allow cells to enter mitosis (*Nurse, 1990*; *Russell and Nurse, 1986*; *Russell and Nurse, 1987*). At the restrictive temperature of 36°C, *cdc25-22* mutant cells arrest at the G$_2$/M transition but continue growing at both ends. We varied the time at 36°C to obtain a range of lengths. After 4–6 hr at 36°C cells were 26.6 ± 5.7 μm long, twice the average length of wild-type cells but did not differ significantly in diameter (3.9 ± 0.1 μm) (*Figure 3A*) and had broader equatorial bands with more cytokinesis nodes (*Vavylonis et al., 2008*). The *cdc25-22* mutant cells produced a wide range of cell sizes, so we grouped them by size. When released to the permissive temperature, most of the cells synchronously entered mitosis within 1 hr and formed a broad band of cytokinesis nodes.

The RhoGAP Rga4 localizes to the division site of cells and inactivates the GTPase Cdc42 (*Das et al., 2007*). Cells with deletion mutations of *rga4* are shorter (11.6 ± 0.7 μm) but have larger diameters (4.4 ± 0.2 μm) than wild-type cells (*Das et al., 2007*; *Tatebe et al., 2008*; *Figure 3D*). Treating *rga4Δ* cells with 15 mM hydroxyurea for 4–6 hr inhibited DNA synthesis by inactivating ribonucleoside reductase (*Krakoff et al., 1968*) and arrested the cells in the S phase during which they grew longer (*Mitchison and Creanor, 1971*; *Figure 3F*). On the other hand, RhoGAP Rga2 localizes to growing cell tips and inactivates the Rho2 GTPase (*Villar-Tajadura et al., 2008*). Deletion of *rga2* does not change the cell length but decreases the cell diameter (3.4 ± 0.2 μm) (*Villar-Tajadura et al., 2008*; *Figure 3E*).

## Numbers of single cytokinesis nodes in a broad band scale with cell size

After correcting for cytoplasmic background fluorescence, we measured the total Blt1 fluorescence in the broad band around the equator of each cell (*Figure 2*). Assuming that this Blt1 is located in single nodes, we estimated the numbers of single cytokinesis nodes in the broad bands by dividing the total Blt1 fluorescence around the equator by the average intensity of the single nodes in that strain. The distributions of Blt1 fluorescence intensities of single cytokinesis nodes were similar (around 2300 AU) in wild-type cells (*Figure 2C*) and all the experimental strains (*Figure 3G*).

The numbers of cytokinesis nodes in the broad band of our sample of 125 cells were strongly correlated with cell volume (*Figure 3H*) with a slope of about 0.7 nodes per μm$^3$ of volume. This line does not extrapolate to the origin. Plots of the same data on node numbers versus cell length (*Figure 3I*) or surface area (*Figure 3—figure supplement 1A*) showed the same linear relationship seen for cell volume. Node numbers did not correlate with cell diameter (*Figure 3—figure supplement 1B*).

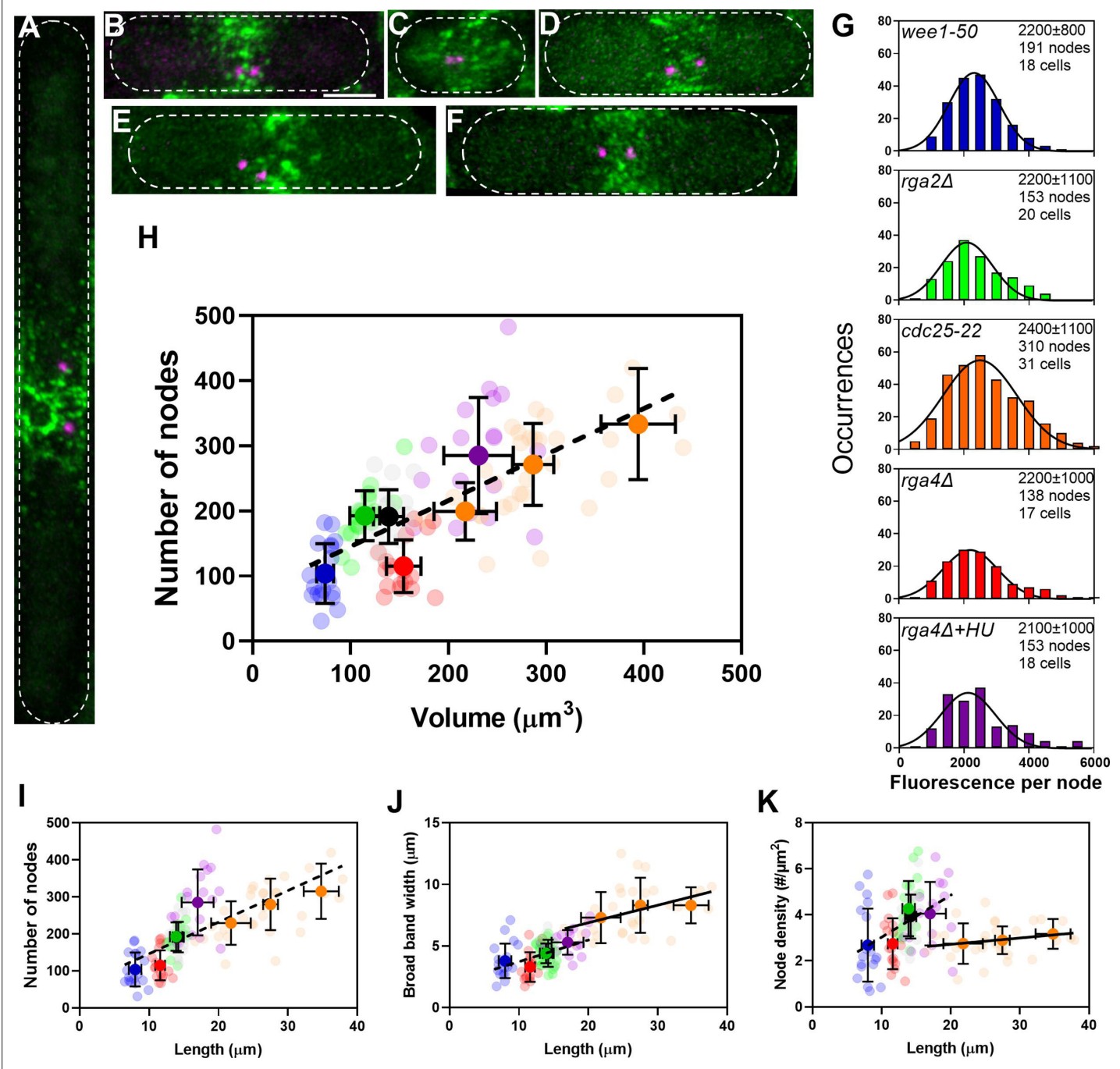

**Figure 3.** The number of cytokinesis nodes scales with the size of the cell. (**A–F**) Maximum intensity projections of Airyscan images of 40 Z-sections of fission yeast cells expressing Blt1-mEGFP and Sad1-RFP showing two spindle pole bodies (SPBs) (magenta). The scale bar is 3 μm. (**A**) *cdc25-22*; (**B**) wild type; (**C**) *wee1-50*; (**D**) *rga4Δ*; (**E**) *rga2Δ*; and (**F**) *rga4Δ* + HU at 25°C. (**G**) Histograms of the distributions of fluorescence intensities of small nodes in five different strains. *Figure 2D* shows the distribution of fluorescence intensities of small nodes in wild-type cells. Insets list average single node fluorescence intensity ± standard deviation (SD) for each strain. Tukey's multiple comparison test showed no significant difference in any strain from the wild-type strain with p > 0.05 except in the *rga4Δ* + HU strain, p = 0.04. The Shapiro–Wilk *W* test did not reject the null hypothesis of normal distribution in all strains with *W* values ≥0.97. The descriptive statistics *Z* score for our sample size 135 > n > 310 was <3.29 for all strains, which does not reject the null hypothesis of a normal distribution. (**H–K**) Relationships between parameters (node numbers, broad band widths, and node densities) and cell size parameters in six strains: (●) *wee1-50* (*n* = 18 cells); (●) *rga2Δ* at (*n* = 20 cells); (●) wild type at (*n* = 21 cells); (●) *rga4Δ* at (*n* = 17 cells); (●) *rga4Δ* + HU (*n* = 18 cells); and *cdc25-22* (●) at 25°C. All measurements for each strain are collected in one bin except for (●) *cdc25-22* cells, which are divided into three volume bins: 150–250 μm³, *n* = 8 cells; 250–350 μm³, *n* = 17 cells; 350–450 μm³, *n* = 6 cells, and three length bins: 15–25 μm, *n* = 14 cells; 25–30 μm, *n* = 10 cells; 30–40 μm, *n* = 7 cells. The transparent symbols are measurements of individual cells. The solid symbols are the mean numbers

*Figure 3 continued on next page*

Figure 3 continued

of node parameters. Except for the *cdc25-22* cells, the vertical error bars show the SD of the mean numbers and horizontal error bars the SD of the lengths. For the *cdc25-22* (volume 350–450 µm³) cells the error bar is the range. (**H**) Node number versus cell volume. The slope of the linear regression (black dashed line) is significantly different from zero (p < 0.0001; $R^2$ = 0.5). (**I**) Node number versus cell length. The slope of the linear regression (black dashed line) is significantly different from zero (p < 0.0001; $R^2$ = 0.5). (**J**) Width of node broad band as a function of cell length. The slope of the linear regression for four strains (black dashed line) is significantly different from zero (p < 0.0001; $R^2$ = 0.2), The slope of the linear regression for *cdc25-22* cells of various sizes (black line) is also significantly different from zero (p = 0.03; $R^2$ = 0.2). (**K**) Node density on the cortex of the broad band area as a function of cell length. The slope of the linear regression (black dashed line) is significantly different from zero (p < 0.0001; $R^2$ = 0.2). The slope of the linear regression for *cdc25-22* cells (black line) is not significantly different from zero (p = 0.3; $R^2$ = 0.05).

The online version of this article includes the following source data and figure supplement(s) for figure 3:

**Source data 1.** Related to *Figure 3G-K*.

**Figure supplement 1.** The number of cytokinesis nodes scales with the surface area of the cell.

**Figure supplement 1—source data 1.** Related to *Figure 3—figure supplement 1A,B*.

**Figure supplement 2.** Fluorescence intensities in the contractile rings of yeast strains as a function of cell volume.

**Figure supplement 2—source data 1.** Related to *Figure 3—figure supplement 2A,B*.

Comparing the numbers of cytokinesis nodes in the broad band of wild-type cells, *rga4Δ* cells, and *rga2Δ* cells provided data on the effect of cell diameter on the number of cytokinesis nodes. The number of cytokinesis nodes in long, narrow *rga2Δ* cells (193 ± 39) was indistinguishable from wild-type cells (191 ± 41) and aligned with the data from the other strains in plots of node number versus length and volume. On the other hand, the short, wide *rga4Δ* cells had fewer cytokinesis nodes (115 ± 40) than the wild-type and *rga2Δ* mutant cells with similar volumes (*Figure 3H*). The *rga4Δ* cells grew longer in 15 mM hydroxyurea for 4–6 hr and had the same number of nodes per volume as the other strains (*Figure 3H*).

The cells might use two strategies to accommodate a total number of nodes in a broad band determined by their size. Either the surface area (width) of the broad band or the density of nodes might increase with the total node number. We observed both. The very large *cdc25-22* mutant cells simply expanded the width of the broad band in proportion to the number of nodes, while the density of nodes and width of broad band both increased with their numbers in the other strains of freely cycling cells (*Figure 3J, K*).

## Blt1-mEGFP in contractile rings of cells with different sizes

We used five different strains to determine if the total Blt1-mEGFP fluorescence in contractile rings scales with cell size. The total Blt1-mEGFP fluorescence in contractile rings was generally higher in large cells (*Figure 3—figure supplement 2A*), but did not scale cleanly with size because the five tested strains incorporated different fractions of the total Blt1-mEGFP in the cytokinesis nodes into their contractile rings: 85% in wild-type cells; 40–70% in *cdc25-22*, *wee1-50*, and *rga2Δ* mutant cells; and nearly 90% in *rga4Δ* mutant cells. In *cdc25-22* mutant cells, the total Blt1-mEGFP fluorescence in cytokinesis nodes increased with cell volume (*Figure 3H*) but did not scale with cell size in their contractile rings (*Figure 3—figure supplement 2B*). Therefore, the largest *cdc25-22* cells had as little as ~50% of the cytokinesis node Blt1-mEGFP fluorescence in their contractile rings. In contrast, *cdc25-22* mutant cells did recruit mEGFP-Myo2 to their contractile rings in proportion to their volumes (*Figure 3—figure supplement 2B*).

## Does Pom1 control the number of cytokinesis nodes?

We performed two types of experiments to investigate the relationship between the membrane-bound Pom1 kinase and the number of cytokinesis nodes. Concentrating Pom1 at the poles depends on microtubules that transport the polarity proteins Tea1 and Wsh3/Tea4 to the poles (*Bähler and Pringle, 1998*; *Martin and Berthelot-Grosjean, 2009*; *Moseley et al., 2009*; *Tatebe et al., 2005*). First, we measured the distributions of Pom1 in cells with a wide range of sizes by time-lapse, spinning disk confocal microscopy (*Figure 4*). The five strains were wild-type, *rga4Δ*, *rga2Δ*, *cdc25-22*, and *wee1-50*. The cells expressed Pom1-GFP along with Sad1-RFP to identify mitotic cells with two SPBs and Blt1-mCherry to track nodes. Second, we varied the concentration of Pom1 and measured the number of nodes (*Figure 5*).

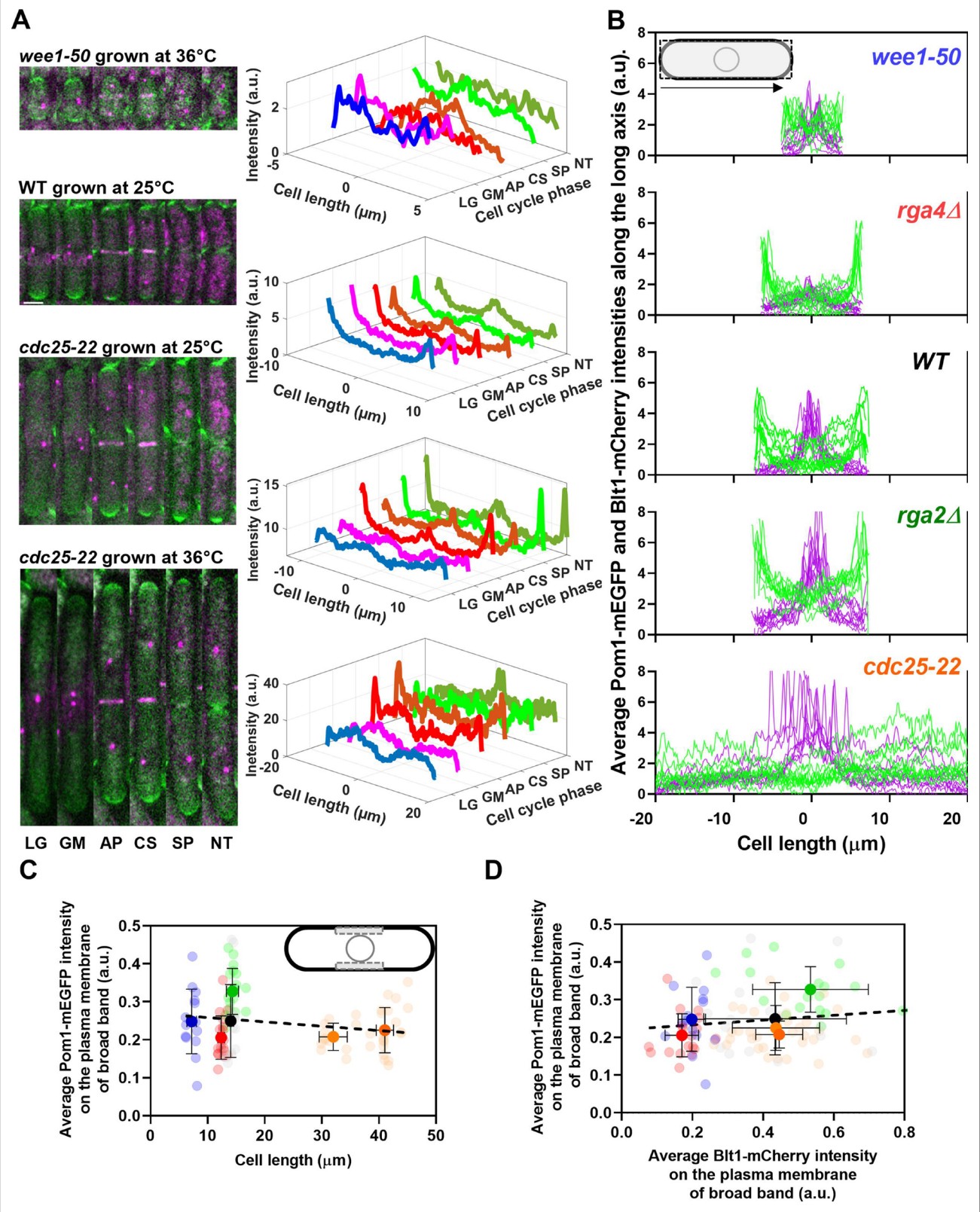

**Figure 4.** Distributions of Pom1 and Blt1 in cells that differ in size. (**A**) Distribution of Pom1 in wild type, *wee1-50*, and *cdc25-22* mutant cells. Each strain expressed Pom1-GFP and Blt1-mCherry-Sad1-RFP from their native loci. Each panel has two parts: (left) sum projected confocal images of 15 Z-slices of wild-type cells taken at different phases of the life cycle abbreviated as LG (late G2), GM (G2/M), AP (anaphase), CS (constriction), SP (separation), and NT (new end take off); and (right) time series of plots of the average intensity of Pom1-GFP fluorescence along the length of the cells. Scale bar 3 μm.

*Figure 4 continued on next page*

*Figure 4 continued*

(Top panel) *wee1-50* grown at 36°C and imaged at 22°C. (Second panel) Wild-type cells were grown at 25°C and imaged at 22°C. (Third panel) *cdc25-22* mutant cells grown at 25°C and imaged at 22°C. (Bottom panel) *cdc25-22* mutant cells grown at 36°C for 4 hr before imaging at 22°C. (**B**) Distributions of Pom1 and Blt1 in cells at the G$_2$/M transition measured by confocal microscopy of wild-type and four mutant strains. Cells were grown at 25°C and imaged at 22°C: (●, *n* = 9 cells) WT, (●, *n* = 7 cells) *wee1-50*, (●, *n* = 8 cells) *cdc25-22*, (●, *n* = 9 cells) *rga2Δ* and (●, *n* = 9 cells) *rga4Δ* mutant cells. Lines are fluorescence intensities of Blt1-mCherry-Sad1-RFP (magenta) and Pom1-GFP (green) across the full width of the cells and along the lengths of the cells. (**C, D**) Relationships between cell length and average Blt1-mCherry intensity on the plasma membrane of broad band with average Pom1-mEGFP intensity on the plasma membrane of broad band in five strains in (**B**). All measurements for each strain are collected in one bin except for (●) *cdc25-22* cells, which are divided into two length bins: 30–36 μm, *n* = 12 cells; 36–45 μm, *n* = 22 cells. The transparent symbols are measurements of individual cells. The solid symbols are the mean numbers of node parameters. The vertical error bars show the SD of the mean numbers and the horizontal error bars the standard deviation (SD) of the lengths. (**C**) Average Pom1-mEGFP intensity per pixel along the plasma membrane across the broad band of nodes in the middle focal plane of the cells as a function of the lengths of the cells. The inset shows the rectangle three pixels wide and the length of the broad band of nodes used to make the measurements. The slope of the linear regression (black dotted line) is not significantly different from zero (p = 0.07; $R^2$ = 0.03). (**D**) Graph of the average Pom1-mEGFP intensity per pixel versus the average Blt1-mCherry intensity per pixel along the plasma membrane in the middle focal plane of the cells (as in the inset of panel C). The slope of the linear regression (black dotted line) is not significantly different from zero (p = 0.13; $R^2$ = 0.02).

The online version of this article includes the following source data for figure 4:

**Source data 1.** Related to *Figure 4C, D*.

Figure 4A shows time series of fluorescence micrographs of individual cells from three strains paired with scans of the fluorescent intensity of Pom1-GFP along the lengths of these cells, while *Figure 4B* shows the line profiles of fluorescence of Blt1-mCherry plus Sad1-RFP (magenta) and Pom1-GFP (green) along the lengths of seven to nine mitotic cells of the five different strains. To our knowledge, this is the first time that the distribution of Pom1 was documented in cells with such a wide range of sizes (lengths of 8–40 μm).

Wild-type cells imaged at 22°C concentrated Pom1 at the cell tips throughout interphase and then at the division site in anaphase (*Figure 4A*, second panel from top) as reported previously (*Bähler and Pringle, 1998*) Time-lapse imaging showed that Pom1 followed the constricting contractile ring and disappeared from the division site after the cells separated. Consequently, after cell division, the concentration of Pom1 was higher at the old cell ends than at the new ends. As the growth of the new ends took off, Pom1 concentrated at the new ends with gradients toward the middles of the cells.

The behavior of Pom1 in the very short *wee1-50* cells grown at 36°C for 4 hr and imaged at 22°C was mostly similar to wild-type cells. However, when cells formed a contractile ring, the Pom1 decreased at the cell tips as it distributed more homogeneously along the length of the cells. Then it concentrated at the division site and followed the constricting contractile ring as seen in wild-type cells (*Figure 4A*, upper panel).

The behavior of Pom1 was similar in wild-type, *rga2Δ*, and *rga4Δ* cells grown at 25°C. During interphase, the Pom1-GFP intensity was highest at the tips and low in the middle of the cells. This was exaggerated in the *rga4Δ* cells (*Figure 4B*, second panel from top). Pom1-GFP accumulated at the division site and followed the cleavage furrow as the cells divided.

During interphase at the nonrestrictive temperature Pom1 concentrated at the cell tips of *cdc25-22* mutant cells as in other strains (*Figure 4A*, third panel). However, at the G$_2$/M transition, the Pom1 concentration decreased at both cell tips, and the fluorescence redistributed homogeneously in the cytoplasm. Similarly, when *cdc25-22* cells were grown at a restrictive temperature of 36°C for 4 hr and released at the imaging temperature of 22°C to synchronize them at the G$_2$/M boundary, Pom1 was initially distributed throughout the cytoplasm (*Figure 4A*, fourth panel). Then Pom1 accumulated at both cell tips peaking during anaphase when the contractile ring formed and matured. Similar to wild-type cells, Pom1 concentrated at the cleavage site and appeared to follow the constricting contractile ring (CS) until it redistributed from the division site to the new ends of daughter cells as they took off and grew (*Figure 4A*, third and fourth panels).

The local intensities of Pom1-mEGFP along the plasma membrane in the broad band did not vary significantly in the five strains with a fivefold range of sizes (*Figure 4C*) and 100–400 nodes (*Figure 3H*). The local Pom1 concentration did not vary like the Blt1-mCherry intensity along the plasma membrane in the middle focal plane of these cells (*Figure 4D*). These measurements extend a previous report over a narrower range of cell lengths (*Pan et al., 2014*). We conclude that during

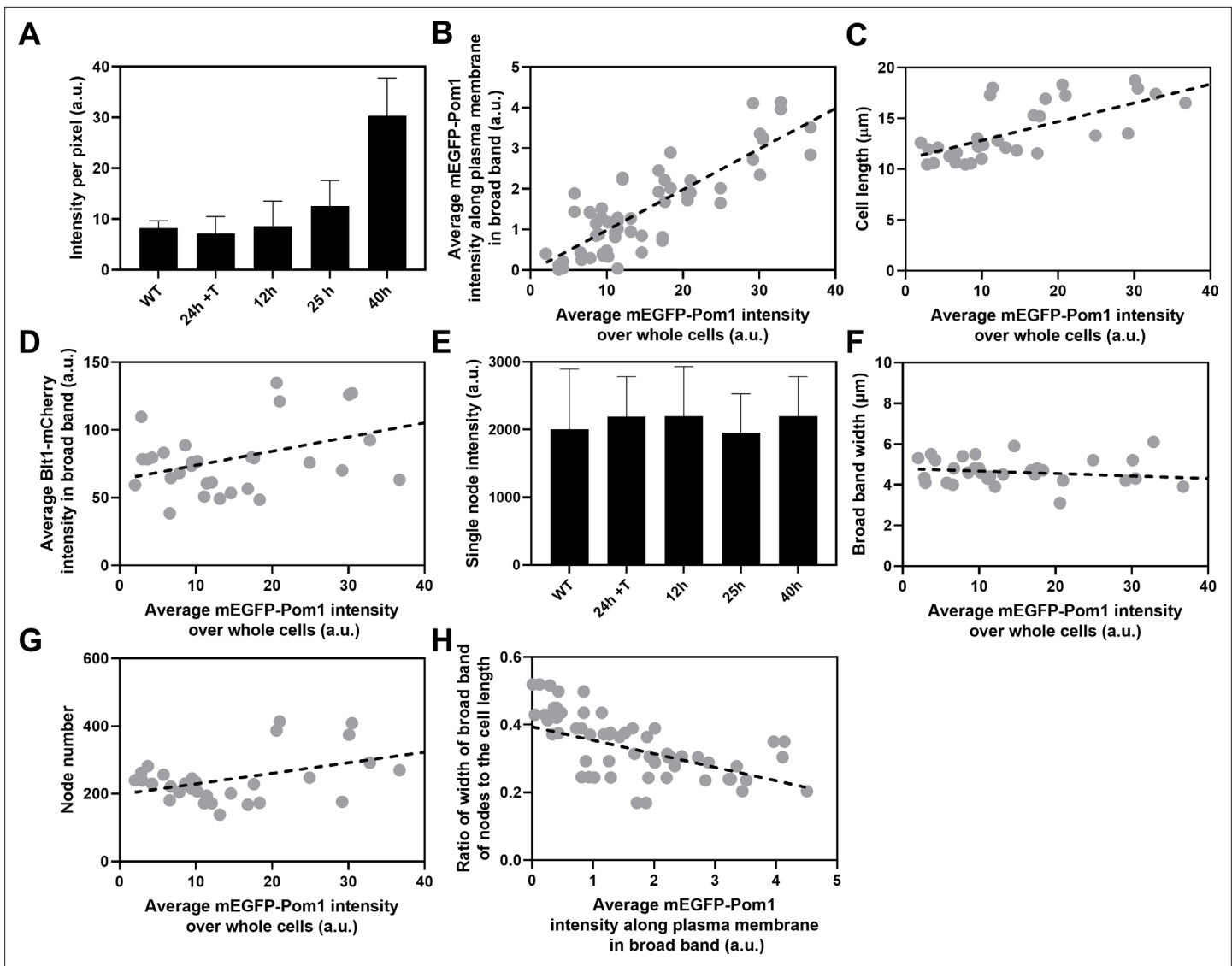

**Figure 5.** Influence of Pom1 concentration on cell size and numbers of cytokinesis nodes. Expression of mEGFP-Pom1 in cells from the *nmt81* promoter under three conditions: in EMM5S with 5 μg/ml thiamine for 24 hr (n = 6 cells) to suppress expression or in EMM5S without thiamine to promote expression for 12 hr (n = 6 cells), 25 hr (n = 8 cells), or 40 hr (n = 8 cells). Gray symbols show individual cells. (**A**) mEGFP-Pom1 expression under four conditions measured as fluorescence intensity per pixel of mEGFP-Pom1 over the whole cells. (**B**) Average mEGFP-Pom1 intensity along the plasma membrane in the broad band of nodes in the middle focal plane of the cells as a function of average mEGFP-Pom1 intensity in whole cells. The slope of the linear regression (black dotted line) is significantly different from zero (p < 0.0001; $R^2$ = 0.7). (**C**) Cell length as a function of average mEGFP-Pom1 intensity in whole cells. The slope of the linear regression (black dashed line) is significantly different from zero (p < 0.0001; $R^2$ = 0.5). (**D**) Blt1-mCherry intensity per pixel in node broad band as a function of average mEGFP-Pom1 intensity in whole cells. The slope of the linear regression (black dashed line) is significantly different from zero (p = 0.01; $R^2$ = 0.2). (**E**) Single node intensities measured by using Blt1-mCherry in WT cells and cells expressing different levels of Pom1. Error bars are standard deviation (SD), p = 0.001 as determined by Tukey's multiple comparison test. (**F**) Width of the broad band region as a function of average mEGFP-Pom1 intensity in whole cells. The slope of the linear regression (black dashed line) is not significantly different from zero (p = 0.3; $R^2$ = 0.04). (**G**) Node number as a function of average mEGFP-Pom1 intensity in whole cells. The slope of the linear regression (black dashed line) is significantly different from zero (p = 0.0017; $R^2$ = 0.3). (**H**) Ratio of width of broad band of nodes to the cell length as a function of average mEGFP-Pom1 intensity along the plasma membrane in the broad band of nodes in the middle focal plane of the cells. The slope of the linear regression (black dashed line) is significantly different from zero (p = 0.0001; $R^2$ = 0.4).

The online version of this article includes the following source data for figure 5:

**Source data 1.** Related to *Figure 5A–H*.

the G$_2$/M phase transition of the cell cycle the local Pom1 concentration in the broad band does not control the variation in node number with cell size.

We used the *nmt81* (no message in thiamine) thiamine-repressible promoter (*Maundrell, 1993*) to control the expression of Pom1 and measured the effects on the cells and their cytokinesis nodes (*Figure 5*). Pom1 expression was lowest under repressing conditions (EMM5S with thiamine) and three times higher than wild-type cells grown in EMM5S without thiamine for 40 hr (*Figure 5A*). The intensity of mEGFP-Pom1 in the broad band along the plasma membrane (*Figure 5B*) and the length of cells undergoing the G$_2$/M transition (*Figure 5C*) increased with the average concentration of Pom1 in the cells. The Blt1-mCherry intensity per pixel in the broad band also increased with the Pom1 concentration (*Figure 5D*), but the single node fluorescence measured with Blt1-mCherry did not change (*Figure 5E*). The width of the broad band of nodes did not change with the total Pom1 (*Figure 5F*). Therefore, the density of nodes in the broad band increased modestly with the cellular Pom1 concentration as the broad band occupied a smaller fraction of the cell length (*Figure 5G*). The ratio of the broad band width to the cell length decreased with the total cellular Pom1, indicating

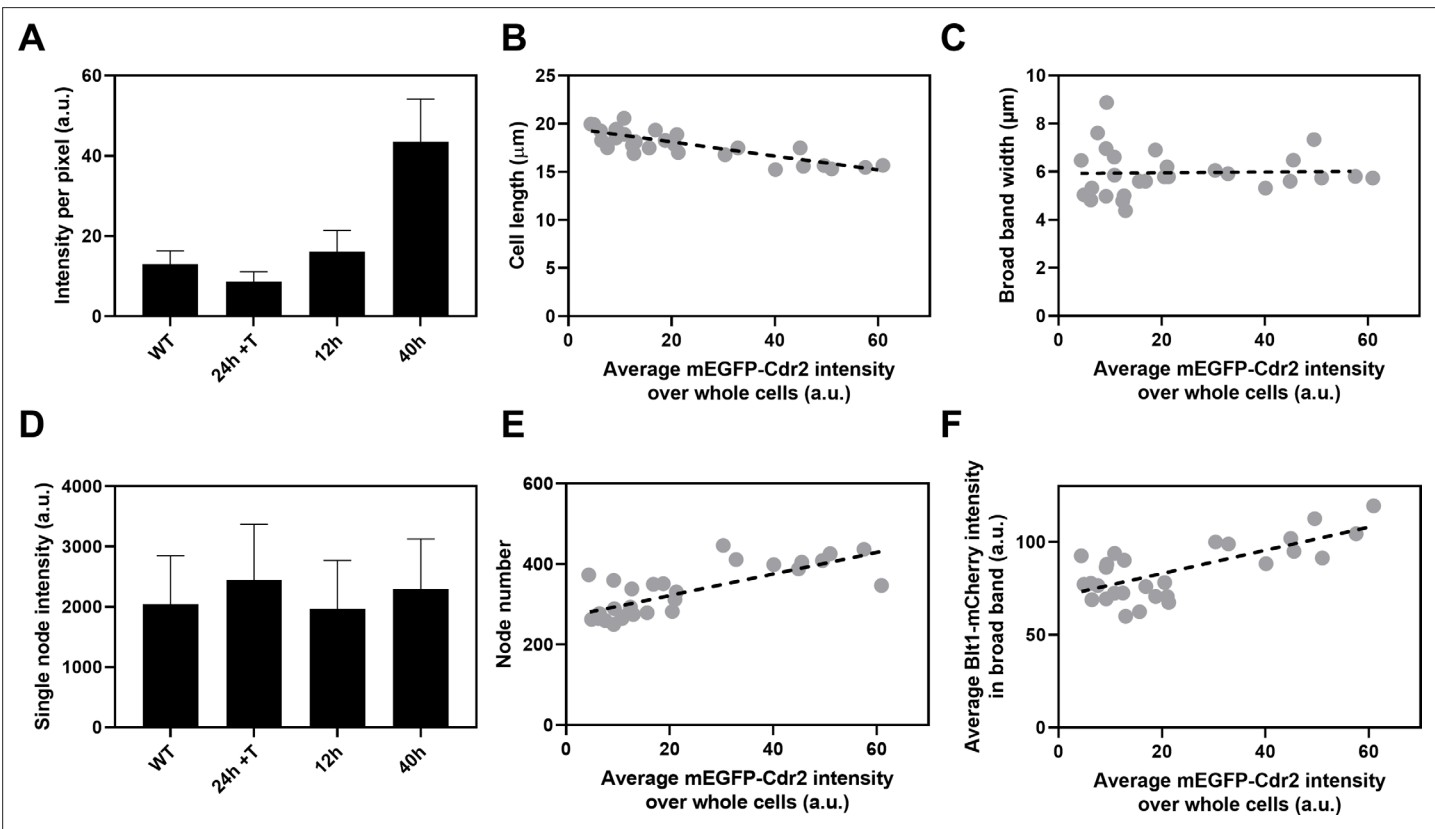

**Figure 6.** Influence of Cdr2 concentration on cell size and the number of cytokinesis nodes. Expression of mEGFP-Cdr2 in wild-type cells (n = 11 cells) and from the *nmt81* promoter under three conditions: in EMM5S with 5 μg/ml thiamine for 24 hr (n = 10 cells) to suppress expression or in EMM5S without thiamine to promote expression for 12 hr (n = 9 cells) or 40 hr (n = 9 cells). Average fluorescence intensities per pixel were measured in sum projection images over whole cells. Gray symbols show individual cells. (**A**) Average mEGFP-Cdr2 fluorescence intensities per pixel over the whole cells under four conditions. (**B**) Cell length as a function of average mEGFP-Cdr2 intensity over whole cells. The slope of the linear regression (black dashed line) is significantly different from zero (p < 0.0001; $R^2$ = 0.6). (**C**) Width of the broad band of nodes as a function of average mEGFP-Cdr2 intensity over whole cells. The slope of the linear regression (black dashed line) is not significantly different from zero (p = 0.9; $R^2$ = 0.0004). (**D**) Single node intensities measured by using Blt1-mCherry in WT cells and cells expressing different levels of Cdr2p. (**E**) Node number as a function of average mEGFP-Cdr2 intensity over whole cells. The slope of the linear regression (black dashed line) is significantly different from zero (p < 0.0001; $R^2$ = 0.6). (**F**) Blt1-mCherry intensity per pixel in the broad band area as a function of average mEGFP-Cdr2 intensity over whole cells. The slope of the linear regression (black dashed line) is significantly different from zero (p < 0.0001; $R^2$ = 0.7).

The online version of this article includes the following source data for figure 6:

**Source data 1.** Related to *Figure 6A–F*.

that Pom1 controls the width of the broad band (*Figure 5H*) as expected from previous work (*Allard et al., 2019*).

## Does Cdr2 control the number of cytokinesis nodes?

The kinase Cdr2 is the scaffold protein for interphase type 1 nodes, which cluster around the middle of the cell and later merge with type 2 nodes to form cytokinesis nodes. To determine whether the numbers of cytokinesis nodes depend on the concentration of Cdr2, we replaced the Cdr2 promoter in the *S. pombe* genome with the *nmt81* thiamine-repressible promoter (*Maundrell, 1993*). To quantify Cdr2 expression in these cells the N-terminus of Cdr2 was tagged with mEGFP. This strain also had a C-terminal RFP tag on Sad1 to mark SPBs and a C-terminal mCherry tag on Blt1 to locate and count cytokinesis nodes. Thiamine suppressed the expression of mEGFP-Cdr2 from the *nmt81* promoter while growth without thiamine produced mEGFP-Cdr2 concentrations higher than in wild-type cells (*Figure 6A*).

The length of the cells as they entered mitosis decreased modestly with the concentration of mEGFP-Cdr2 (*Figure 6B*), but the width of the broad band of nodes did not change (*Figure 6C*). The average single node intensity measured with Blt1-mCherry in cells grown for 12, 25, or 40 hr in EMM5S media and 24 hr in EMM5S + thiamine did not differ from wild-type cells (*Figure 6D*). Both the intensity of Blt1-mCherry per pixel in the broad band and the number of cytokinesis nodes increased with the expression of mEGFP-Cdr2 (*Figure 6E, F*). We conclude that the cellular concentration of Cdr2 has a modest effect on the number and density of cytokinesis nodes and the size of cells entering mitosis but not the width of the broad band of nodes.

## Discussion

### Numbers of cytokinesis nodes

Using Airyscan imaging we measured the number of cytokinesis nodes in *S. pombe* and discovered that the number during early mitosis is proportional to cell size (*Figure 3*). Previous studies underestimated the number of cytokinesis nodes due to limitations of imaging techniques and/or methods used to choose nodes (*Akamatsu et al., 2014*; *Akamatsu et al., 2017*; *Allard et al., 2018*; *Deng and Moseley, 2013*; *Laplante et al., 2016*; *Lenz et al., 2011*; *Pan et al., 2014*; *Vavylonis et al., 2008*; *Willet et al., 2019*). Conventional confocal microscopy cannot resolve closely spaced nodes, so confocal images give the impression that cytokinesis nodes vary considerably in size and fluorescence intensity. Several studies considered only the largest nodes when estimating their numbers. On the other hand, super-resolution single-molecule localization microscopy showed that nodes are uniform in size and composition (*Laplante et al., 2016*). However, the method cannot make a 3D reconstruction of live cells to count the total number of nodes, so the numbers were estimated by extrapolation.

Airyscan imaging overcomes these limitations by allowing 3D reconstructions of live cells at a higher resolution than conventional confocal microscopes. We imaged 40 Z-slices covering the entire thickness of the cells and used the method of *Akamatsu et al., 2017* to count cytokinesis nodes. The first step was to measure the average fluorescence intensity of a sample of single nodes marked with Blt1-mEGFP (*Figure 2D*). The Airyscan microscope resolved single nodes better than the confocal microscope (*Akamatsu et al., 2017*; *Figure 1*), so we observed a large, main peak in the distribution of fluorescence intensities and only a few spots in our sample with higher fluorescence intensities, likely arising from overlapping single nodes. These single nodes move closer to each other to form binary and multiples of single nodes and eventually coalesce into a contractile ring. The second step was to measure the total Blt1-mEGFP fluorescence in the broad band of nodes in the cortex around the equator of the cell, correcting for background cytoplasmic fluorescence. Dividing the total Blt1-mEGFP fluorescence in the broad band by the average fluorescence of a single node gave ~190 single nodes in wild-type fission yeast cells.

Similar measurements on contractile rings in wild-type cells (*Figure 2E*) showed about 85% of total Blt1-mEGFP fluorescence from the broad band of nodes is incorporated into contractile rings. This may be related to the observation that 10 min before SPB divide, about 75% of late interphase nodes contained markers of both type 1 and 2 nodes (*Akamatsu et al., 2014*) at the time they begin to accumulate Myo2, IQGAP Rng2, F-BAR Cdc15, and formin Cdc12 (*Wu et al., 2003*). During 10–15 min after SPBs separate, cytokinesis nodes containing these proteins and Blt1 condense into a contractile

ring (*Vavylonis et al., 2008*; *Wu et al., 2006*). The type 1 node scaffold protein Cdr2 leaves the contractile ring as it forms (*Akamatsu et al., 2014*). Anillin Mid1 is released from the fully formed contractile rings as unconventional myosin-II Myp2 joins the ring (*Wu et al., 2003*). It is possible that the 25% of cytokinesis nodes with Blt1 but lacking Cdr2 are left behind when the contractile ring forms, because they failed to grow actin filaments or were deficient in Myo2, which are required for contractile ring formation.

Our approach may be useful to extend studies on how the density of nodes marked with Cdr2 senses the cell surface area, activates cyclin-dependent kinase Cdk1 through inhibitory phosphorylation of *wee1*, and promotes the transition into mitosis (*Facchetti et al., 2019b*; *Pan et al., 2014*; *Russell and Nurse, 1987*; *Simanis and Nurse, 1986*). Rather than measuring the total intensity of Cdr2 around the equator (*Facchetti et al., 2019b*; *Pan et al., 2014*; *Russell and Nurse, 1987*; *Simanis and Nurse, 1986*) or counting only bright nodes (*Pan et al., 2014*), we measured the number of single nodes marked with Blt1 and found that their density on the membrane in the broad band varied with cell sizes except in *cdc25-22* mutant cells (*Figure 3K*). Using our method to count single nodes marked with Cdr2 or Blt1 may provide further insights into the relation of node numbers to the timing of cell division.

## Cytokinesis node number in broad bands scales with cell size

Starting from the observation that the number of interphase nodes in wild-type cells increases with the twofold increase in cell size during the cell cycle (*Akamatsu et al., 2017*; *Allard et al., 2018*; *Deng and Moseley, 2013*; *Pan et al., 2014*), we measured the number of cytokinesis nodes in wild type and mutant cells that varied fivefold in size as they entered mitosis (*Figure 3*). To avoid the time-dependent compositional changes in cytokinesis nodes during their ~22 min life span and to be able to compare the number of cytokinesis nodes as well as the cell size among the wild-type and four mutant cells, we restricted our measurements to a specific cell cycle time point. The average fluorescence intensity of the single nodes was the same in wild-type and these mutant cells (*Figure 3G*). The total Blt1-mEGFP fluorescence in a broad band of nodes and therefore the total number of nodes increased with cell size measured as volume (*Figure 3H*), length (*Figure 3I*), or surface area (*Figure 3—figure supplement 1A*).

## Influence of the kinases Pom1 and Cdr2 on the broad band of cytokinesis nodes

We investigated the roles of Pom1, the kinase that restricts type 1 nodes from the poles of cells, and Cdr2, the kinase and scaffold protein for type 1 nodes, in setting the number of cytokinesis nodes. Pom1 forms a gradient from the cell tips to their middles and phosphorylates Cdr2 at multiple sites. Phosphorylation of one site reduces Cdr2 oligomerization and membrane binding at the poles, while phosphorylation of another site inhibits Cdr2 from phosphorylating the Wee1 kinase (*Bhatia et al., 2014*; *Deng and Moseley, 2013*; *Martin and Berthelot-Grosjean, 2009*; *Moseley et al., 2009*; *Rincon et al., 2014*). However, both *cdr2Δ* and *pom1Δ* cells form nodes and complete cytokinesis, although the cleavage site is off-center in *pom1Δ* cells because they only exclude nodes from one pole by an unknown mechanism (*Martin and Berthelot-Grosjean, 2009*; *Moseley et al., 2009*).

*Pan et al., 2014* found a constant concentration of Pom1 along the equatorial plasma membrane of wild-type cells as they grew twofold, and we extended the range of sizes to fivefold. Thus, the scaling of interphase or cytokinesis node numbers with cell size is not a consequence of differences in the local concentration of Pom1 in the broad band. Furthermore, *Pan et al., 2014* also reported that total Cdr2 and Cdr2 in interphase nodes increased with cell size during the cell cycle in cells with or without Pom1. This is very strong evidence that Pom1 does not control type 1 node formation in the cell middle directly, rather as we explain below, Pom1 controls the cytokinesis node number indirectly via its influence on cell size.

Although neither Cdr2 nor Pom1 controls the scaling of interphase node number with cell size, both can influence the number and density of cytokinesis nodes in the broad bands of mitotic cells. Artificially varying the total cellular concentration of Pom1 revealed the following dependencies as cells entered mitosis: the lengths of the cells (*Figure 5C*), the concentration of Pom1 along the plasma membrane in the broad band (*Figure 5B*), and the numbers of cytokinesis nodes (*Figure 5G*) all increased with the cellular concentration of Pom1, while both the fluorescence of single nodes

(*Figure 5E*) and the width of the broad band (*Figure 5F*) were constant. Therefore, high Pom1 increased the density of cytokinesis nodes (*Figure 5F*) as the broad band occupied a smaller fraction of the cell (*Figure 5H*). The numbers of cytokinesis nodes in cells expressing high Pom1 concentrations exceeded the numbers in cells of similar lengths and normal Pom1 concentrations (*Figure 3I*).

On the other hand, cells expressing low concentrations of Pom1 were shorter when they divided (*Figure 5C*) with fewer cytokinesis nodes (*Figure 5G*) than cells with high Pom1 concentrations. The node numbers were higher than in cells of a given length with normal Pom1 levels (*Figure 3I*).

Artificially increasing the total cell concentration of Cdr2 (*Figure 6*) reduced cell length at the time of division (*Figure 6B*) but did not change the width of the broad band of cytokinesis nodes (*Figure 6C*), so the broad band occupied a larger fraction of the cell surface. High Pom1 concentrations had the opposite effects on cell size (*Figure 5C*) and broad band width (*Figure 5F*). On the other hand, the number of cytokinesis nodes increased with the concentration of Cdr2 (*Figure 6E*). The short cells expressing high Cdr2 concentrations had almost double the number of cytokinesis nodes in their slightly wider broad bands than cells of similar lengths with wild-type levels of Cdr2 (*Figure 3I*). Longer cells expressing low concentrations of Cdr2 had numbers of cytokinesis nodes similar to cells of the same lengths in *Figure 3I*. The plot of cytokinesis node number versus Cdr2 concentration extrapolates to more than 190 nodes when the Cdr2 concentration is zero (*Figure 6E*), as expected for cells longer than wild-type cells at division (*Figure 6B*). Therefore, Cdr2 can drive cytokinesis node numbers above the basal level of about 190.

## Mechanisms
### Cell size dominates in setting the basal number of nodes

Low SIN activity during interphase is a prerequisite for the assembly of type 1 interphase nodes (*Pu et al., 2015*), but cell size has a dominant influence in setting the number of nodes in wild-type cells. Accordingly, as cells grow, the number of interphase nodes doubles as they mature into cytokinesis nodes. A limiting amount of an essential node protein might set the number of nodes, but only about half of each of the known node proteins assemble into nodes (*Akamatsu et al., 2017*). Alternatively, a cell-size-dependent posttranslational modification of one of these proteins may limit the number of nodes. Phosphorylation of the type 1 node scaffold protein Cdr2 by Pom1 is a candidate mechanism,

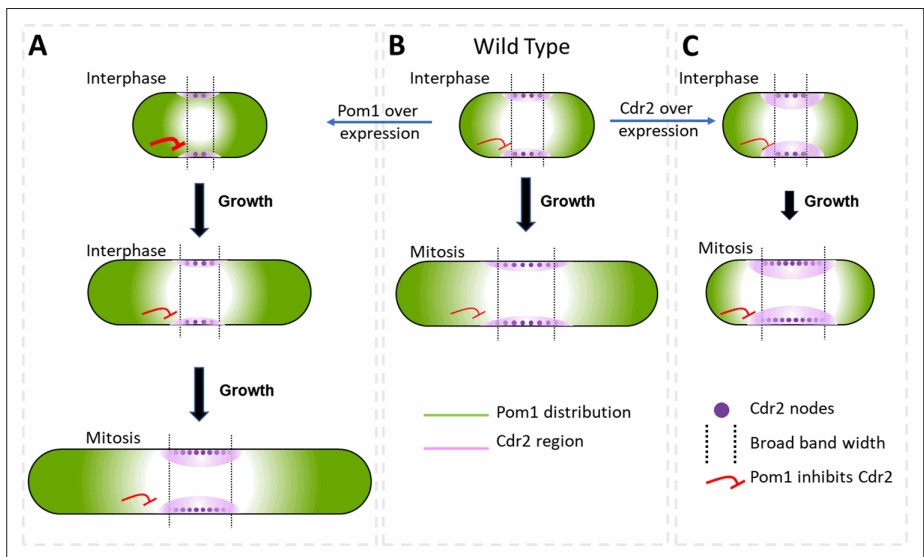

**Figure 7.** Factors that influence cell size and the broad band of cytokinesis nodes in (**A**) cells overexpressing Pom1, (**B**) wild-type cells and (**C**) cells overexpressing Cdr2. The gradient of Pom1 along the cell plasma membrane is shown in green. The distribution of Cdr2 on the plasma membrane is shown in magenta and called the Cdr2 region. Purple dots show the density of Cdr2 nodes. Blunt end red lines show the inhibitory action of Pom1 on Cdr2, with the line thickness showing the strength. Lengths of black arrows show the time spent growing. The two dotted lines show the borders of the broad band of nodes.

but deletion of Pom1 does not affect the number of nodes (**Bhatia et al., 2014**; **Pan et al., 2014**). Nodes turn over continuously, so the rates of node formation and turnover must have a strong influence on their numbers. However, we do not yet know whether one protein at a time or whole nodes turn over or how this process is controlled. All these options are attractive subjects for future research.

## Pom1 and Cdr2 modulate a basal number of nodes

We find that cell size is the primary determinant of the number of cytokinesis nodes (**Figure 3**) but that the cellular concentrations of the Cdr2 and Pom1 kinases can independently influence cell size and drive the number of cytokinesis nodes above a baseline (**Figure 7**). The modulating effect of the two kinases is seen in plots of node number vs. the concentrations of either kinase, which intercept the *y*-axis well above zero (**Figures 3H, 5G, and 6E**). The presence of a basal number of cytokinesis nodes is expected, since cells without either Cdr2 or Pom1 form cytokinesis nodes **Almonacid et al., 2009**; **Martin and Berthelot-Grosjean, 2009**; **Moseley et al., 2009**. We propose that interactions of Pom1 with Cdr2 in type1 interphase nodes modulate the cell size, which in turn is the dominant determinant of the basal numbers of cytokinesis nodes in mitotic cells (**Figure 3H**). On top of this basal number of cytokinesis nodes, the concentrations of Cdr2 (**Figure 6E**) and Pom1 (**Figure 5G**) in mitotic cells can increase the number of cytokinesis nodes.

Pom1 regulates interphase node localization by phosphorylating Cdr2, which reduces its oligomerization and membrane binding (**Bhatia et al., 2014**; **Rincon et al., 2014**). This direct effect of Pom1 on Cdr2 is one way the gradient of Pom1 from the poles of the cell can restrict nodes to a broad band around the equator (**Figure 7**). As cells grow, the overlap of the Pom1 gradient with Cdr2 in the middle of the cell decreases.

In addition, Pom1 regulates cell size at the time of division through a series of three inhibitory phosphorylation reactions: Pom1 phosphorylation inhibits the kinase activity of Cdr2; Cdr2 phosphorylates and inhibits the Wee1 kinase; Wee1 phosphorylates inhibitory sites on Cdk1, the master mitotic kinase that promotes entry into mitosis (**Allard et al., 2018**; **Breeding et al., 1998**; **Russell and Nurse, 1987**). As cells grow, the influence of Pom1 on Cdr2 and Wee1 diminishes, contributing to the activation of Cdk1 and the transition into mitosis (**Figure 7B**; **Allard et al., 2018**, **Allard et al., 2019**; **Martin and Berthelot-Grosjean, 2009**; **Moseley et al., 2009**).

Based on these mechanisms, overexpression of Cdr2 might increase the number of cytokinesis nodes (**Figures 6E and 7C**) by mass action. In addition, mass action may drive Cdr2 to inhibit enough Wee1 to promote premature mitosis, resulting in short daughter cells (**Figure 6B**) with more nodes packed more densely in a broad band of normal width (**Figures 6C, E and 7C**).

Unexpectedly, the number of cytokinesis nodes also increases with the Pom1 concentration (**Figure 5G**). Based on the Pom1 to Cdk1 pathway, one expects high concentrations of Pom1 to inhibit the assembly of Cdr2 into nodes. However, inhibiting Cdr2 results in high Wee1 activity that inhibits Cdk1, delays the cell cycle, and allows cells to grow longer (**Figure 7A**). The dominant effect of cell size on driving node formation produces more nodes (**Figure 5C**).

## Pom1 and Cdr2 influence the width of the broad band and density of nodes

Wild-type cells form type 1 nodes around the daughter nuclei and use a gradient of Pom1 from the poles to confine these nodes to the equator throughout interphase (**Figure 7B**). As cells elongate during interphase, three parallel events result in twofold longer cells with twice the number of nodes located in a twofold wider broad band ready for mitosis. First, the Pom1 concentration on the plasma membrane is low in a zone in the middle of the cell that widens in proportion to a wide range of cell sizes (**Figures 3J, 4C, and 7B**; **Bhatia et al., 2014**; **Pan et al., 2014**). Second, except in *cdc25-22* mutant cells, the density (**Figure 3K**) and the total number of nodes marked with Blt1 (**Figure 3H**) or Cdr2 (**Bhatia et al., 2014**; **Pan et al., 2014**) in this broad band increases with cell size in cells with or without Pom1 (**Bhatia et al., 2014**; **Pan et al., 2014**). Third, the broad band widens in proportion to cell size (**Figure 3J**). Two additional factors contribute to limiting nodes to the equator. One is the uncharacterized activity that excludes nodes from one pole in the absence of Pom1 (**Martin and Berthelot-Grosjean, 2009**; **Moseley et al., 2009**). The other is anillin, Mid1, given that nodes spread over the entire plasma membrane in *mid1Δ* cells (**Saha and Pollard, 2012**).

The experimental conditions may explain why *cdc25-22* mutant cells did not pack their nodes more densely as they grow larger than the other strains (*Figure 3K*). As cells of the other strains move naturally through the cell cycle, their nodes sense the surface area and use the Cdr2–Wee1–Cdk1 pathway to reach the mitotic threshold of node density or number. On the other hand, we hold the *cdc25-22* mutant cells at the $G_2/M$ transition until releasing them into mitosis. As the arrested cells grow at their tips, the extra nodes formed after the threshold time have room to spread out ever further from the edges of node broad band. The large nucleus in the *cdc25-22* mutant cells and the effect of ER–plasma membrane contacts on node distribution (*Zhang et al., 2016*) need further attention.

The concentrations of Pom1 and Cdr2 to Pom1 also influence the width of the broad band and the density of nodes therein (*Figure 7*). Overexpression of Cdr2 drives both node formation and premature cell division, while the natural level of Pom1-driven inhibition from the poles does not allow space for new nodes at the edges of the broad band, so the surface density of cytokinesis nodes is higher (*Figure 7C*). Cells over expressing Pom1 also have a higher density of nodes around their equators owing to more nodes (*Figure 5G*) in broad bands of normal widths (*Figure 5F*). Like short wild-type interphase cells (*Figure 7B*), overexpression of Pom1 increases its concentration in the middle of cells (*Figure 5B*) and suppresses the spread of the broad band in spite of delayed mitosis increasing the length of these cells (*Figures 5C and 7A*).

## Influence of cell shape

The shape of the cell can also influence the number of nodes. For example, the large diameter but short *rga4Δ* mutant cells lacking the RhoGAP Rga4 are normal in many ways (single node intensity, numbers, and densities of nodes for their lengths) but due to their large diameters, they are an outlier with fewer cytokinesis nodes than expected for cells with a given volume. In addition, the gradient of Pom1 from the cell tips is steeper in *rga4Δ* mutant cells than in wild-type cells and the Blt1-mCherry + Sad1-RFP fluorescence was spread more widely across the middle of *rga4Δ* mutant cells than in wild-type or other strains (*Figure 4B*, second and third panels from the top).

**Table 1.** Strains used in this study.

| Strain | Genotype | Source |
|---|---|---|
| WS40 | h− blt1-mEGFP:KanMX6 sad1-RFP:KanMX6 ade6-M21X leu1-32 ura4-Δ18 | Laboratory Stock |
| WS46 | h− *cdc25-22* blt1-mEGFP:KanMX6 sad1-RFP:KanMX6 ade6-M21X leu1-32 ura4-Δ18 | This study |
| WS47 | h− *wee1-50* blt1-mEGFP:KanMX6 sad1-RFP:KanMX6 ade6-M21X leu1-32 ura4-Δ18 | This study |
| WS51.1 | h− *rga2Δ* blt1-mEGFP:KanMX6 sad1-RFP:KanMX6 ade6-M21X leu1-32 ura4-Δ18 | This study |
| WS52.1 | h− *rga4Δ* blt1-mEGFP:KanMX6 sad1-RFP:KanMX6 ade6-M21X leu1-32 ura4-Δ18 | This study |
| WS59 | h− pom1-GFP kanMX6 blt1-mCherry natMX6 sad1-mRFP kanMX6 ade6-M21X leu1-32 ura4-Δ18 | Laboratory Stock |
| WS54 | h− *rga2Δ* pom1-GFP kanMX6 blt1-mCherry natMX6 sad1-mRFP kanMX6 ade6-M21X leu1-32 ura4-Δ18 | This study |
| WS55 | h− *rga4Δ* pom1-GFP kanMX6 blt1-mCherry natMX6 sad1-mRFP kanMX6 ade6-M21X leu1-32 ura4-Δ18 | This study |
| WS56 | h− *cdc25-22* pom1-GFP kanMX6 blt1-mCherry natMX6 sad1-mRFP kanMX6 ade6-M21X leu1-32 ura4-Δ18 | This study |
| WS57 | h− *wee1-50* pom1-GFP kanMX6 blt1-mCherry natMX6 sad1-mRFP kanMX6 ade6-M21X leu1-32 ura4-Δ18 | This study |
| WS70 | h− HygMX6-Pnmt81-mEGFP-Pom1 blt1-mCherry natMX6 sad1-mRFP kanMX6 ade6-M21X leu1-32 ura4-Δ18 | This study |
| WS71 | h− KanMX6-Pnmt81-mEGFP-Cdr2 blt1-mCherry natMX6 sad1-mRFP kanMX6 ade6-M21X leu1-32 ura4-Δ19 | This study |
| CL181 | h+ KanMX6-Pmyo2-mEGFP-myo2 Sad1-RFP-KanMX6 ade6-M216 his3-D1 leu1-32 ura4-D18 | Laboratory Stock |
| JW875 | h+ cdc25-22 kanMX6-Pmyo2-GFP-myo2 ade6-M210 leu1-32 | Laboratory Stock |

## Incorporation of the proteins of cytokinesis nodes into contractile rings

Most of the proteins comprising cytokinesis nodes during mitosis (Blt1, anillin Mid1, F-BAR Cdc15, IQGAP Rng2, and Myo2) are initially incorporated into contractile rings (*Goss et al., 2014*; *Wu and Pollard, 2005*). Thereafter, Myo2 is retained and concentrated along with newly arriving Myp2, while first Mid1 and then subsequently actin are lost in proportion to the declining circumference of the constricting contractile ring (*Courtemanche et al., 2016*; *Malla et al., 2021*). This gave the strong impression that the core of the cytokinesis nodes, with the exception of Mid1, remained intact in the contractile ring. However, we report here that the fraction of Blt1 in cytokinesis nodes incorporated into contractile rings varies: nearly 90% in *rga4Δ* mutant cells, 85% in wild-type cells, and 40–70% in *cdc25-22*, *wee1-50*, and *rga2Δ* mutant cells. In the case of the longest *cdc25-22* mutant cells, a much larger fraction of the Myo2 from the cytokinesis nodes was incorporated into the rings than Blt1 (*Figure 3—figure supplement 2*). Arresting and then releasing the cell cycle of the *cdc25-22* mutant cells at the $G_2/M$ transition may contribute to this imbalance, which is greater than in any of the strains allowed to cycle normally. In any case, this new observation indicates that nodes observed in rings by single-molecule localization microscopy (*Laplante et al., 2016*) are likely to have different compositions than cytokinesis nodes. Documenting and characterizing this maturation of nodes transitioning into contractile rings is worthy of further exploration.

## Materials and methods

### Yeast strains

*Table 1* lists the strains used in this work. We used standard methods for *S. pombe* growth and genetics (*Moreno et al., 1991*). We made most observations on a strain with SPBs marked by Sad1-RFP, and nodes marked by Blt1-mEGFP (*Goss et al., 2014*). The sequences encoding the fluorescent proteins were incorporated into the haploid genome, so all copies of these proteins were tagged. The yeast strains *rga2Δ* (FC3158) and *rga4Δ* (FC3157) were a kind gift from the laboratory of Prof. Fred Chang (University of California, San Francisco). All the other strains were obtained from laboratory stocks or by crossing laboratory stock strains. Strains were confirmed by growth on selective media and microscopy. Except for *Figures 5 and 6*, cells were grown in YE5S medium for 24 hr at 25°C and then overnight in synthetic EMM5S medium at 25°C before imaging.

To control the expression of Cdr2 and Pom1 (*Figures 5 and 6*), we inserted by homologous recombination the weak *nmt81* promoter along with mEGFP upstream of the open reading frames for Cdr2 or Pom1 in their chromosomal loci. Cells were grown in the EMM medium for 25 hr at 25°C to induce moderate expression or 40 hr at 25°C to overexpress mEGFP-Cdr2 or mEGFP-Pom1. Cells were grown in 5 µg/ml thiamine for >24 hr at 25°C to repress the expression of these two fusion proteins. The $OD_{595}$ of the cultures was kept below 0.6 over the entire period.

### Airyscan microscopy

Imaging was performed on LSM 880 laser scanning confocal microscope (ZEISS) equipped with 100x Plan Apochromatic 1.46 NA oil objective, an Airyscan super-resolution module, GaAsP detectors, and Zen Black acquisition software (ZEISS). We acquired 40 Z-slices with a step size of 170 nm covering the entire volume of a cell. Pixel size was 41 nm by 41 nm. Images were collected with a pixel dwell time of 1.02 µs and an image size of 520 × 520 pixels corresponding to 22 × 22 µm. Cells were imaged for mEGFP with a 488 nm laser and BP 420–480 excitation filter and BP 495–620 emission filter; and for RFP with a 561 nm laser and with a BP 495–550 excitation filter and LP 570 emission filter. The pixel dwell time, laser intensity, and detector gain were kept low to avoid saturation and photobleaching during image acquisition. To increase signal-to-noise ratio and resolution, acquired images were processed by 3D Airyscan filter strength 7.0 with Zen Black software.

We identified cytokinesis nodes as spots of Blt1-mEGFP fluorescence around the equator of cells with two SPBs marked by Sad1-RFP, indicating that the cells were in mitosis. Since these cytokinesis nodes formed contractile ring within 10 min of their formation, we chose cells with broad band of nodes and neglected others which show partially formed contractile rings. We identified and measured the fluorescence intensity of single cytokinesis nodes using the following procedure to process the fluorescent images. First, we used ImageJ software to make a maximum intensity projection from 40 Z-slices of Airyscan images containing all of the fluorescence spots in the broad

band of a cell (*Figure 2A*). These spots varied considerably in size and brightness depending on the number of 'single nodes' clustered in a diffraction-limited spot (*Akamatsu et al., 2017*; *Laplante et al., 2016*). We searched for small spots and marked their *X–Y* positions. Then we inspected the fluorescence along the *Z*-axis at these *X–Y* positions in 40 *Z*-slices. To qualify as a single node, a spot had to satisfy two conditions: (1) the fluorescence was within a circle 12-pixels (~0.5 μm) in diameter, and (2) the fluorescence was present in more than 3 and up to 7 consecutive *Z*-slices, with a peak in the middle slice and diminishing toward either or both sides. This middle slice was designated as the *Z*-axis position of the node. The *X–Y–Z* positions of each node were stored in the ROI manager of ImageJ software and exported to MATLAB using the add-on @ReadImageJROI (*Muir, 2021*). We developed a MATLAB code for further analysis (*Source code 1*). We sum projected the 7 *Z*-slices containing the putative single node and measured the total fluorescence intensity within a circle 12-pixels in diameter at the *X–Y* position of the node (*Figure 2B*, black dotted circle). From the same sum-projected image of 7 *Z*-slices, we measured the average cytoplasmic fluorescence from 8 circles 12-pixels in diameter from regions of cytoplasm flanking the node broad band (*Figure 2B*, blue dotted circles). We subtracted this average cytoplasmic fluorescence from the fluorescence of the node to remove the background. For each node, we used SUM projected image of 7 *Z*-slices containing all the fluorescence of the same node to calculate its total and average cytoplasmic background as explained above. A histogram of the sample of fluorescence intensities of single nodes after subtracting their respective average cytoplasmic background was fit with Gaussian distribution using Solver add-in Microsoft Excel, which minimizes the sum of the chi-squared values, showed a dominant peak. The analysis method is similar to Figure 2 of *Akamatsu et al., 2017*, but the higher resolution of Airyscan microscopy separated some closely space nodes that would be counted as one by confocal microscopy. To calculate the mean value of fluorescence of single nodes, we considered only the nodes with fluorescence values less than or equal to twice the peak of the Gaussian curve fitted to the distribution and average it. In *Figures 2 and 3G*, the Gaussian fits show the distribution of fluorescence of only single nodes on histograms.

To obtain the total fluorescence of a broad band of nodes, we sum projected 40 *Z*-slices of Airyscan images and measured the fluorescence of an ROI drawn around the broad band of nodes (*Figure 2C*, red dashed line area). The mean pixel intensity from polygonal ROIs from either side of the broad band (*Figure 2C*, yellow dashed line) was averaged and multiplied by the number of pixels in the broad band ROI to calculate the background fluorescence in the volume of the broad band of nodes. This value was subtracted from the fluorescence in the broad band of nodes to remove the background. To estimate the number of nodes in a cell we divided the total fluorescence of the broad band of nodes after subtracting the background by the average total fluorescence of individual single nodes after subtracting the background.

In *Figure 3K*, we calculated the density of nodes in broad bands by taking into account the cell diameters, which were the same in wild-type, *wee1-50*, and *cdc25-22* mutant cells but larger in *rga4Δ* cells and smaller in *rga2Δ* cells.

## Confocal microscopy

To compare the confocal and Airyscan imaging shown in *Figure 1* we obtained confocal images using the same Airyscan detector. To make and improve the signal-to-noise ratio comparable to Airyscan microscopy we used the pinhole 0.6 AU, 8× averaging, linear constrained iterative deconvolution, and twice the 488 nm laser intensity implemented in Airyscan imaging. We acquired 20 *Z*-slices with step size 340 nm covering the entire fluorescence of a cell with pixel size 82 nm by 82 nm. Images were processed for linear constrained iterative deconvolution using the ImageJ plugin 'Deconvolution'.

To determine the Pom1 localization in *Figure 5*, we used an Olympus IX-71 microscope with a 100x, numerical aperture (NA) 1.4 Plan Apo lens (Olympus), and a CSU-X1 (Andor Technology) confocal spinning-disk system equipped with an iXON-EMCCD camera (Andor Technology). We used Coherent OBIS 488 nm LS 20 mW and OBIS 561 nm LS 20 mW lasers for excitation. Images were acquired using Micromanager 1.4 (*Edelstein et al., 2014*).

We acquired 15 *Z*-slices covering the fluorescence of entire cells every 2–5 min with 0.5 μm spacing between slices; 22 μW 561 nm laser power (measured after the objective) and 50 ms exposures; 89 μW 488 laser power and 200 ms exposures. Images were sum projected and corrected for camera noise and uneven illumination. Photobleaching was compensated by choosing a small ROI in both

channels that was present in another nondividing cell throughout the acquisition and using the Fiji plugin 'Bleach correction' for Exponential Fitting.

We studied cytokinesis nodes in cells during the interval between SPB separation and the formation of the contractile ring. The intensity per pixel of an ROI where no cells were present was subtracted from both channels to remove substrate background intensity. Then a line equal to cell width was drawn along the length of the cell, and a line profile was obtained using the ImageJ-Plot profile plugin, which generated a graph of intensity per pixel across the width of a cell vs the cell length. The intensity per pixel of wild-type cells not expressing a fluorescent protein was subtracted from each point to remove auto-fluorescence.

For each strain except in the *cdc25-22* mutant strain, 10–12 line profiles were obtained as above and aligned at the centers of the cells. To measure the distributions of the fluorescence from Pom1-GFP or Blt1-mCherry plus Sad1-RFP at different phases of cell cycles of wild-type, *wee1-50*, and *cdc25-22* mutant cells, we plotted the intensities per pixel across the width of cell vs the cell length across the cell cycle (*Figure 4B*). The temperature-sensitive *cdc25-22* mutant cells were grown at the restrictive temperature of 36°C for 4 hr and released for imaging at 22°C. The diffuse fluorescence of Pom1-GFP was higher on one side of the nucleus than the other, so the line profiles were aligned with the higher concentration of Pom1 on the same side (*Figure 4B*, lower panel). We measured the width of the broad band of cytokinesis nodes as the zone where the Blt1-mEGFP signal was >20% of the peak value.

We outlined manually whole cells or the broad band area and used the ImageJ plugin 'Measure' to measure the Mean intensity in these areas from the intensity per pixel calculated as the total intensity in all the pixels in a region divided by the number of pixels, which is equivalent to the density (*Figures 5 and 6*).

In *Figure 4C*, the Pom1-mEGFP intensity along the equatorial plasma membrane was measured from confocal images of the middle focal plane (thickness ~0.5 μm) of the cells after subtracting the average background intensity from outside the cells in the same image. A rectangle 3 pixels wide (~0.4 μm) and as long as the broad band of nodes was manually drawn over the plasma membrane of cells in the image and used to calculate the Pom1-mEGFP intensity per pixel.

The mEGFP-Pom1 intensity in *Figure 5B* was measured similar to *Figure 4C* from the sum projected image of three consecutive middle focal planes Airyscan images (thickness ~0.5 μm) of the cells. The average pixel intensity outside cells was subtracted from the sum image. A rectangle 10 pixels wide (~0.4 μm) and as long as the broad band of nodes was manually drawn over the plasma membrane of cells in the image and used to calculate the Pom1-mEGFP intensity per pixel in the broad band as shown in *Figure 4C*, inset.

## Acknowledgements

Research reported in this publication was supported by the National Institute of General Medical Sciences of the National Institutes of Health under award number R01GM026132. The content is solely the responsibility of the authors and does not necessarily represent the official views of the National Institutes of Health. We thank Prof. Fred Chang for the *rga4Δ* and *rga2Δ* mutant strains.

## Additional information

### Funding

| Funder | Grant reference number | Author |
| --- | --- | --- |
| National Institute of General Medical Sciences | R01GM026132 | Wasim A Sayyad |

The funders had no role in study design, data collection, and interpretation, or the decision to submit the work for publication.

## Author contributions
Wasim A Sayyad, Data curation, Validation, Investigation, Visualization, Methodology, Writing – review and editing, Conceptualization, Formal analysis, Software, Writing – original draft; Thomas D Pollard, Resources, Data curation, Supervision, Funding acquisition, Investigation, Visualization, Methodology, Writing – original draft, Project administration, Writing – review and editing

## Author ORCIDs
Wasim A Sayyad ⓘ http://orcid.org/0000-0002-6064-2860
Thomas D Pollard ⓘ http://orcid.org/0000-0002-1785-2969

## Ethics
This work did not involve human subjects, animals or clinical trials.

## Decision letter and Author response
Decision letter https://doi.org/10.7554/eLife.76249.sa1
Author response https://doi.org/10.7554/eLife.76249.sa2

---

# Additional files

## Supplementary files
- Transparent reporting form
- Source code 1. Matlab code to extract data from Airyscan images to make *Figures 2, 3, 5 and 6*.

## Data availability
All data generated or analyzed during this study are included in the manuscript and supporting files.

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
