## [Editor Report]

This manuscript reports important quantitative results on cytokinetic nodes using fission yeast as a model. Using an Airyscan microscopy the authors develop images with ~ 140 nm resolution and estimate ~ 190 cytokinetic nodes per cell prior to actomyosin ring assembly. The work also describes important conclusions on scaling parameters, all of which will help refine quantitative models of cytokinesis.

---

## [Decision Letter]

**Decision letter after peer review:**

[Editors’ note: the authors submitted for reconsideration following the decision after peer review. What follows is the decision letter after the first round of review.]

Thank you for submitting the paper "The number of cytokinesis nodes in mitotic fission yeast scales with cell volume" for consideration by *eLife*. Your article has been reviewed by 3 peer reviewers, and the evaluation has been overseen by a Reviewing Editor and a Senior Editor. The following individual involved in review of your submission has agreed to reveal their identity: Dimitrios Vavylonis (Reviewer #3). The Reviewing and Senior Editors were not involved in the peer review, but oversaw the discussions and drafted the final letter.

Comments to the Authors:

We are sorry to say that, after consultation with the reviewers, we have decided that this work will not be considered further for publication by *eLife*.

There was a great deal of enthusiasm from all three Reviewers and the editors for a compelling and quantitative elucidation of aspects of cytokinesis and in particular quantitation of cytokinetic nodes. However, three major concerns have been raised about 1. the concept of unitary nodes and the differences between the dimmer cytosolic spots is unclear and some pitfalls in the approach and interpretations were raised. Furthermore, as in the attached files that Reviewer 1 has prepared based on the raw data, there is concern about counting errors and how the cytosolic fluorescent spots are differentiated from membrane localized "unitary nodes". 2. The extent of broad interest in the work beyond the fission yeast community and 3. the somewhat unclear epistasis between cdr2, pom1 in cytokinetic node establishment and how these relate to mitotic nodes.

In addition to the specific comments reproduced verbatim from the three Reviewers, the following issues were raised during the discussion between the Reviewers and RE.

We all have now looked at Reviewer 1's files, as well as the raw data provided by the authors. We all agree that the unusual intensity fluctuations of the background should be discussed. From this analysis, it seems that roughly, 8/10 ROIs would correspond to what we can reliably call nodes (being close to the membrane and stationary structures with signal over several z slices). But 2/10 ROIs seem to be in the cytoplasm. This observation could imply some counting errors (not sure how much, maybe 20%?). Then there is the separate issue of possibly *not* counting bright nodes because they may have considered them not to be unitary (they say "We searched for dim spots" on line 680).

*Reviewer #1 (Recommendations for the authors):*

Cytokinesis nodes are plasma membrane-associated assemblies of multiple proteins including myosin-II and play key roles in fission yeast cytokinesis by nucleating actin polymerization at the cell equator and promoting assembly and contraction of the contractile ring. However, due to technical limitations, a definite count of cytokinesis nodes has been missing. In this manuscript, Sayyad and Pollard employed Airyscan confocal microscopy, which allows live fluorescent observation with higher spatial resolution than spinning disk confocal microscopy, and detected cytokinesis nodes in a smaller size than previously observed using Blt1-mEGFP as a marker. They coined these dim spots they detected "unitary nodes" and claimed that they are the basic units of the cytokinesis nodes and the previously reported nodes are their assemblies.

There are problems in the analysis of the data from the Airyscan confocal microscopy. The definition of the new terminology "unitary nodes" is unclear. As seen in Figure 2A, dim fluorescence spots are detected throughout the cell. Although they are in general dimmer than the "unitary nodes" marked by dotted circles, some of them are comparable. This is not simply due to the artificial impression of overlapping signals by the maximum z-projection, as confirmed by 3D perspective view of the original data NodeBroadband.czi (overview.mov generated by annotate_nodes.ijm script, refer to the blue arrowheads in summary PDF, in *eLife*'s dropbox: https://www.dropbox.com/sh/gbay207esrvzs78/AADramgOx_7Q0oUck8kmoiexa?dl=0).

The difference between the dim "unitary nodes" and the cytoplasmic dimmer spots is not obvious. Indeed, 2 out of 10 ROIs on the sample image provided by the authors reside slightly inside the cell, clearly not on the cortex or cell membrane as the others do. These non-cortical spots are inconsistent with the widely accepted picture of the cytokinesis nodes. Spots with similar intensities can be found throughout the cell, including the polar regions. In summary, according to their intensities, there are at least three classes of Blt1 spots (1) dimmest spots found everywhere, (2) the dim spots on the equatorial cortex ("unitary nodes"), and (3) brighter equatorial spots observable by spinning disk confocal microscopy (conventional nodes). However, distinguishing between the class 1 and class 2 spots doesn't seem to be trivial and has not been sufficiently detailed in the current manuscript.

It is also unclear what the authors' interpretation of the class 1 spots is. They look different from typical background signal, which is usually derived of rapidly diffusing GFP-tagged molecules, cleaved free GFP molecules, or other endogenous molecules (auto-fluorescence), and appears noisy but largely uniform with some temporal and spatial fluctuations. The class 1 spots could be artifacts of Airyscan microscopy or real Blt1-GFP signals that represent the monomer or assemblies in a smaller size than the "unitary nodes" (class 2 spots). If the former was the case, this would raise a question about the nature of the class 2 spots, the dimmest of which are comparable to the brightest of the class 1 spots. If the latter was the case, the authors would need to explain how they remain sufficiently immobile during image acquisition and reconsider the appropriateness of their method of background subtraction. This would also question the suitability of the word "unitary". In either case, clearer and more objective criteria for distinguishing between class 1 and class 2 spots, which can ideally be executed by automation, should be provided.

The term "unitary nodes" contains strong implications that this is the basic, indivisible unit of the nodes and that the brighter nodes are the assembly of them. In Figure 2D, a histogram of their intensity distribution and the Gaussian fitting to it were displayed as supporting evidence for their monodispersity. However, a statistical test for normality is missing. A proper test such as the Kolmogorov-Smirnov (K-S) test and the Shapiro-Wilk test is necessary.

One of the possible criteria for the "unitary nodes" would be the localization at the cell cortex as the authors implicitly assume. An appropriate marker for the plasma membrane or the cell wall would provide a solid reference for this.

To clarify the points raised above, 3D distribution of Blt1 was reconstituted from the provided data, NodeBroadband.czi, by a custom ImageJ/Fiji script (annotate_nodes.ijm script) and exported as 'overview.mov'. The signals that are comparable to the 'unitary nodes' and slightly brighter than the dimmest cytoplasmic spots can be found in the polar regions as well (blue arrowheads in summary.PDF). 10 ROIs in Node_RoiSet.zip were marked by color. The 3D reconstruction of the equatorial broad band region (nodes.mov) and its maximum y-projection (projection along the long cell axis, nodes_max_prj.tif) indicate that 2 of them (magenta) are not on the cell surface. Please find these files at https://www.dropbox.com/sh/gbay207esrvzs78/AADramgOx_7Q0oUck8kmoiexa?dl=0.

The FFT of an image depends both on the microscope and the sample. To compare the abilities of different microscopy techniques, ideally, the same sample should be observed. The cells in Figure 1A (Airyscan) and 1C (spinning disk) don't look like the same cell. What does the diagonal line lowering towards the right in Figure 1B represent? A (dim) structure that exists only in the Figure 1A 'CR' not in Figure 1C 'CR'? Or, an artifact by Airyscan microscopy?

What are the magenta signals in Figure 1C?

*Reviewer #2 (Recommendations for the authors):*

This paper studies cytokinesis nodes, which are precursors of the cytokinetic ring in fission yeast *S. pombe* cells. The project addresses a timely question in the field, and the authors' data support their conclusion that the number of nodes scales with cell size.

Overall, the study fits with past work in the field and strengthens the model that node number scales with cell size. The conceptual advance beyond these past studies appears limited, and in several places the conclusions are not sufficiently supported by the data. I also have some concerns regarding the new method for estimating node number as discussed below.

Strengths:

1. In the Introduction and the Discussion section, the authors do an excellent job of comprehensively discussing previous studies that have investigated how nodes scale with cell size.

2. Several unexpected results reveal novel layers of node regulation that can be studied in the future. For example, nodes were more dispersed in small wee1-50 cells and size scaling of nodes was altered in wide rga4∆ cells. Since these two mutants are shorter that other cells examined, it suggests that some unidentified mechanism related to cell length is altering the known node regulatory system.

Weaknesses:

1. The paper mostly confirms previous studies that also showed node number scaling with cell size. The authors have done a nice job of using small and large mutants to increase the range of cell sizes beyond previous work, but the conclusion remains very similar. I would also note that some aspects of node positioning and density appear to break down at the extreme sizes – e.g. node distribution changing in small wee1 and rga4 mutants, and node density failing to scale with cell size in large cdc25-22 mutants. These exceptions seem potentially very interesting but are not explored.

2. Figure 2: I am concerned that the authors' new method for counting the number of nodes may not be accurate, which has implications for many conclusions in the paper. In short, the authors identify the dimmest population of Blt1-mEGFP nodes and define this population as "unitary" nodes. They then measure total Blt1-mEGFP fluorescence in mid-cell and assume that this entire fluorescence is comprised of unitary nodes. They suggest that brighter nodes are multimers of unitary nodes, but this assumption may not be correct based on many past studies in the field that have shown a range of node sizes. The authors even state "the fluorescence intensities of these spots ranged from very bright to dim" (line 142). Since all quantification in the paper relies on this assumption, it becomes a major caveat. In addition, Blt1 appears to be a peripheral membrane protein such that it might not all be contained in nodes. Can the authors be sure that the mid-cell region lacks diffuse Blt1 proteins that are diffusing around between nodes? Their signal would be attributed to unitary nodes based on the methods described. I don't mean to suggest that the authors' method is definitively "wrong," but rather it carries some heavy assumptions and caveats that leave me uncertain about the resulting data.

3. Figure 3: The authors show that the number of nodes scales with cell volume and with cell length across a range of sizes (except for rga4 mutant). Given that size involves many parameters including volume, length, surface area, etc, I am confused that the major conclusion and title of the paper specifies that nodes scale with "volume." Recent studies from the groups of Fred Chang and Martin Howard have shown a role for cell surface area as opposed to volume or length in node pathways. The authors may have missed an opportunity to extend their results by comparing and contrasting how nodes scale with different aspects of cell size/geometry. As presented, this major conclusion of the paper does not extend beyond previous studies and perhaps overstates the importance of cell volume, limiting the overall impact of the work.

4. Figure 3K: The authors conclude that node density in the cell middle does not scale with cell size. However, the graph appears to be driven by the elongated cdc25-22 mutant measurements. To my eye, it looks like node density scales with cell length in all other strains but then does not scale specifically in the cdc25-22 mutant. It is important because the authors' conclusion contrasts with previous studies in the field (e.g. Pan et al., 2014).

5. Figures 4-5: These figures examine how node number scales with Pom1 concentration using Blt1-mEGFP as the marker for nodes. However, the role of Pom1 on nodes appears to be through Cdr2, which is not imaged in these experiments. In addition, the earlier figures in the paper focused on mitotic cells whereas these figures and the role of Pom1 are on interphase cells. This combination makes it hard to connect results on Pom1 with the earlier results in the paper.

6. Figure 6: The authors conclude that when Cdr2 concentration is increased, it results in smaller cells and more nodes. However, the actual experiment shows that restoring Cdr2 expression into a cdr2 deletion mutant restores cell size and node regulation. I do not see data indicating that Cdr2 overexpression reduces cell size below wild type levels. Instead, the data support the expected conclusions for restoring the function of a deleted gene.

7. The authors conclude that Pom1 and Cdr2 do not act directly on cytokinesis node number but instead act indirectly by influencing cell size. This conclusion is confusing because nodes act to regulate cell size at mitosis prior to their role in cytokinesis. The authors discuss some of the pathways connecting these proteins to cell size in the Discussion, but it becomes confusing as interphase nodes and cytokinesis nodes are not fully distinguished in the discussion. If I understand their logic correctly, their main point is that interphase nodes controlled by Cdr2 and Pom1 do not influence the number of cytokinesis nodes later in the cell cycle. If this is correct, then I might suggest clarifying the text.

I found the logic of the Discussion section hard to follow at times. The authors make a number of interesting points but might consider consolidating some of their conclusions.

*Reviewer #3 (Recommendations for the authors):*

Using quantitative Airyscan microscopy, Sayyad and Pollard present detailed measurements of the number of cytokinetic nodes, the scaling of their number with cell size, as well as concentration of Pom1 and Cdr2. This work is a great contribution to quantitative studies of cytokinesis and more broadly to regulation of cell size. It's an invaluable resource for the development of mathematical models of cytokinesis and cell size regulation.

I am very enthusiastic about this work but I have a few comments to consider.

1) Node counting method. The counting of node numbers is based on calibrating the intensity of unitary nodes. This results in a number of nodes for WT cells that is considerably larger (~ 200) than earlier estimates, even by Laplante et al. who used single molecule super-resolution microscopy to measure ~140 cytokinetic nodes in cdc25-22 cells.

While Laplante et al. concluded that both WT and cdc25-22 arrested and released cells have the same number of nodes, in this paper (Figure 3H) as well as in Vavylonis et al. (2008), cdc25-22 cells were found to have more cytokinetic nodes as compared to WT. If the estimate of WT nodes in Laplante et al. needs to be revised downwards, to resolve this apparent disagreement, such a change would make the difference in node number counting between this paper and earlier studies even larger.

Evidence for unitary nodes comes the observed quantization of node intensities by Akamatsu et al. (2017) who fitted histograms of node intensities by multiple Gaussians. However, histograms with small number of data per bin show fluctuations that may appear as peaks, which change depending on the bin width. I am sure that Akamatsu et al. were very careful in their analysis, but I couldn't find how this issue of binning fluctuations was dealt with in their paper. The current paper does not attempt to reproduce the quantization seen by Akamatsu et al. (which could also have provided the intensity of unitary nodes).

Given the above comments, I am concerned that the intensity distribution of what are considered unitary nodes in Figure 3G might reflect the tail of a broader intensity distribution (representing a larger variation of node intensities whether unitary or not), which is cut off by a selection procedure. This might result in overcounting of nodes.

2) Is the microscope's PSF varying significantly along distance from slide, and if so, should this be accounted for when measuring the total intensity of the broad band through sum projection?

3) Scaling of nodes with volume versus area. The scaling of number of nodes with volume is emphasized in this work. However Pan et al. (2014) had reported that Cdr2 and cell size at division scales with surface area, see Figure 6 in that paper. I was confused on whether the measurements between these two papers are in disagreement or else if they reflect different phenomena. Could an area scaling be related to how cells such as rga4Δ that have different diameter deviate the most from the trend line in Figure 3H? (Incidentally, it's hard to observe different diameters between WT, rga4Δ and rga2Δ in the images of Figure 3B,D,E). Some explicit discussion would be helpful.

4) Use of unbiased statistical/data clustering methods might provide a more reliable unitary node calibration method. An alternative approach could be to be more conservative in terms of absolute node numbers and instead present the results in relative terms.

5) Figure 1 shows a comparison between Airyscan and confocal images to indicate the improved resolution of Airyscan. However, both of these images appear to have less signal to noise and resolution as compared to prior published confocal images of nodes with similar fluorophores. So the improvement in resolution is not superficially obvious.

6) The authors make the general assumption that unitary cytokinetic nodes are similar in terms of composition and stoichiometry. Can the authors discuss the implications of this assumption through the paper? For example, do all Blt1 nodes become cytokinetic nodes and how is stoichiometry preserved when Cdr2 concentration is varied (is the Blt1 to Cdr2 ratio in nodes constant?).

[Editors’ note: further revisions were suggested prior to acceptance, as described below.]

Thank you for submitting the paper entitled "The number of cytokinesis nodes in mitotic fission yeast scales with cell volume" for consideration at *eLife*. Your article and your letter of appeal have been considered by a Senior Editor, BRE member (Mohan Balasubramanian), and three Reviewers (including Dimitris Vavylonis). We regret to inform you that we are upholding our original decision. The Reviewers have put in an enormous amount of time in evaluating your rebuttal letter as well as looking closely at the primary data.

The Reviewers and editors are enthusiastic about the new approach of using Airyscan to investigate node architecture, properties, and scaling. However, the Reviewers have raised many similar questions, which require new experiments and / or analysis of existing data.

Should you be able to carry out the necessary experiments / analyses, *eLife* will be able consider the manuscript as a new submission (sent to the same three Reviewers).

Please find below the comments made by the Reviewers to your rebuttal letter and the revised manuscript.

*Reviewer 1:*

I have read the authors' rebuttal and remain concerned about limited impact and significance of the work. I was unable to access the revised manuscript, which is referenced in the rebuttal letter, but the changes and responses seemed clear from the point-by-point response. There are 3 main concerns that are unchanged from the initial submission (and I would also note that the authors did not really address several of my other comments).

1. The authors have measured total blt1-mEGFP signal in the cell middle, and then assumed that all of this signal is comprised of single, unitary nodes. Based on this assumption, they estimate the number of nodes in the cell middle. This approach relies heavily on two major assumptions: (a) that all blt1-mEGFP signal is contained in nodes, and (b) that all nodes are the same size. The important concern is that the authors have not in fact counted the number of nodes, but rather they have estimated their numbers based on a method that relies heavily on assumptions. A major claim in the paper is improved node counting over past publications, but there appear to be just as many caveats in the methods in the current paper as in past publications.

2. Regardless of differences in the methods of counting, the current paper reaches the same conclusion as past publications: node number scales with cell size. Node number scaling with cell size is an interesting correlation that fits with past publications, but the functional significance of this scaling for cytokinesis is not investigated limiting the impact of the work. Do large cells need more nodes to form a functional cytokinetic ring? What are the differences in assembly of a cytokinetic ring formed by fewer nodes (i.e. small cell) versus a ring formed by more nodes (i.e. large cell)? Without functional investigation into this scaling, the overall impact of the work is limited.

3. Several pieces of data show that node number and node density do not simply scale with cell size. rga4 mutants do not scale node number with cell volume compared to wild type, and cdc25-22 do not scale node density with cell volume. The fact that these mutants break the scaling properties means that nodes do not simply scale with cell size. To extend beyond the previously published conclusion that node number scales with cell size, the authors could have investigated why these mutants lose scaling and what is means for cytokinetic ring assembly. Without such investigation, the impact of their work is limited to supporting past conclusions on these structures.*Reviewer 2:*

I read the paper and modified my review as follows. I have included a dropbox link to an image (that could be provided through *eLife* dropbox link, thought it doesn't matter since I have provided my name).

The authors have addressed many of my questions and the paper is now much clearer in terms of the assumptions underlying this work. I am still however similarly concerned regarding the node counting method.

The identification of dim nodes as unitary ones, as the authors describe in their response, relies heavily on the findings of Akamatsu et al. who reported quantization of node intensities.

For example, it is stated on line 180: "We interpret the dim nodes as unitary nodes, while the brighter nodes consist of two or more unitary nodes" and on line 707: "We searched for dim spots and marked their 708 X-Y positions. Then we inspected the fluorescence along the Z-axis at these X-Y positions in 40 Z- slices. To qualify as a unitary node, a spot had to satisfy two conditions: (1) the fluorescence of a spot was present within a circle 12 pixels (~0.5 µm) in diameter; and (2) the fluorescence of a spot was present in 7 or fewer consecutive Z-slices with a peak in the middle slice and diminishing towards both sides"

From these statements I understand that the authors did not include bright nodes in the intensity distributions (such as in Figure 2D), even if they satisfied conditions (1) and (2) because these bright nodes were considered to be multiples of unitary nodes. I assume this is why the intensity distribution in Figure 2D does not have a long tail at high intensities, unlike the intensity distributions of Akamatsu et al. or Figure S6 of Laplante et al.

Now superficially, the number of data points in the histograms of Akamatsu et al. seem to me to leave some room for ambiguity with respect to interpreting the high intensity tail as integer multiples of unitary node intensity. To better illustrate this simple point, that may have been unclear in my first review, I attach graphs showing how 150 points (similar to the 138 node counts in Akamatsu) picked from a singly-peaked distribution may appear as multiple peaks as a result of fluctuations:

https://www.dropbox.com/s/suti2gz9ovgmfyw/histogram.pdf?dl=0

One could also argue that the fact that the width of intensity distribution in Figure 2D is comparable to its average indicates that the node size is not very tightly regulated, leaving room for other interpretations (for example node formation though nucleation and growth and a continuous size distribution with a long tail above a minimal nucleus).

Perhaps the authors could verify or critically reexamine the unitarity of nodes, which is so central in this work, in the light of the improved microscopy method. That may include quantifying brighter nodes (along the lines of Akamatsu) as well as the low intensity signal (such as measurements of signal to noise).

*Reviewer 3:*

I regret that I used the same pair of colors (green and magenta) in my summary.pdf for Blt1 and Sad1 signals, respectively, in the NodeBroadband.czi panel (top left) and for ROIs on the cortex and inside the cytoplasm, respectively, in the other panels. However, as clearly indicated in the PDF (The caption "Blt1 channel" + color coding of the ROIs in the ROI manager, i.e., ROIs "0032-0271-0078" and "0033-0285-0077" in magenta while others in green ) and in the ImageJ/Fiji script, to analyze the node positions, only the Blt1 channel was used. As in the maximum z-projection of the original stack overlaid with the ROIs defined by the authors ("rois_on_max_prj+sad1.png", to be shared via the Editorial office), all the ROIs defined by the authors are clearly distinguished from the two Sad1 peaks, which mark the spindle pole bodies. To make this point even clearer, the Sad1-RFP signals were overlaid in red on a 3D reconstruction as well ("nodes+sad1.mp4" movie, to be shared via the Editorial office).

In the rebuttal, as to the unitarity of the cytokinesis nodes, the authors refer to the FPALM analysis in Laplante et al. (2016). In principle, this technique can detect only a structure that is sufficiently immobile during the data acquisition. In Materials and methods of this paper, they wrote "Clusters of localized emitters distributed around the equator of the cells were manually selected from the reconstructed FPALM images into two different categories, face views and side views. Broad bands of nodes formed during prophase, and nodes remained stationary until node movements began during metaphase. For broadband nodes (prophase/metaphase cells), nodes were cropped from images reconstructed from 5,000 frames (25 s)." Neither clear criterion for the 'clusters' nor the raw data of FPALM was provided. Being immobilized within the small region (~100 nm) for a few seconds might be too selective. We can't exclude a possibility that smaller clusters resize on the cortex, though diffusing more rapidly than the level that can be recognized as 'clusters'.

In contrast, Airyscan can potentially capture smaller and moving clusters. Although it takes ~11 s or longer to scan a full volume of 22 µm x 22 µm x 6.8 µm (40 z slices), voxels close to each other can be completed in a shorter time (eg. < 1 s). Blt1 spots that are comparable to or slightly dimmer than those picked up by the authors as unitary nodes are found all over the cortex including the polar regions that were used as the background by the authors. Some of them were marked with blue arrows in my summary.pdf and their nature was questioned in my public review. However, this was not addressed in the rebuttal. No clear logic to treat them as just background was provided (although it might be described in the correct, revised manuscript).

As to the statistical test of Figure 2D

As stated in the paper, we used the D'Agostino-Pearson test which is more versatile and powerful than the Kolmogorov-Smirnov (K-S) test or the Shapiro-Wilk test for normality. Please see line #225

I can't make a definite comment on this as the revised manuscript is not available. However, if the authors are referring to the same sentence as the lines 209 to 211 in the original manuscript, this is not addressing my point about Figure 2D. This sentence is about Figure 2E, in which a test was done for the normality of the number of total nodes calculated as the ratio of the total fluorescence to the fluorescence per unitary node.

[Editors’ note: further revisions were suggested prior to acceptance, as described below.]

Thank you for submitting your article "The number of cytokinesis nodes in mitotic fission yeast scales with cell size" for consideration by *eLife*. Your article has been reviewed by 3 peer reviewers (with no conflict of interest in any way), and the evaluation has been overseen by a Reviewing Editor (not one of the three Reviewers) and Anna Akhmanova as the Senior Editor. The following individuals involved in review of your submission have agreed to reveal their identity: Dimitrios Vavylonis (Reviewer #1).

The reviewers have discussed their reviews with one another, and the Reviewing Editor has drafted this to help you prepare a revised submission. Please note that we have taken the very unusual step of inviting a further revision, and please note that this round of revision will be final. We hope that all points will be addressed satisfactorily since the Reviewers have spent an enormous amount of time on the review on multiple occasions. If you prefer not address some of the points of the reviewers in full, please seek publication elsewhere.

Please note that 2 of the 3 the Reviewers have stated that your revised manuscript did not address the key points. I have looked at the comments and agree with the Reviewers that there are two major issues (along with some minor issues) that remain ambiguous and therefore affect the strength of your conclusions and the extent of advance. Although Reviewer 1 was satisfied, in further discussion they mentioned the following agreeing with the unresolved issue of background correction.

Reviewer 1 comment during discussion:

I was myself mostly satisfied because the new analysis of brighter nodes does not show a long tail. So I thought that this means there is a characteristic average Blt1 node size (which may be called a "unitary node") even if the distribution of specks at low intensities is not measured accurately.

However, there are caveats, as explained by your reviews and I don't know how to explain it any better. I further realize that my short statement regarding evidence for unitary nodes was too strong and probably incorrect. So I would just like to say in my review:

"The revised analysis in Figure 2D, which now includes brighter nodes shows that these nodes have intensities within a certain range. The average intensity of this distribution can be called a "unitary node" so this addresses the main concern I had."

The issue of background correction and its dependence on cell size (and a possible solution by Reviewer 3) seem important to address.

Essential revisions:

1. Please note that addressing the point # 2 raised by Reviewer 3 is essential for the strength of the conclusions and should be performed.

2. Please also address points # 2-7 by Reviewer 2. Although they seem numerous, I have gone through the list. Most of these pertain to giving more information, reanalysis of data and the use of the most appropriate statistical treatments.

*Reviewer # 1:*

The revised analysis in Figure 2D, which now includes nodes of all intensities, provides additional support, and is consistent with prior evidence, for unitary nodes, as explained in the response letter. This addresses the main concern I had.

I have a comment regarding the "dim specks" (that could possibly be cytoplasmic oligomers) mentioned in Response to Reviewer 3. The authors mention that these specks were not confused with the brighter unitary nodes. It was not clear to me if this is true because the dim specks did not satisfy the two node criteria (fluorescence within 12 pixels and 7 or fewer consecutive Z-slices with a peak in the middle slice), or else if their intensity was smaller than the first bin in Figure 2D, or both.

*Reviewer #2:*

In the previous rounds, we pointed out various problems in data analysis and the limited impact of the work. Unfortunately, these were not substantially improved in this version. As detailed below, it is unclear whether the scaling of the node numbers with cell size is real. Insights into the mechanism of scaling or into the functional role of this phenomenon on cytokinesis are still missing. Please refer to slides in my 'problems.pdf', shared here: https://drive.google.com/file/d/1CdsqS3PgZNmfpZvgD_p4ZL6Z5UL1VZJf/view?usp=sharing.

The authors' response to the Reviewer 1 (second round, I was Reviewer 3) prompted me to look into the background and the method of measuring the fluorescence intensity of the 'unitary nodes' more carefully. The fluorescence intensity of a small node was measured by the intensity in a circular ROI of 12-pixel diameter on the sum stack of 7 slices (3 below and above the target node). The background was corrected by subtracting the mean of the equivalent ROIs arbitrary set near the cell tip on the same sum stack. The fact that both the cell diameter and the cytoplasmic background are not uniform along the cell axis make the background correction highly unreliable.

1.Geometry of the cell

According to the trace of the cell periphery in "background_RoiSet.zip", the diameter of the cell near the cell tips, where the background was measured, is smaller (~2.7 µm) than that at the equator (~3.7 µm). A circular ROI of 12-pixel diameter in the sum stack of 7 slices corresponds to a cylinder of ~0.5 µm diameter and ~1.2 µm height. For a node at the top of the cell, the corresponding background volumes near the tips are largely outside of the cytoplasm (page1 of 'problems.pdf Geometrical problem 1). This underestimates the background intensity. On the other hand, for a node on the midplane of the cell, the corresponding background volume is entirely inside the cytoplasm (page2 Geometrical problem 2). This overestimates the background intensity. These effects are not negligible because the raw background intensity varies between 1000 to 2000 depending on the depth of the z-slices while the peak raw intensity of the nodes is 3000~4000 (page 5 of 'problems.pdf').

2.Variability of the cytoplasmic signal and measurement of the node intensity and the node number.

Apart from the granularity, the cytoplasmic Blt1-GFP signals in the cell in "NodeBroadband.czi" marked with "background_RoiSet.zip" and "Node_RoiSet.zip" show significant variation along the cell axis. This can be demonstrated both by reslicing the cell volume and measuring a circular ROI at the center of the cell (page 3 of 'problems.pdf') or by line-profiling in the sum projection of the central 7 slices (z=23~29, page 4 of 'problems.pdf'). In addition to the large fluctuation due to the cytoplasmic spots, there is a clear trend that the cytoplasmic signal increases towards the cell equator. To make the situation more complicated, the intensity drops in the region corresponding to the nucleus around the equator.

The intensity of the cylindrical ROI put on the empty space outside of the cell (z=1~7) is 770 +/- 30 (mean +/- SD) (page 6 of 'problems.pdf'). In the central 7 slices (z=23~29), the intensity of the background ROIs set by the authors (green) is 2010 +/- 360. This increases to 2740 +/- 480 closer to the equatorial (magenta, still outside of the broad equatorial band) and 3050 +/- 610 at the periphery of the nucleus (yellow). These are not negligible in comparison with the raw intensity of the nodes 4170 +/- 690 (min. 3280 ~ max. 5510).

The authors reasoned usage of the ROIs near the cell tip as the regions without brighter spots. However, this is not an ideal choice because the intensity for background correction should be taken from the region with a similar level of the background. The authors' method will systematically underestimate the background signal. If the background was measured near the equator avoiding the cortical signals such as the ROIs in the perinuclear region (yellow on page 6 of 'problems.pdf'), we would get the WT node intensity, in the worst case, ~50% smaller than those reported in the current manuscript. Using the background ROIs near the cell tips also introduces unnecessary uncertainty that might vary depending on the cells and on the positioning of the ROIs.

This problem becomes more serious when we compare the strains with different cell sizes. Consistent with the above observation, Blt1-mCherry in Figure 4B outside the equatorial band shows a rather constant decrease towards the poles. The gradient seems to be common across different strains (page 7 of 'problems.pdf'). This suggests that the bigger the cell size, the lower the background signal measured near the poles. This would result in a larger over-estimation of the total intensity of Blt1-GFP in the broad equatorial band in larger cells. Dividing this with the intensity per node, which is largely constant across the strains (Figure 3G), would result in the size-dependent over-estimation of the node number. In other words, the scaling of the node number might just be an artifact of the cell size-dependent under-estimation of the background.

An additional twist is the drop of the cytoplasmic signal at the equator due to exclusion by the nucleus. Although this effect might be less significant than the cell size-dependent drop of the background measured near the cell tips, its influence on different measurements should be checked.

2.1 What is the variety of the background in the same strain and across the strains? Provide the raw data.

2.2 How stable the current results are against the selection of the background ROIs closer to the equator in the wild-type or at the same distance from the equator for inter-strain comparison?

2.3 Check the influence of the drop of Blt1-GFP signal at the nucleus on various measurements

3.Dimmer spots on the cell surface

Criteria for the 'unitary node' by the authors are "To qualify as a unitary node, a spot had to satisfy two conditions: (1) the fluorescence of a spot was present within a circle 12 pixels (~0.5 μm) in diameter; and (2) the fluorescence of a spot was present in 7 or fewer consecutive Z-slices with a peak in the middle slice and diminishing towards both sides." However, these are not sufficiently specific for unanimously identifying the 'unitary nodes'. Near the spots hand-picked as 'unitary nodes' by the authors (or at an equivalent cortical position), there are spots that are significantly dimmer but satisfy these criteria (pages 8 to 13 of 'problems.pdf', 'Dimmer cortical spots, example' 1~6). The majority of them are brighter than the background signals (at least those in the volumes defined by the authors near the cell tips) (pages 14 and 15 of 'problems.pdf').

3.1 Why were they excluded from the node measurement?

3.2 Analysis in Figure 2 should be done, including these dimmer spots on the cell surface.

3.3 Whether the node counting based on the intensity of 'unitary node' is still valid needs to be reconsidered.

3.4 If the authors insist on excluding the dimmer spots, more strict (but reasonable and not arbitrary) criteria for the 'unitary nodes' should be provided (although, to be honest, I doubt the concept of 'unitary node', see below).

4.Statistical test for normal distribution

The D'Agostino-Pearson normality test uses the skewness and kurtosis of the distribution (https://en.wikipedia.org/wiki/D%27Agostino%27s_K-squared_test). It is not suitable to examine whether the samples are from a normal distribution or a sum of multiple normal distributions. As shown on page 16 of 'problems.pdf', with the simulated samples from combined two normal distributions (not normal as a whole), the D'Agostino-Pearson test fails to reject the null hypothesis of normality while the Shapiro-Wilk test properly detected the deviation from the normality and rejected the null hypothesis (R script in page 17 of 'problems.pdf').

The Shapiro-Wilk test and the Kolmogorov-Smirnov test were explicitly suggested in my first review (based on the above simulation, the Shapiro-Wilk test seems to be more appropriate for our case). I suspect that the authors might have obtained p-values < 0.05 with these tests, indicating that the distribution is not likely normal. This would provide another line of reason to doubt the existence of the 'unitary node'.

4.1 Provide the raw data behind Figure 2D and 2E including both the raw node intensity and the background measurements so that other people can examine them

5.Distributions of fluorescence intensities of small nodes in mutant strains

Although "The distributions of intensities Blt1 fluorescence of cytokinesis nodes were similar (around 2500 AU) in wild-type cells (Figure. 2C) and three of the experimental strains (Figure. 3G)." (line 292-4), none of the histograms in Figure 3G look like a normal distribution.

5.1 Why aren't they normally distributed?

5.2 What would they look like if the dimmer spots on the cell surface were included?

5.3 Provide the raw data behind Figure 3G-K so that other people can examine them

6.Unitarity of the 'unitary nodes'

The spots on the cell surface dimmer than the 'unitary nodes' hand-picked by the authors were indeed observed also in the FPALM image in Figure S2 of Laplante (2017). There were smaller clusters of emissions marked by arrows (pages 18 of 'problems.pdf'), which were excluded from the analysis without any reasoning.

Moreover, even within the data of the clusters that the authors believed to be 'unitary nodes', there is a clear sign of heterogeneity. The distributions of the numbers of emissions per cluster show a long tail, which contains ~30% or more clusters. This indicates that, in addition to the dimmer clusters that were not counted as 'nodes', there exist significant heterogeneities even among the hand-picked 'nodes'. It is not clear what "Nodes in the tail were seen only in long reconstructions (25 s), not in short reconstructions (5 s)." (page E5880 Laplante (2017)) means without the real data for the short reconstructions.

The concept of 'unitary node' might be supported if the brighter spots were able to be explained as integer multiples of unit intensity. In Akamatsu (2017), the authors argued that the distributions of fluoresce intensity of the interface nodes can be better fitted by multiple Gaussian distributions than by continuous log-normal distributions. However, the fitting was done by setting the amplitudes, SDs, and means for each of the Gaussian distributions as free parameters (9 or 12 parameters for 3 or 4 Gaussians, respectively). This is simply wrong since there must be restrictions both on the means and SDs. The mean and SD of the n'th peak (m_n and s_n, respectively) should follow m_n = m_1 x n and sd_n = sd_1 x square root of n. A simple comparison of the goodness of fit with the other distribution that has only a limited degree of freedom (the log-normal distribution has only 2 parameters) is non-sense. The comparison between the statistical models should have been done based on an appropriate estimator such as the Akaike information criterion (AIC) (https://en.wikipedia.org/wiki/Akaike_information_criterion). With a simulation https://www.dropbox.com/s/suti2gz9ovgmfyw/histogram.pdf?dl=0, Reviewer 2 raised a valid example of artificial appearance of multimodality from a log-normal distribution.

Considering all the points discussed so far, i.e.,

-Unexplained exclusion of smaller clusters in the analysis of FPALM data (Laplante (2017))

-The long tail towards the bigger clusters even among the selected '(unitary) nodes' (Laplante (2017))

-Inappropriate statistical analysis of the distribution of the intensity of the nodes with a long tail (Akamatsu (2017))

-The intensity of the spots picked by the authors as 'unitary nodes' from the Airyscan images doesn't seem to follow a normal distribution (Figure 2D and Figure 3G),

evidence for the 'unitary node' is pretty weak. We can't exclude a possibility that there is no such thing as 'unitary node' but the nodes are rather amorphous aggregates of proteins as illustrated on page 20 of 'problems.pdf'. A critical re-examination of the author's own past data must be done.

7.Nature of the dotty cytoplasmic signal

The authors responded, "The processing of Airyscan images increases the contrast of these dim inhomogeneities". It remains as a black box what image processing was done as the only information provided is "To increase signal-to-noise ratio and resolution, acquired images were processed by 3D Airyscan filter strength 7.0 with Zen Black software."

7.1 Clarify the principle of the filter with an appropriate reference and specify the parameter settings (Is 'strength 7.0' sufficient? What does this mean?)

7.2 Does this processing preserve the linearity of the fluorescence signal?

7.3 Why are the grayscale levels so small? Only a range 0 to <~30 in 0-65535 grayscale levels was used.

*Reviewer #3:*

The revised manuscript demonstrates a correlation between cell size and the number of cytokinetic node precursor structures. This conclusion supports and extends previous studies in the field. The strengths of the work are balanced by some limitations that have been expressed in earlier reviews and remain largely unresolved:

(1) The overall impact of the work is limited because it reaches the same conclusion as previous work. As the authors have pointed out, their current work moves beyond previous studies in two ways: (a) they examine these node structures during cytokinesis whereas previous studies examined them during interphase using different markers; and (b) authors have used cell size mutants to extend the range of sizes examined. However, the overall conclusion is the same as earlier work. It is also worth noting that several of the size mutants (wee1-50, rga4D, and cdc25-22) do not follow similar size-scaling properties, but these potentially interesting differences remain unexamined.

(2) The major conclusions of the paper rely on the number of cytokinetic nodes present in different cells, but the authors have not actually counted the number of nodes. Their methods for estimating the number of nodes rely on several tenuous assumptions and have raised concerns from all of the reviewers at each stage of review. It is not definitively known if Blt1 nodes are all the same size (referred to as 'unitary nodes' in the paper). It is also not known if all plasma membrane-bound Blt1 in the cell middle is contained in nodes, or alternatively if there is membrane-bound Blt1 diffusively localized in this region outside of nodes. Both 'unknowns' leave me concerned about the accuracy of the numbers presented. I would like to propose a potential solution to this problem. The authors discuss the improved ability of airyscan imaging to resolve unitary nodes within the broad band. They could use their high-resolution images to count Blt1 nodes from the same images, and then present these numbers as a 'proof-of-principle' that node number scales with cell size by 2 separate analyses (counting versus their current estimations). The counting method would underestimate due to areas with multiple, unresolved unitary nodes; but similar scaling should be observed. It would not be necessary to reanalyze all strains/images this way, but perhaps just wild type and some cdc25-22 cells to establish a similar scaling trend by both methods.

[Editors’ note: further revisions were suggested prior to acceptance, as described below.]

Thank you for resubmitting your work entitled "The number of cytokinesis nodes in mitotic fission yeast scales with cell size" for further consideration by *eLife*. Your revised article has been evaluated by Anna Akhmanova (Senior Editor), a Reviewing Editor (Mohan Balasubramanian), and the Reviewers. Thank you for sending the further revised manuscript, and we apologize for the delay in its review due to travel schedules of the editors and the Reviewers.

The manuscript has been improved but there are some remaining issues that need to be addressed, as outlined below. Please note that Reviewer 3 (an imaging expert with absolutely no conflict of interest) has invested an enormous amount of their time and raised many points mostly about data analysis and statistics that we would like you to answer. Since this paper is very strongly based on quantitative data analysis, addressing these concerns is essential to support the conclusions.

The authors expressed concerns about the reviews. Specifically, they "note that the critique from reviewer #2 has expanded for a third time to include not just questions about the paper under consideration but also published work from our lab and other groups". This remark does not accurately reflect the peer review history of this manuscript. The issues around the unitarity of the 'unitary nodes' have been a concern from the beginning. The possibility of the appearance of pseudo-multimodality from log-normally distributed data was suggested by Dr. Vavylonis upon reviewing the first rebuttal. In the second revision, the authors referred to Laplante et al., 2016 "we must emphasize that the strongest evidence for the unitary node concept comes from the FPALM super-resolution of Laplante et al. (Laplante et al., 2016) not confocal data.", instead of seriously reconsidering the reasonable possibility raised by Dr. Vavylonis. This prompted reviewer #3 to look into Laplante 2016 and Akamatsu 2017 more carefully and led to identification of an an issue common to the current work, i.e., dimmer cortical signals of 1/3~1/2 intensities (or a number of localized emissions) of the 'unitary nodes'. The authors' objection thus does not appear justified.

The authors' strategy to support the unitarity of the 'unitary nodes' is to distinguish the slightly dim spots from the 'unitary nodes' and combine them with the much dimmer cytoplasmic specks. However, as detailed below, their arguments are not sufficiently strong and appear subjective. Please note that a newly cited, recent paper, Bellingham-Johnstun et al. (2021), whose corresponding author is the same person as the first author of Laplante et al. (2016), carefully and reasonably avoided the term 'unitary node'.

Furthermore, there arises a new issue of data handling. The numbers of the various cell types analyzed for this version are different from those in the previous versions (WT: 24 reduced to 21, wee1:24 to18, rga2: 23 to ?, cdc25: 32 to 31, rga4: 18 to 17). There was no mention or explanation of this significant change while the new look of the distributions in Figure 3G was attributed to the change in the background correction.

Issue 1 Geometry of the cell

"The background fluorescence was 9% higher next to the broad band than at the tips of the cells, but subtracting this higher value reduced the average intensities per node by only ~5% and the total intensity of the node broad band by only 13%." (page 4 of the rebuttal)

This is quite different from what the reviewer measured and presented to the authors in the previous round. One would need to look at the raw data. Was the image data of a cell presented by the authors an exception?

Issue 2 Variability of the cytoplasmic signal and measurement of the node intensity and the node number

The reviewer pointed out the difficulties in handling the background signal (1) large fluctuation and (2) systematic decrease towards the cell tips.

The regions closer to the equatorial zone used in this version are better than the regions near tips, addressing the point 2 to some extent although, as was previously mentioned, the cytoplasmic signals are even higher in the perinuclear region next to the equatorial cortex.

To draw a conclusion that the fluctuation is small, the line scans were shown in Figure R2. The authors claimed that the signal along the cell edge (green) containing the nodes is much higher (51{plus minus}21 AU, mean{plus minus}sd) than the cytoplasm (14{plus minus}5 AU). This itself is correct. However, this doesn't support that the background fluctuation is small relative to the intensity of the 'unitary nodes' since the major peaks (60~90 AU) on the green profile in Figure R2B correspond to the clusters of the many 'unitary nodes' (please refer to page1 of problems2.pdf). The spot measured as a 'unitary node' (ROI 0025-0270-0044, arrow) corresponds to a minor peak of ~30 AU, which is comparable to the fluctuations of the blue profile. They brought the intensity of the clusters of 'unitary node' when discussing the intensity of individual 'unitary nodes' relative to the background fluctuation.

The authors tried to explain the difference between their claim and the reviewer's analysis "Our measurement of background fluorescence (blue curve, in Figure R2B) differed from the reviewer's analysis in two ways: (1) They did not subtract the background outside cells from the image, so the background intensity is higher in their analysis; and (2) their ROI extended outside the tips of the cell, where the fluorescence is lower." In addition to the above issue, both the remarks miss the point. As to (1), firstly, subtraction of the common background (outside of the cell) doesn't influence the comparison between the peak of the 'unitary node' and the fluctuation of the cytoplasmic background. Secondly, in the previous review, when discussing the mean levels of the cytoplasmic background, they were standardized with the background outside the cell set to 0 and the mean node intensity set to 1 and pointed out that the cytoplasmic background levels vary 33% to 64% of the mean intensity of the 'unitary nodes' depending on the location with large spreads. As to (2), the line ROI was simply set from one end to another end of the outline of the cell defined by the authors, which seems to be overlapping with the other cell on the righthand side.

Issue 3 Dimmer spots on the cell surface

A concern was raised about the criteria to spot the 'unitary nodes' by picking up just slightly dimmer spots observed next to the 'unitary nodes' on the equatorial cortex, which were not counted by the authors. The authors responded to this by measuring the dim specks that are found everywhere and are easily distinguishable from the 'unitary nodes' (~170 AU, ~15 times dimmer). However, the key point was how one can define criteria that distinguish those just slightly dimmer spots the reviewer picked up from the authors' 'unitary nodes'. This was because, if it was difficult, (1) this would question the concept of the 'unitary node' and (2) this would affect the average intensity of the 'unitary nodes' and hence the node counts. The authors' argument using much dimmer cytoplasmic specks is out of focus. The authors' argument based on the minimal contribution of the dim specks to the overall signal in the broad band doesn't address these problems.

They also made comments on the spots the reviewer picked up. 5 of 8 spots were rejected because "Spot x is small and the fluorescence is limited to 2-3 Z-slices" However, these are ad-hoc and highly subjective criteria as we still find in the Methods section "To qualify as a unitary node, a spot had to satisfy two conditions: (1) the fluorescence of a spot was present within a circle 12 pixels (~0.5 μm) in diameter; (2) the fluorescence of a spot was present in 7 or fewer consecutive Z-slices. with a peak in the middle slice and diminishing towards both sides.". Note the terminologies, "within" and "7 or fewer consecutive Z-slices". They also commented on Spot 1 "Spot 1 looked like two spots together in both the X-Y and X-Z-slices, so we did not count it as a node." However, this is also the case for the ROI 0016-0244-0082 on the same z-slices, ROI 0018-0306-0104 and 0019-0249-0116, which have irregular, non-circular appearances (pages 8-10 of problem.pdf shared previously).

The node is a structure with a diameter of less than 100 nm. The observed spots are the convolution of the geometry of fluorescent proteins and the point spread function of the microscope, followed by deconvolution. The major factor that determines the size of a spot is the intensity of the fluorescence. The exclusion based on the size of the spot is almost equivalent to the exclusion by the fluorescence intensity, which is proportional to the number of the tagged molecules.

Issue 4 Statistical test for normal distribution

It was admitted that the proper test rejected that the distribution of the intensity of the 'unitary nodes' follows a Gaussian distribution unless an additional selection is introduced. This is consistent with the uncertainty and subjectiveness of the criteria for the 'unitary nodes' discussed above. Anyway, the reason for the change in the number of cells needs to be clarified. Does the original dataset pass the normality test after the same manipulation (cutting the long tail)?

It is difficult to understand why the authors insist on the normal distribution. Akamatsu (2017) showed a distribution of Blt1 with a similar spread, which was fit with multiple Gaussian. Figure 2D can easily be fit with multiple Gaussians as well (page 2 of the Reviewer problems2.pdf)

In this version, there are two different Figure 2D histograms, one on page 9 and the other on page 55. They are different in bins centered on 3000 and 3500 (arrows in page 3 of problems2.pdf) in addition to those centered on 5000, 5500, 6000, and 6500. This difference must influence the Gaussian fitting with the nodes whose intensity was less than 4600 A.U. However, the two curves of Gaussian fitting seem to be identical (both annotated as "2300{plus minus}900"). Something is wrong here.

Issue 5 Distributions of fluorescence intensities of small nodes in mutant strains

The authors claim that the distributions in the current Figure 3G follow a normal distribution based on the Shapiro-Wilk test, attributing this change to the new analysis with the background ROIs adjacent to the node broad band. However, the number of the cells analyzed is quite different from the previous versions (24 cells to 18 cells in an extreme case). Why were some data silently omitted? Can we get the same results with the original set of cells?

Issue 6 Unitarity of the 'unitary nodes'

The authors accepted that, by FPALM, they had detected clusters smaller (or less number of localized emissions) than the 'unitary nodes' but excluded them from their analysis. They rationalize the omission of these smaller clusters by arguing (1) they are distributed across the entire surface and (2) some of them are not on the flat surface. However, these arguments are weak. As for (1), it is true that there are small clusters across the entire cell surface (Laplante 2016 Figure 1A, an enhanced image on page 4 of problems2.pdf). However, they are much smaller/dimmer than those the reviewer previously spotted in Figure S2E, and the similar ones in Figure 1A found in the equatorial zone with roughly about a half the density of those marked as the 'unitary nodes' (indicated by arrows on page 4 of problems2.pdf). Much smaller clusters everywhere are not a good reason to exclude the slightly smaller clusters (with about a half the number of localized emissions of the 'unitary node') found next to the 'unitary node' in the equatorial cell cortex. As for (2), the dimmer clusters spotted are within 500 nm from those scored as nodes containing one or two 'unitary nodes'. Is the curvature of the cell surface so steep?

Issue 7 Nature of the dotty cytoplasmic signal

Thank you for the detailed answers. Why don't you cite Korobchevskaya et al. (2016) in the Introduction?

---

## [Author Response]

[Editors’ note: The authors appealed the original decision. What follows is the authors’ response to the first round of review.]

Comments to the Authors:There was a great deal of enthusiasm from all three Reviewers and the editors for a compelling and quantitative elucidation of aspects of cytokinesis and in particular quantitation of cytokinetic nodes. However, three major concerns have been raised about 1. the concept of unitary nodes and the differences between the dimmer cytosolic spots is unclear and some pitfalls in the approach and interpretations were raised. Furthermore, as in the attached files that Reviewer 1 has prepared based on the raw data, there is concern about counting errors and how the cytosolic fluorescent spots are differentiated from membrane localized "unitary nodes". 2. The extent of broad interest in the work beyond the fission yeast community and 3. the somewhat unclear epistasis between cdr2, pom1 in cytokinetic node establishment and how these relate to mitotic nodes.In addition to the specific comments reproduced verbatim from the three Reviewers, the following issues were raised during the discussion between the Reviewers and RE.We all have now looked at Reviewer 1's files, as well as the raw data provided by the authors. We all agree that the unusual intensity fluctuations of the background should be discussed. From this analysis, it seems that roughly, 8/10 ROIs would correspond to what we can reliably call nodes (being close to the membrane and stationary structures with signal over several z slices). But 2/10 ROIs seem to be in the cytoplasm. This observation could imply some counting errors (not sure how much, maybe 20%?). Then there is the separate issue of possibly *not* counting bright nodes because they may have considered them not to be unitary (they say "We searched for dim spots" on line 680).

We thank the reviewers for their thoughtful evaluations of our paper. Three apparently serious concerns arose multiple times, so we decided to address those issues followed by responding to the remaining questions from each review line-by-line.

Fortunately, the underlying problem behind all three major issues was a lack of information about previous work. We assumed incorrectly that reviewers had accepted the conclusions of that prior work (explained in the introduction) as the starting point for the current project. Clearly, we needed to provide the reviewers more background, which we do here in the response to review and the revised introduction to the paper. For each issue, we provide a summary, followed the verbatim review comments in a box, and then respond.

*Issue 1. What are unitary nodes?* Reviewer 1 thought we coined the term “unitary nodes” and both reviewers 1 and 2 were unsure about the nature of a “unitary node.” In contrast, we started with a molecular model in mind (Laplante et al., 2016). Understandably, the reviewers’ uncertainty about this basic concept cast a negative cloud over the whole paper. Here are the reviewer's comments on this point.

Reviewer #1 (Public Review):Cytokinesis nodes are plasma membrane-associated assemblies of multiple proteins including myosin-II and play key roles in fission yeast cytokinesis by nucleating actin polymerization at the cell equator and promoting assembly and contraction of the contractile ring. However, due to technical limitations, a definite count of cytokinesis nodes has been missing. In this manuscript, Sayyad and Pollard employed Airyscan confocal microscopy, which allows live fluorescent observation with higher spatial resolution than spinning disk confocal microscopy, and detected cytokinesis nodes in a smaller size than previously observed using Blt1-mEGFP as a marker. They coined these dim spots they detected "unitary nodes" and claimed that they are the basic units of the cytokinesis nodes and the previously reported nodes are their assemblies.The term "unitary nodes" contains strong implications that this is the basic, indivisible unit of the nodes and that the brighter nodes are the assembly of them.Reviewer #2 (Public Review):2. Figure 2: I am concerned that the authors’ new method for counting the number of nodes may not be accurate, which has implications for many conclusions in the paper. In short, the authors identify the dimmest population of Blt1-mEGFP nodes and define this population as “unitary” nodes. They then measure total Blt1-mEGFP fluorescence in mid-cell and assume that this entire fluorescence is comprised of unitary nodes. They suggest that brighter nodes are multimers of unitary nodes, but this assumption may not be correct based on many past studies in the field that have shown a range of node sizes.Reviewer #3 (Public Review):6) The authors make the general assumption that unitary cytokinetic nodes are similar in terms of composition and stoichiometry. Can the authors discuss the implications of this assumption through the paper?

The concept of cytokinesis nodes has evolved over 25 years, all based on fluorescence microscopy, since nodes have not been purified or imaged by electron microscopy. The first sentence in the paper spelled out that concept: “Cytokinesis nodes are stoichiometric assemblies of multiple proteins, which associate with the plasma membrane around the middle of fission yeast cells and polymerize actin filaments that form the cytokinetic contractile ring.”

The super-resolution images (Figure 2 of Akamatsu et al., 2017) confirmed that interphase nodes are also discrete structures. By confocal microscopy, these interphase nodes range in size and intensity. Dim nodes are the most abundant but easily missed when surrounded by larger particles. In favorable cases (such as the abundant protein Cdr2 during interphase) histograms of node intensities have peaks with higher intensities corresponding to multiples of these dim spots, which Akamatsu et al. (Akamatsu et al., 2017) called “unitary nodes” corresponding to Laplante’s single nodes.

Clusters of cytokinesis nodes form by active translocation during contractile ring assembly; Myo2 in nodes pulls on actin filaments connected to neighboring nodes (Vavylonis et al., 2008). Clusters of interphase nodes appear to form by lateral diffusion on the plasma membrane (Akamatsu et al., 2017).

The thoroughly revised introduction explains this prior evidence for unitary nodes and cluster formation. We hope this paper will help to correct some misconceptions about the nature of nodes.

Issue 2. Did we confuse unitary nodes with Blt1 fluorescent specks in the cytoplasm? All three reviewers recognized the technical challenges of measuring the fluorescence intensities of nodes and the equatorial regions of the cell against a high background of cytoplasmic Blt1-mEGFP.

Reviewer #1 (Public Review)There are problems in the analysis of the data from the Airyscan confocal microscopy. The definition of the new terminology "unitary nodes" is unclear. As seen in Figure 2A, dim fluorescence spots are detected throughout the cell. Although they are in general dimmer than the "unitary nodes" marked by dotted circles, some of them are comparable. This is not simply due to the artificial impression of overlapping signals by the maximum z-projection, as confirmed by 3D perspective view of the original data NodeBroadband.czi (overview.mov generated by annotate_nodes.ijm script, refer to the blue arrowheads in summary PDF, in eLife's dropbox: https://www.dropbox.com/sh/gbay207esrvzs78/AADramgOx_7Q0oUck8kmoiexa?dl=0).The difference between the dim "unitary nodes" and the cytoplasmic dimmer spots is not obvious. Spots with similar intensities can be found throughout the cell, including the polar regions. In summary, according to their intensities, there are at least three classes of Blt1 spots (1) dimmest spots found everywhere, (2) the dim spots on the equatorial cortex ("unitary nodes"), and (3) brighter equatorial spots observable by spinning disk confocal microscopy (conventional nodes). However, distinguishing between the class 1 and class 2 spots doesn't seem to be trivial and has not been sufficiently detailed in the current manuscript.It is also unclear what the authors' interpretation of the class 1 spots is. They look different from typical background signal, which is usually derived of rapidly diffusing GFP-tagged molecules, cleaved free GFP molecules, or other endogenous molecules (auto-fluorescence), and appears noisy but largely uniform with some temporal and spatial fluctuations. The class 1 spots could be artifacts of Airyscan microscopy or real Blt1-GFP signals that represent the monomer or assemblies in a smaller size than the "unitary nodes" (class 2 spots). If the former was the case, this would raise a question about the nature of the class 2 spots, the dimmest of which are comparable to the brightest of the class 1 spots. If the latter was the case, the authors would need to explain how they remain sufficiently immobile during image acquisition and reconsider the appropriateness of their method of background subtraction.Reviewer #2 (Public Review)The authors even state "the fluorescence intensities of these spots ranged from very bright to dim" (line 142). Since all quantification in the paper relies on this assumption, it becomes a major caveat. In addition, Blt1 appears to be a peripheral membrane protein such that it might not all be contained in nodes. Can the authors be sure that the mid-cell region lacks diffuse Blt1 proteins that are diffusing around between nodes? Their signal would be attributed to unitary nodes based on the methods described. I don't mean to suggest that the authors' method is definitively "wrong," but rather it carries some heavy assumptions and caveats that leave me uncertain about the resulting data.

We thank the reviewers for calling attention to the atypical cytoplasmic background signal in Airyscan images of cells expressing Blt1-mEGFP. It pervades the cytoplasm and provides a challenge for quantitative measurements. The reviewers were concerned that these specks might be confused with the brighter unitary nodes. However, we feel that careful background subtraction successfully removed it from measurements of cytokinesis nodes

Live cells have an average of 8400 Blt1 molecules with 2100 molecules in broad bands (Akamatsu et al., 2017; Table 1). Therefore, the cytoplasmic concentration of Blt1 is about 0.3 µM, explaining the high cytoplasmic fluorescence. The oligomeric state of the cytoplasmic Blt1 is not known. By spinning disk confocal microscopy this cytoplasmic fluorescence is diffuse or slightly granular (our Figure. 1C; Goss et al., 2014, Figure 1; Saha and Pollard, 2012, Figure 1).

Airyscan microscopy includes a deconvolution and pixel reassignment process called Airyscan filter strength (AF) during image reconstructions are obtained from raw images. The ZEISS software uses the Weiner filter for deconvolution and adds signals from all 32 channel detectors through pixel reassignment to increase the resolution. This process also enhances the contrast of irregularities in the cytoplasmic fluorescence, which Reviewer 1 called Class 1 spots.

To eliminate the interference from the cytoplasm, including the class1 specks, with ‘unitary nodes’ we (1) used rigorous criteria to identify unitary nodes and (2) subtracted the intensity of the cytoplasmic background including the class1 specks from the intensity of each unitary node. Candidate unitary nodes were chosen by eye and then confirmed by the step-by-step procedure described in detail on lines #699720. All unitary nodes were brighter than the cytoplasmic class 1 specks. The lowest bin (0-400) in the histogram of the frequency of node intensities (Figure 2D) has negligible numbers of particles, showing that our process eliminated the class1 specks.

We calculated the number of cytokinesis nodes in a cell by dividing the total, background-subtracted Blt1-mEGFP intensity of the fluorescence around the equator by the average intensity of a backgroundsubtracted unitary node. So far, this is the best way to estimate the node number.

We should also note problems with previous attempts to count nodes by our lab and others. The nodes in 3D reconstructions of confocal Z-series of cells (Moseley et al., 2009; Vavylonis et al., 2008) varied considerably in fluorescence intensity (see Vavylonis et al., 2008, Figure. S1C), so their counts included clusters of unresolved nodes. Deng et al. (Deng and Moseley, 2013), Pan et al. (Pan et al., 2014), and Willet et al. (Willet et al., 2019) used the Find Maxima macro function in ImageJ to find maximum intensity pixels in maximum projection images. They ignored dim nodes by choosing the only pixels in ROI with intensities greater than approximately twice the mean background intensity. Allard et al. (Allard et al., 2018) used the superior (140 nm) resolution of Airyscan microscopy and an automated particle tracking plugin to count ~50 nodes tagged with Cdr2-mEGFP. However, the algorithm used in the plugin was designed to find only local intensity maxima and did not count the dim nodes with low intensities. Unfortunately, it was impossible to make 3D reconstructions of whole, living cells from the superresolution data, so Laplante et al. (Laplante et al., 2016) estimated the total numbers by extrapolation. Akamatsu et al. (Akamatsu et al., 2017) compared images from a diffraction-limited confocal microscope and FPLAM. They showed that interphase nodes are discrete structures with brighter spots having a multiple of the intensity of a large population of dim spots, which they called ‘unitary nodes’. Both FPALM and confocal microscopy have limitations, so in this paper, we used Airyscan microscopy due to its ability to do 3D reconstruction and super-resolution. We followed the analysis method used by Akamatsu et al. (Akamatsu et al., 2017) to count cytokinesis node numbers with high resolution.

Issue 3. Is showing that cytokinesis node numbers scale with cell size novel and does it extend our knowledge of cytokinesis? The answer is “yes,” because this is the first time that cytokinesis node numbers have been counted accurately owing to prior limitations in the imaging and/or analysis methods. Furthermore, it is the only analysis in mitotic cells, the only time that cytokinetic nodes exist.

Reviewer #2 (Public Review)Overall, the study fits with past work in the field and strengthens the model that node number scales with cell size. The conceptual advance beyond these past studies appears limited, and in several places the conclusions are not sufficiently supported by the data. I also have some concerns regarding the new method for estimating node number as discussed below.The paper mostly confirms previous studies that also showed node number scaling with cell size. The authors have done a nice job of using small and large mutants to increase the range of cell sizes beyond previous work, but the conclusion remains very similar.The conceptual advance beyond these past studies appears limited, so the paper mostly confirms previous studies.

Previous work showed that the numbers of interphase nodes increased with the two-fold increase in cell size across the cell cycle (Pan et al., 2014), but the method employed missed most of the unitary nodes. On the other hand, we used four mutant strains along with WT cells to study for the first time the dependence of the numbers of cytokinesis nodes during mitosis over a five-fold range of cell sizes at a specific cell cycle time. Thus, our study alone establishes that the number of cytokinesis nodes scales with cell size at a specific time point in the cell cycle during mitosis, an essential parameter for understanding the mechanism of cytokinesis. Therefore, our study is not simply confirmative but is really the first to count all of the cytokinesis nodes accurately over a much wider range of cell sizes.

Reviewer #1 (Public Review):In Figure 2D, a histogram of their intensity distribution and the Gaussian fitting to it were displayed as supporting evidence for their monodispersity. However, a statistical test for normality is missing. A proper test such as the Kolmogorov-Smirnov (K-S) test and the Shapiro-Wilk test is necessary.

As stated in the paper, we used the D’Agostino-Pearson test which is more versatile and powerful than the Kolmogorov-Smirnov (K-S) test or the Shapiro-Wilk test for normality. Please see line #225

One of the possible criteria for the "unitary nodes" would be the localization at the cell cortex as the authors implicitly assume. An appropriate marker for the plasma membrane or the cell wall would provide a solid reference for this.

The boundary of the cytoplasm against the plasma membrane can easily be determined from the diffuse cytoplasmic fluorescence, so a plasma membrane marker is not needed.

Reviewer #1 (Recommendations for the authors):To clarify the points raised in the Public Review, 3D distribution of Blt1 was reconstituted from the provided data, NodeBroadband.czi, by a custom ImageJ/Fiji script (annotate_nodes.ijm script) and exported as ‘overview.mov’. The signals that are comparable to the ‘unitary nodes’ and slightly brighter than the dimmest cytoplasmic spots can be found in the polar regions as well (blue arrowheads in summary.PDF). 10 ROIs in Node_RoiSet.zip were marked by color. The 3D reconstruction of the equatorial broad band region (nodes.mov) and its maximum y-projection (projection along the long cell axis, nodes_max_prj.tif) indicate that 2 of them (magenta) are not on the cell surface. Please find these files at https://www.dropbox.com/sh/gbay207esrvzs78/AADramgOx_7Q0oUck8kmoiexa?dl=0.Indeed, 2 out of 10 ROIs on the sample image provided by the authors reside slightly inside the cell, clearly not on the cortex or cell membrane as the others do. These non-cortical spots are inconsistent with the widely accepted picture of the cytokinesis nodes.

Thank you for taking the time and effort to reanalyze our raw data. The two ROIS (magenta) in the maximum y projection (projection along the long cell axis, nodes_max_prj.tif) are not nodes but the Sad1-RFP signal from spindle pole bodies. As mentioned in the text on lines #138 -142 and in the legend of Figure 1, we imaged spindle pole bodies to confirm that the nodes being considered are cytokinesis nodes. We apologize to the reviewer for the incomplete information in the ReadMe.txt file to analyze the sample data (‘NodeBroadband.czi’). Although it was mentioned that it is a two-colored Z-stack image, it was not clear that channel 1 has green fluorescence from Blt1-mEGFP and channel 2 has magenta Sad1-RFP signal from spindle pole bodies.

The FFT of an image depends both on the microscope and the sample. To compare the abilities of different microscopy techniques, ideally, the same sample should be observed. The cells in Figure 1A (Airyscan) and 1C (spinning disk) don't look like the same cell.

We could not use the same cell for imaging both ways for quantitative measurements due to photobleaching.

What does the diagonal line lowering towards the right in Figure 1B represent? A (dim) structure that exists only in the Figure 1A 'CR' not in Figure 1C 'CR'? Or, an artifact by Airyscan microscopy?

The diagonal line lowering towards the right in Figure 1B represents dark unwanted pixels included in the top left corner of Figure 1A, left lower panel when rotating the image to make Figure 1B. They are removed.

Reviewer #2 (Public Review):This paper studies cytokinesis nodes, which are precursors of the cytokinetic ring in fission yeast *S. pombe* cells. The project addresses a timely question in the field, and the authors' data support their conclusion that the number of nodes scales with cell size.Overall, the study fits with past work in the field and strengthens the model that node number scales with cell size. The conceptual advance beyond these past studies appears limited, and in several places the conclusions are not sufficiently supported by the data. I also have some concerns regarding the new method for estimating node number as discussed below.Strengths1. In the Introduction and the Discussion section, the authors do an excellent job of comprehensively discussing previous studies that have investigated how nodes scale with cell size.

Thank you for the compliment. Given the comments of reviewer 1, we reorganized the introduction to clarify what is known about the structure of single cytokinesis nodes from single-molecule localization microscopy and to explain that larger node structures are clusters of these single (unitary) nodes.

2. Several unexpected results reveal novel layers of node regulation that can be studied in the future. For example, nodes were more dispersed in small wee1-50 cells and size scaling of nodes was altered in wide rga4∆ cells. Since these two mutants are shorter that other cells examined, it suggests that some unidentified mechanism related to cell length is altering the known node regulatory system.Weaknesses:Some aspects of node positioning and density appear to break down at the extreme sizes – e.g. node distribution changing in small wee1 and rga4 mutants, and node density failing to scale with cell size in large cdc25-22 mutants. These exceptions seem potentially very interesting but are not explored.

Thanks for making this point, which we analyzed and included in the revised paper. The reviewer is correct. Separating the *cdc25-22 mutants* from the other strains (Figure 3K) shows two different responses. We suggest in the discussion that experimental conditions, that is freely cycling cells vs. arrested and released cells contribute to this difference.

3. Figure 3: The authors show that the number of nodes scales with cell volume and with cell length across a range of sizes (except for rga4 mutant). Given that size involves many parameters including volume, length, surface area, etc, I am confused that the major conclusion and title of the paper specifies that nodes scale with "volume." Recent studies from the groups of Fred Chang and Martin Howard have shown a role for cell surface area as opposed to volume or length in node pathways. The authors may have missed an opportunity to extend their results by comparing and contrasting how nodes scale with different aspects of cell size/geometry. As presented, this major conclusion of the paper does not extend beyond previous studies and perhaps overstates the importance of cell volume, limiting the overall impact of the work.

Node numbers also scale with respect to cell length and surface area (see new supplementary Figure. S1). We have no reason to doubt Fred Chang and Martin Howard that node signaling pathways depend on cell surface area rather than volume or length. We agree with the reviewer to use cell size in the title and emphasize it more in the text. Cell volume is relevant when checking the dependence of node number on the cytoplasmic concentration of molecules forming nodes.

Our conclusion does not contradict the previous studies, because we showed that the number of nodes depends upon the cell size, which later senses the surface area and decides the cell size at division through the Cdr2-Wee1-Cdk1 pathway. In fact, we may be pointing out a feedback mechanism between the cell size and nodes.

4. Figure 3K: The authors conclude that node density in the cell middle does not scale with cell size. However, the graph appears to be driven by the elongated cdc25-22 mutant measurements. To my eye, it looks like node density scales with cell length in all other strains but then does not scale specifically in the cdc25-22 mutant. It is important because the authors' conclusion contrasts with previous studies in the field (e.g. Pan et al., 2014).

We agree with the reviewer that the node density scales with cell length in all the strains except in *cdc25-22* mutant cells. In *cdc25-22* mutant cells, the nodal density remained unaffected over the range of cell lengths. We reanalyzed the two sets of data in Figure 3K separately, revised the text, and added a possible explanation to the discussion. Thanks. Please see lines #318-321

5. Figures 4-5: These figures examine how node number scales with Pom1 concentration using Blt1mEGFP as the marker for nodes. However, the role of Pom1 on nodes appears to be through Cdr2, which is not imaged in these experiments. In addition, the earlier figures in the paper focused on mitotic cells whereas these figures and the role of Pom1 are on interphase cells.

The point of Figure 4A is to show the distribution of Pom1 in different strains across the life cycle. To our knowledge, this has not been reported for the *cdc25-22* strain and is a prerequisite for interpreting the overexpression experiment. Figure 4B-D and Figure 5 are still focused on mitotic cells. Figures 4B-D show that the Pom1 density in the equatorial cortex of mitotic cells is similar in strains with a range of sizes. Figure 5 shows how variation in total cellular Pom1 in WT cells influences the numbers and densities of cytokinetic nodes and cell size, which is an essential parameter that influences node numbers independent of Pom1. We agree that Cdr2 is likely to mediate any effect of Pom1 on the node numbers, but we imaged Blt1, the best marker for cytokinesis nodes, rather than Cdr2 for consistency with the rest of the paper.

6. Figure 6: The authors conclude that when Cdr2 concentration is increased, it results in smaller cells and more nodes. However, the actual experiment shows that restoring Cdr2 expression into a cdr2 deletion mutant restores cell size and node regulation.

This is not a complementation experiment as assumed by the reviewer. The text states “we replaced the Cdr2 promoter in the *S. pombe* genome with the nmt81 (no message in thiamine) thiamine repressible promoter (Maundrell, 1993).“

I do not see data indicating that Cdr2 overexpression reduces cell size below wild type levels. Instead, the data support the expected conclusions for restoring the function of a deleted gene.

We did not delete a gene but simply used the nmt81 promoter and thiamine to control the Cdr2 expression. The *S. pombe* strain with thiamine and the nmt81 (no message in thiamine) promoter show longer (mean length ~18 µm) than WT (~15 µm) cells at division when both strains expressed similar Cdr2 levels (12 a.u.). We did not investigate this difference but used thiamine to adjust the Cdr2 level in the nmt81-Cdr2 strain. The cell size declined with the Cdr2 expression level but not below WT cells (Figure 6B).

7. The authors conclude that Pom1 and Cdr2 do not act directly on cytokinesis node number but instead act indirectly by influencing cell size. This conclusion is confusing because nodes act to regulate cell size at mitosis prior to their role in cytokinesis. The authors discuss some of the pathways connecting these proteins to cell size in the Discussion, but it becomes confusing as interphase nodes and cytokinesis nodes are not fully distinguished in the discussion. If I understand their logic correctly, their main point is that interphase nodes controlled by Cdr2 and Pom1 do not influence the number of cytokinesis nodes later in the cell cycle. If this is correct, then I might suggest clarifying the text.

Thank you for the suggestion. We agree that nodes regulate cell size as observed in our experiments. We clarified this point in the text as suggested. Please see line # 543-545

Reviewer #2 (Recommendations for the authors):I found the logic of the Discussion section hard to follow at times. The authors make a number of interesting points but might consider consolidating some of their conclusions.

Thanks for the suggestion. We used the advice from the reviewers to reorganize and improve the discussion*.*

Reviewer #3 (Recommendations for the authors):Using quantitative Airyscan microscopy, Sayyad and Pollard present detailed measurements of the number of cytokinetic nodes, the scaling of their number with cell size, as well as concentration of Pom1 and Cdr2. This work is a great contribution to quantitative studies of cytokinesis and more broadly to regulation of cell size. It's an invaluable resource for the development of mathematical models of cytokinesis and cell size regulation.I am very enthusiastic about this work but I have a few comments to consider.1) Node counting method. The counting of node numbers is based on calibrating the intensity of unitary nodes. This results in a number of nodes for WT cells that is considerably larger (~ 200) than earlier estimates, even by Laplante et al. who used single molecule super-resolution microscopy to measure ~140 cytokinetic nodes in cdc25-22 cells.

We thank the reviewer for endorsing the importance of our study for the quantitative biology field.

Lines #97-107 and 473-474 in the paper explain the limitations of the method used by Laplante et al. (Laplante et al., 2016). Because photobleaching is an intrinsic feature of single-molecule localization, Laplante et al. (Laplante et al., 2016) could only image a 400 nm thick slice of each cell. They counted the nodes in a small segment of the cell surface and extrapolated the density of nodes around the circumference of the cell to estimate the number of nodes. These were the best counts to date, but they underestimated the total by about 35% according to our new data. We collected a Z-stack of sections and made a 3D reconstruction from which we measured the total Blt1-GFP intensity near the plasma membrane around the entire equator of the cells. Dividing to total fluorescence by the average intensity of a unitary node gave a count of the number of nodes. In both cases, the numbers are much higher than studies that overpassed unitary nodes.

While Laplante et al. concluded that both WT and cdc25-22 arrested and released cells have the same number of nodes, in this paper (Figure 3H) as well as in Vavylonis et al. (2008), cdc25-22 cells were found to have more cytokinetic nodes as compared to WT

Thank you for raising this important question. We added to the text the original observation of Vavylonis et al., 2008 with regard to more nodes in longer *cdc25-22 arrested and released cells.*

If the estimate of WT nodes in Laplante et al. needs to be revised downwards, to resolve this apparent disagreement, such a change would make the difference in node number counting between this paper and earlier studies even larger.

Laplante et al. (Laplante et al., 2016)’s conclusion that WT and *cdc25-22* arrested and released cells have the same number of nodes was based on a comparison of the total Rlc1p-3GFP intensity of contractile rings rather than measurements of nodes in the broad bands of WT and *cdc25-22* mutant cells. The intensities were not significantly different, so they concluded that node numbers are similar in WT and *cdc25-22* arrested and released cells. However, interpreting this data from contractile rings is complicated by the fact that a large fraction of the Rlc1 in contractile rings is associated with the Myp2 heavy chain, which is not a node component. Clearly, more work needs to be done on nodes in contractile rings.

Laplante’s estimate of the number of nodes depends upon extrapolation of the local density of nodes to the surface area of the node broad band, which is a function of the width and circumference. The diameters of WT and *cdc25-22* mutant cells are similar, so the number of nodes depends upon the width of the broad band of nodes. The broad band width shown in their Figure. S2E (~3.0 µm) is half of that we measured in *cdc25-22* mutant cells (> 6.0 µm). Using 6.0 µm would give an extrapolation to >280 nodes, which agrees more closely with our measurement.

Evidence for unitary nodes comes the observed quantization of node intensities by Akamatsu et al. (2017) who fitted histograms of node intensities by multiple Gaussians. However, histograms with small number of data per bin show fluctuations that may appear as peaks, which change depending on the bin width. I am sure that Akamatsu et al. were very careful in their analysis, but I couldn't find how this issue of binning fluctuations was dealt with in their paper. The current paper does not attempt to reproduce the quantization seen by Akamatsu et al. (which could also have provided the intensity of unitary nodes). Given the above comments, I am concerned that the intensity distribution of what are considered unitary nodes in Figure 3G might reflect the tail of a broader intensity distribution (representing a larger variation of node intensities whether unitary or not), which is cut off by a selection procedure. This might result in overcounting of nodes.

The higher resolution of Airyscan simplified the search for isolated dim cytokinesis nodes, which were difficult to resolve by conventional confocal microscopy. Moreover, we used rigorous criteria to choose from these selected dim, well-resolved unitary cytokinesis nodes and subtracted background to establish their intensities.

The histogram of 200-300 unitary nodes has a single, bell-shaped peak as expected from the measurements of a uniform particle with some experimental error arising from the small numbers of molecules and their variable maturation rates and photophysical responses. The same is true for the other strains (Figure 3G). The reviewer is correct that the peak in the histogram will change according to the bin size and the number of nodes would vary accordingly and cited *Akamatsu et al. (2017) paper*. However, in our case, the improved resolution and rigorous selection criterion eliminated the larger clusters and background subtraction eliminated the small clusters from the unitary node intensity distribution and constrained the bin size (Figure 1D). We agree with the reviewer that the selection criterion may bias the measurement of unitary node intensities as the exact composition of unitary nodes will not be known until further studies with novel approaches. We feel that our imaging and analysis methods to count the cytokinesis nodes are an improvement over previous studies.

(2) Is the microscope's PSF varying significantly along distance from slide, and if so, should this be accounted for when measuring the total intensity of the broad band through sum projection?

To measure all of the fluorescence intensity in the z-direction from particles associated with the plasma membrane, we collected optical sections <7.0 µm deep from the slide. Over such a small distance from the slide, any variation in the PSF should be minimal. Furthermore, the PSF is considered during the reconstruction of the Airyscan image from raw data obtained in 32 channel Airyscan detector. For each pixel, this 3D processing analyzes the distribution of intensities over the Airyscan array and determines any axial shift in the PSF. The axial resolution is improved by rejecting a signal from pixels that display suboptimal PSF.

3) Scaling of nodes with volume versus area. The scaling of number of nodes with volume is emphasized in this work. However Pan et al. (2014) had reported that Cdr2 and cell size at division scales with surface area, see Figure 6 in that paper. I was confused on whether the measurements between these two papers are in disagreement or else if they reflect different phenomena.

The papers differ in two ways, so they reflect different phenomena. First, Pan et al. (Pan et al., 2014) studied interphase nodes, which differ in composition from the cytokinesis nodes we studied at the definite time point in the cell cycle to avoid time-varying compositional changes. Second, our rigorous methods are designed to count all the cytokinesis nodes. The method of Pan et al.(Pan et al., 2014) ignored dim nodes and counted <50 interphase nodes, only those with high-intensity pixels. Due to the changing interphase node composition and undercounting the relationship between interphase node numbers and surface area is uncertain.

Could an area scaling be related to how cells such as rga4Δ that have different diameter deviate the most from the trend line in Figure 3H?

The number of cytokinesis nodes also scales with the surface area (new Supplemental Figure S1) and *rga4Δ* mutant cells deviate from the trend line in Figure 3H despite surface area at division similar to WT cells. The largely biological noise in the data makes it impossible to make fine distinctions without much further work.

(Incidentally, it's hard to observe different diameters between WT, rga4Δ and rga2Δ in the images of Figure 3B,D,E). Some explicit discussion would be helpful.

The reviewer is correct, the differences in diameters between WT, *rga4Δ,* and *rga2Δ* cells are only around 0.5 µm. The difference in diameters between *rga2Δ* and *rga4Δ* is more evident.

*4)* Use of unbiased statistical/data clustering methods might provide a more reliable unitary node calibration method. An alternative approach could be to be more conservative in terms of absolute node numbers and instead present the results in relative terms.

We thank the reviewer for the suggestions. We understand the bias in the method due to manual detection of dim spots and the initial screening for a prospective unitary node is by eye. Therefore, we used two rigorous criteria to screen for unitary as explained on lines #699-720. These criteria are (1) the fluorescence intensity should be within 7 consecutive z-slices with high intensity in the middle image and decreasing towards the ends, and (2) the intensity in 7 z-slices should present within a 12-pixel diameter circle. We also subtracted the cytoplasmic background. So far this is the best method available to estimate the node numbers. In the future, automatic image segmentation and selection of nodes would remove any bias.

5) Figure 1 shows a comparison between Airyscan and confocal images to indicate the improved resolution of Airyscan. However, both of these images appear to have less signal to noise and resolution as compared to prior published confocal images of nodes with similar fluorophores. So the improvement in resolution is not superficially obvious.

The reviewer is correct. The confocal image in Figure 1 is not up to our standard. We used a scanning confocal microscope and not a spinning disk confocal microscope to compare with Airyscan imaging. In fact, Figure 1 is obtained using the same microscope in two different modes: Laser scanning confocal microscope and Airyscan mode to avoid any optical differences. To further visualize the differences in the resolution of the two imaging techniques we used Fast Fourier transform analysis of images with cells containing a definite structure such as a contractile ring. The horizontal line in the frequency domain of the Airyscan image corresponds to the contractile ring (Figure 1B), which is absent in the frequency domain of the confocal image (Figure 1D). The retrieval of higher frequencies and more extensively distributed information in the frequency domain of the Airyscan image showed a higher resolution compare to the confocal image.

6) The authors make the general assumption that unitary cytokinetic nodes are similar in terms of composition and stoichiometry. Can the authors discuss the implications of this assumption through the paper?

Thank you for asking us to emphasize more strongly our basic assumption that unitary cytokinetic nodes are similar in terms of composition and stoichiometry. Laplante et al. (Laplante et al., 2016) used Fluorescence Photoactivation Localization microscopy (FPLAM) to show that cytokinesis nodes are discrete structures with stoichiometric ratios and distinct distributions of constituent proteins.

Akamatsu et al. (Akamatsu et al., 2017) confirmed that interphase nodes are discrete structures with the brighter spots in confocal micrographs having multiples of the intensity of a large population of dim spots, which they called unitary nodes.

For example, do all Blt1 nodes become cytokinetic nodes and how is stoichiometry preserved when Cdr2 concentration is varied (is the Blt1 to Cdr2 ratio in nodes constant?).

Akamatsu et al. (Akamatsu et al., 2014) reported that early in the mitosis 75% of type 2 nodes containing Blt1 colocalize with type 1 nodes containing Cdr2 at the plasma membrane around the nucleus before they condense into a contractile ring. We found that about 75% of cytokinesis nodes marked with Blt1 combined to form a contractile ring, which may be a coincidence but may reflect the failure of Blt1 nodes without Cdr2 to incorporate into the ring.

The reviewer’s questions about the preservation of the stoichiometry when Cdr2 concentration is varied (is the Blt1 to Cdr2 ratio in nodes constant?) is beyond the scope of the current study and can be addressed in future studies along with many other questions about the composition of the nodes.

References:

Akamatsu, M., Berro, J., Pu, K. M., Tebbs, I. R., and Pollard, T. D. (2014). Cytokinetic nodes in fission yeast arise from two distinct types of nodes that merge during interphase. *Journal of Cell Biology*, *204*(6), 977–988. https://doi.org/10.1083/jcb.201307174

Akamatsu, M., Lin, Y., Bewersdorf, J., and Pollard, T. D. (2017). Analysis of interphase node proteins in fission yeast by quantitative and superresolution fluorescence microscopy. *Molecular Biology of the Cell*, *28*(23), 3203–3214. https://doi.org/10.1091/mbc.e16-07-0522

Allard, C. A. H., Opalko, H. E., Liu, K.-W., Medoh, U., and Moseley, J. B. (2018). Cell size–dependent regulation of Wee1 localization by Cdr2 cortical nodes. *Journal of Cell Biology*, *217*(5), 1589–1599. https://doi.org/10.1083/jcb.201709171

Deng, L., and Moseley, J. B. (2013). Compartmentalized nodes control mitotic entry signaling in fission yeast. *Molecular Biology of the Cell*, *24*(12), 1872–1881. https://doi.org/10.1091/mbc.E13-02-0104

Goss, J. W., Kim, S., Bledsoe, H., and Pollard, T. D. (2014). Characterization of the roles of Blt1p in fission yeast cytokinesis. *Molecular Biology of the Cell*, *25*(13), 1946–1957. https://doi.org/10.1091/mbc.E13-06-0300

Laplante, C., Huang, F., Tebbs, I. R., Bewersdorf, J., and Pollard, T. D. (2016). Molecular organization of cytokinesis nodes and contractile rings by super-resolution fluorescence microscopy of live fission yeast. Proceedings of the National Academy of Sciences, 113(40), E5876–E5885. https://doi.org/10.1073/pnas.1608252113

Moseley, J. B., Mayeux, A., Paoletti, A., and Nurse, P. (2009). A spatial gradient coordinates cell size and mitotic entry in fission yeast. *Nature*, *459*(7248), 857–860. https://doi.org/10.1038/nature08074

Pan, K. Z., Saunders, T. E., Flor-Parra, I., Howard, M., and Chang, F. (2014). Cortical regulation of cell size by a sizer cdr2p. *ELife*, *2014*(3), 1–24. https://doi.org/10.7554/*eLife*.02040

Saha, S., and Pollard, T. D. (2012). Anillin-related protein Mid1p coordinates the assembly of the cytokinetic contractile ring in fission yeast. *Molecular Biology of the Cell*, *23*(20), 3982–3992. https://doi.org/10.1091/mbc.E12-07-0535

Vavylonis, D., Wu, J.-Q., Hao, S., O’Shaughnessy, B., and Pollard, T. D. (2008). Assembly Mechanism of the Contractile Ring for Cytokinesis by Fission Yeast. *Science*, *319*(5859), 97–100. https://doi.org/10.1126/science.1151086

Willet, A. H., DeWitt, A. K., Beckley, J. R., Clifford, D. M., and Gould, K. L. (2019). NDR Kinase Sid2 Drives Anillin-like Mid1 from the Membrane to Promote Cytokinesis and Medial Division Site Placement.

*Current Biology*, *29*(6), 1055-1063.e2. https://doi.org/10.1016/j.cub.2019.01.075

Wu, J.-Q. Q., Sirotkin, V., Kovar, D. R., Lord, M., Beltzner, C. C., Kuhn, J. R., and Pollard, T. D. (2006). Assembly of the cytokinetic contractile ring from a broad band of nodes in fission yeast. *Journal of Cell Biology*, *174*(3), 391–402. https://doi.org/10.1083/jcb.200602032

[Editors’ note: what follows is the authors’ response to the second round of review.]

The Reviewers and editors are enthusiastic about the new approach of using Airyscan to investigate node architecture, properties, and scaling. However, the Reviewers have raised many similar questions, which require new experiments and / or analysis of existing data.Should you be able to carry out the necessary experiments / analyses, eLife will be able consider the manuscript as a new submission (sent to the same three Reviewers).Please find below the comments made by the Reviewers to your rebuttal letter and the revised manuscript.Reviewer 1:I have read the authors' rebuttal and remain concerned about limited impact and significance of the work. I was unable to access the revised manuscript, which is referenced in the rebuttal letter, but the changes and responses seemed clear from the point-by-point response. There are 3 main concerns that are unchanged from the initial submission (and I would also note that the authors did not really address several of my other comments).1. The authors have measured total blt1-mEGFP signal in the cell middle, and then assumed that all of this signal is comprised of single, unitary nodes. Based on this assumption, they estimate the number of nodes in the cell middle. This approach relies heavily on two major assumptions: (a) that all blt1-mEGFP signal is contained in nodes, and (b) that all nodes are the same size. The important concern is that the authors have not in fact counted the number of nodes, but rather they have estimated their numbers based on a method that relies heavily on assumptions. A major claim in the paper is improved node counting over past publications, but there appear to be just as many caveats in the methods in the current paper as in past publications.

We do not understand how the reviewer evaluated our revised paper without access to the revised document. Was this review written in response to our appeal of the rejection rather than after we submitted the revised manuscript?

The reviewer missed the important point in our response to review that we did not just measure the Blt1-mEGFP signal in the cell middle but also subtracted the cytoplasmic background from it to avoid including Blt1 diffusing in the cytoplasm. We used a SUM projected image of 40 z-slices containing the entire fluorescence of a cell to calculate the total cytoplasmic background in the node broad band region. From the SUM projected image, the mean pixel intensity in the cytoplasm near cell tips, where nodes were absent was multiplied by the node broad band area that is the total number of pixels in the node broad band area to calculate the cytoplasmic background in the node broad band region. We subtracted this cytoplasmic background from the total intensity of the broad band region to calculate the intensity contributed by Blt1-mEGFP in the nodes broad band. The methods section, which the reviewer apparently did not see, provides a detailed explanation of the approach. Subtracting cytoplasmic background ensures that we measured Blt1-mEGFP signal only from nodes in the broad band near the plasma membrane around the equator of the cells.

The reviewer assumed incorrectly that we considered that all of the Blt1-mEGFP signal is contained in nodes. In fact, we explained in the response to review and the revised manuscript that only 25% of Blt1 fluorescence is in nodes and 75% is in the cytoplasm as shown previously by our group (Akamatsu et al., 2017).

The reviewer’s second assumption is correct, namely that we considered all unitary nodes to have the same size and composition, however, we also assumed that unitary nodes might cluster too closely to be resolved. We did not find any other fluorescence microscopy method to count nodes more accurately than the one we used.

2a. Regardless of differences in the methods of counting, the current paper reaches the same conclusion as past publications: node number scales with cell size. Node number scaling with cell size is an interesting correlation that fits with past publications, but the functional significance of this scaling for cytokinesis is not investigated limiting the impact of the work.

The reviewer disregarded our replies to this question in the response to review and revised manuscript. We studied cytokinesis nodes at specific time in the cell cycle, early mitosis, and not during interphase as done previously by others. Showing that number of cytokinesis nodes during mitosis scales with cell size is novel in two ways: first, as elaborated in the text, previous studies from our lab and others did not count the number of any type of node accurately owing to limitations in the imaging and/or analysis methods. The revised text and our response to review both explained how Airyscan microscopy is more suitable to count the total number of nodes than confocal or single molecule localization microscopy and why previous studies undercounted them. Confocal images do not resolve many closely spaced unitary nodes. To make matters worse, some labs used selection criteria that excluded dim unitary nodes. Pan et al. (Pan et al., 2014) reported that the number of nodes increased across the cell cycle as cells grow in size, but their counting method missed most of the unitary nodes and thus undercounted the number of nodes. Most of the “nodes” they considered were actually clustered interphase nodes, which differ in composition from cytokinesis nodes. Second, we studied for the first time the dependence of the numbers of cytokinesis nodes in early mitosis over a five-fold range of cell sizes in wild-type (WT) and four mutant cells. Thus, our study alone establishes that the number of cytokinesis nodes scales with cell size, an essential parameter for understanding the mechanism of cytokinesis.

2b. Do large cells need more nodes to form a functional cytokinetic ring? What are the differences in assembly of a cytokinetic ring formed by fewer nodes (i.e. small cell) versus a ring formed by more nodes (i.e. large cell)? Without functional investigation into this scaling, the overall impact of the work is limited.

We thank the reviewer for raising this interesting point. We measured the Blt1-mEGFP fluorescence in the contractile rings of cells with a range of sizes after subtracting the diffuse Blt1mEGFP in the cytoplasmic background as explained in the methods section. We added a figure (Figure 3—figure supplement 2) and describe the observations in the results and Discussion sections of the main text.

Wild-type cells incorporated 75% of Blt1-mEGFP fluorescence of cytokinesis nodes in the broad band into a contractile ring (Figure 2E). The mutant strains *cdc25-22, wee1-50* and *rga2Δ* cells incorporated 50-75% of Blt1-mEGFP fluorescence of broad band cytokinesis nodes into contractile rings, while *rga4Δ* mutant cells incorporated most of the Blt1-mEGFP fluorescence of cytokinesis nodes in the broad band into the contractile ring (Figure 3—figure supplement 2, A).

The total Blt1-mEGFP fluorescence in contractile rings increased with cell size but was about 25% lower than the total Blt1-mEGFP in the cytokinesis nodes in broad bands of wild-type cells and 25-50% less in the contractile rings of *cdc25-22, wee1-50, and rga2Δ* mutant cells than in their broad bands. Therefore, judging from total Blt1-mEGFP fluorescence, larger cells assemble contractile rings with more cytokinesis node components than small cells, although some Blt1 is lost during this transition. This loss may be related to the observation that 10 min before spindle pole bodies divide, 75% of late interphase nodes have combined to form cytokinesis nodes with markers of both type 1 and type 2 nodes (Akamatsu et al., 2014) before maturing by accumulating Myo2, IQGAP Rng2, F-BAR Cdc15, and formin Cdc12 (Wu et al., 2003). About 25% of both type 1 and type 2 nodes were excluded from this process and may be left behind when the cytokinesis nodes form the contractile ring. This is a topic for future research, well beyond the scope of this study.

Contractile ring formation in *cdc25-22* mutant cells is a special case that differs from the other strains, since about the same amount of Blt1-mEGFP was incorporated into rings in cells that varied in size and the total Blt1-mEGFP in their cytokinesis nodes (Figure 3—figure supplement 2, B). On the other hand, like wild-type cells, the contractile rings of *cdc25-22* mutant cells did recruit mEGFP-Myo2 in proportion to their volumes (Figure 3—figure supplement 2, B). Therefore, the longest *cdc25-22* mutant cells incorporated a much larger fraction of the Myo2 from the cytokinesis nodes into the rings than Blt1. Arresting and then releasing the cell cycle of the *cdc2522* mutant cells at the G_2_/M transition may contribute to this imbalance, which is greater than in any of the strains allowed to cycle normally. This new observation indicates that nodes observed in rings by single-molecule localization microscopy are likely to have different compositions than cytokinesis nodes. Documenting and characterizing this maturation of nodes transitioning into contractile rings is worthy of further exploration but is well beyond the scope of this study.

3. Several pieces of data show that node number and node density do not simply scale with cell size. rga4 mutants do not scale node number with cell volume compared to wild type, and cdc25-22 do not scale node density with cell volume. The fact that these mutants break the scaling properties means that nodes do not simply scale with cell size. To extend beyond the previously published conclusion that node number scales with cell size, the authors could have investigated why these mutants lose scaling and what is means for cytokinetic ring assembly. Without such investigation, the impact of their work is limited to supporting past conclusions on these structures.

The number of cytokinesis nodes in wild type, *wee1-50, rga2Δ,* and *cdc25-22* mutant cells increased with the cell length, volume, and surface area, the exception being in *rga4Δ* mutant cells. Our revised discussion explained that even if this is the case the mode of increasing the node numbers in different cells may differ. Figure 3J-K shows that the high number of cytokinesis nodes in *cdc25-22* mutant cells correlates only with the increase in broad band width. In other cells both broad band width and node density contribute to increasing the numbers of cytokinesis nodes in larger cells. As in other cells, the density of nodes in *cdc25-22* mutant cells did not vary to increase the cytokinesis node number may depend upon the experimental conditions. For example, we did not interfere in the cell cycle of any mutant cells which move naturally through the cell cycle. The nodes in these cells sense the surface area and use the Cdr2-Wee1-Cdk1 pathway to reach the mitotic threshold of node density or number. On the other hand, we use temperature to hold the *cdc25-22* mutant cells at the G2/M transition for hours, until releasing them into mitosis. The cells grow at their tips while arrested, forming extra nodes after reaching the threshold of node density. These extra nodes have room to spread out at the edges of broad band towards the tips of the long arrested cells. So, in these cells node broad band is wider.

Reviewer 2:I read the paper and modified my review as follows. I have included a dropbox link to an image (that could be provided through eLife dropbox link, thought it doesn't matter since I have provided my name).The authors have addressed many of my questions and the paper is now much clearer in terms of the assumptions underlying this work. I am still however similarly concerned regarding the node counting method.The identification of dim nodes as unitary ones, as the authors describe in their response, relies heavily on the findings of Akamatsu et al. who reported quantization of node intensities.For example, it is stated on line 180: "We interpret the dim nodes as unitary nodes, while the brighter nodes consist of two or more unitary nodes" and on line 707: "We searched for dim spots and marked their 708 X-Y positions. Then we inspected the fluorescence along the Z-axis at these X-Y positions in 40 Z- slices. To qualify as a unitary node, a spot had to satisfy two conditions: (1) the fluorescence of a spot was present within a circle 12 pixels (~0.5 µm) in diameter; and (2) the fluorescence of a spot was present in 7 or fewer consecutive Z-slices with a peak in the middle slice and diminishing towards both sides"From these statements I understand that the authors did not include bright nodes in the intensity distributions (such as in Figure 2D), even if they satisfied conditions (1) and (2) because these bright nodes were considered to be multiples of unitary nodes. I assume this is why the intensity distribution in Figure 2D does not have a long tail at high intensities, unlike the intensity distributions of Akamatsu et al. or Figure S6 of Laplante et al.

Note that (1) Blt1 nodes with multiples of the unitary node fluorescence intensities in Akamatsu et al. (Figure 2H, 2017) were interphase nodes, not the cytokinesis nodes studied here and, (2) those data were collected by standard spinning disk confocal microscopy, not Airyscan microscopy used here. Therefore, the differences are real and not unexpected.

Now superficially, the number of data points in the histograms of Akamatsu et al. seem to me to leave some room for ambiguity with respect to interpreting the high intensity tail as integer multiples of unitary node intensity. To better illustrate this simple point, that may have been unclear in my first review, I attach graphs showing how 150 points (similar to the 138 node counts in Akamatsu) picked from a singly-peaked distribution may appear as multiple peaks as a result of fluctuations:https://www.dropbox.com/s/suti2gz9ovgmfyw/histogram.pdf?dl=0One could also argue that the fact that the width of intensity distribution in Figure 2D is comparable to its average indicates that the node size is not very tightly regulated, leaving room for other interpretations (for example node formation though nucleation and growth and a continuous size distribution with a long tail above a minimal nucleus).Perhaps the authors could verify or critically reexamine the unitarity of nodes, which is so central in this work, in the light of the improved microscopy method. That may include quantifying brighter nodes (along the lines of Akamatsu) as well as the low intensity signal (such as measurements of signal to noise).

We thank the reviewer for their comments and suggestions. However, before explaining the new data we collected in response to this request, we must emphasize that the strongest evidence for the unitary node concept comes from the FPALM super resolution of Laplante et al. (Laplante et al., 2016) not confocal data.

Nevertheless, we reexamined the Airyscan micrographs of our sample cells and added to our sample of nodes all that satisfied our two ROI conditions even if they were brighter than the dominant dim nodes. Only a few extra nodes fit these broadened criteria, but they did create a slightly longer tail at high intensities than with the narrower criteria as anticipated by the reviewer. However, the Gaussian fit to the main peak of the distribution of node intensities and the mean intensity did not change significantly (two-sample t test, P = 0.04). So, we did not reanalyze the data to count nodes but modified the text. The small second peak at 6000 may correspond to pairs of closely-spaced nodes. Please see below.

Thus, the higher resolution of the Airyscan largely eliminated the small clusters of unitary cytokinesis nodes that could not be resolved by confocal microscopy using Blt1 as the marker as in Figure 2B of Akamatsu et al. MBoC 2017.

Reviewer 3:I regret that I used the same pair of colors (green and magenta) in my summary.pdf for Blt1 and Sad1 signals, respectively, in the NodeBroadband.czi panel (top left) and for ROIs on the cortex and inside the cytoplasm, respectively, in the other panels. However, as clearly indicated in the PDF (The caption "Blt1 channel" + color coding of the ROIs in the ROI manager, i.e., ROIs "0032-0271-0078" and "0033-0285-0077" in magenta while others in green ) and in the ImageJ/Fiji script, to analyze the node positions, only the Blt1 channel was used. As in the maximum z-projection of the original stack overlaid with the ROIs defined by the authors ("rois_on_max_prj+sad1.png", to be shared via the Editorial office), all the ROIs defined by the authors are clearly distinguished from the two Sad1 peaks, which mark the spindle pole bodies. To make this point even clearer, the Sad1-RFP signals were overlaid in red on a 3D reconstruction as well ("nodes+sad1.mp4" movie, to be shared via the Editorial office).

Thank you for accepting our explanation. This reviewer (Referee #1 in the first revision) is responding to the question raised in the previous review. The reviewer remarked that “Indeed, 2 out of 10 ROIs on the sample image provided by the authors reside slightly inside the cell, clearly not on the cortex or cell membrane as the others do. These non-cortical spots are inconsistent with the widely accepted picture of the cytokinesis nodes”. The two ROIs mentioned by the reviewer (*"0032-0271-0078" and "0033-0285-0077")* are indeed slightly separated from the plasma membrane as others have observed for a small fraction of cytokinesis nodes (Laplante et al., 2016; Wang and O’Shaughnessy, 2019). Thus, some nodes may be separated slightly from the inner surface of the plasma membrane.

We imaged cytokinesis nodes after spindle pole body separation when they start moving to condense into a narrow contractile ring. Actin filaments form transient connections between nodes and Myo2 pulls the actin filaments from other nodes. It is possible that some nodes may lose their connection with the plasma membrane at this time. The fluorescence intensities of two nodes at the two ROIs mentioned by the reviewer (*"0032-0271-0078" and "0033-0285-0077")* had higher intensities than the cytoplasmic background.

In the rebuttal, as to the unitarity of the cytokinesis nodes, the authors refer to the FPALM analysis in Laplante et al. (2016). In principle, this technique can detect only a structure that is sufficiently immobile during the data acquisition. In Materials and methods of this paper, they wrote "Clusters of localized emitters distributed around the equator of the cells were manually selected from the reconstructed FPALM images into two different categories, face views and side views. Broad bands of nodes formed during prophase, and nodes remained stationary until node movements began during metaphase. For broadband nodes (prophase/metaphase cells), nodes were cropped from images reconstructed from 5,000 frames (25 s)." Neither clear criterion for the 'clusters' nor the raw data of FPALM was provided. Being immobilized within the small region (~100 nm) for a few seconds might be too selective. We can't exclude a possibility that smaller clusters resize on the cortex, though diffusing more rapidly than the level that can be recognized as 'clusters'.

The Laplante paper reported a remarkable uniformity in the composition and sizes of cytokinesis nodes imaged over a few seconds by FPALM super resolution. Earlier papers (Vavylonis et al., 2008; Laplante et al. 2015) reported slow movements of cytokinesis nodes at ~20 nm/s when two nodes were connected by actin filaments. Therefore, we do not understand why the reviewer is postulating resizing and rapid diffusion of these structures, which were not observed.

In contrast, Airyscan can potentially capture smaller and moving clusters. Although it takes ~11 s or longer to scan a full volume of 22 µm x 22 µm x 6.8 µm (40 z slices), voxels close to each other can be completed in a shorter time (eg. < 1 s). Blt1 spots that are comparable to or slightly dimmer than those picked up by the authors as unitary nodes are found all over the cortex including the polar regions that were used as the background by the authors. Some of them were marked with blue arrows in my summary.pdf and their nature was questioned in my public review. However, this was not addressed in the rebuttal. No clear logic to treat them as just background was provided (although it might be described in the correct, revised manuscript).

Our response to review and the revised manuscript explained the uniformly dispersed Blt1 spots in the cytoplasm as possibly Blt1 oligomers. About 25% of Blt1 is located in nodes and 75% in the cytoplasm (Akamatsu et al., 2017) where it appears somewhat granular in confocal micrographs. The processing of Airyscan images increases the contrast of these dim inhomogeneities. The reviewers were concerned that these specks might be confused with the brighter unitary nodes. However, we subtracted the intensity of the background including these specks before selecting (using rigorous criteria explained in the text) and measuring the intensities of unitary nodes. No dim specks appear in the population of nodes (Figure 2D).

The nodes satisfying two conditions were considered to be ‘unitary node’. One condition is that the fluorescence should be within 7 Z- slices. These 7 consecutive Z- slices were acquired within 1 s during which node movement (20 nm/S) cannot be detected by the Airyscan (resolution is ~140 nm).

As to the statistical test of Figure 2DAs stated in the paper, we used the D'Agostino-Pearson test which is more versatile and powerful than the Kolmogorov-Smirnov (K-S) test or the Shapiro-Wilk test for normality. Please see line #225.I can't make a definite comment on this as the revised manuscript is not available. However, if the authors are referring to the same sentence as the lines 209 to 211 in the original manuscript, this is not addressing my point about Figure 2D. This sentence is about Figure 2E, in which a test was done for the normality of the number of total nodes calculated as the ratio of the total fluorescence to the fluorescence per unitary node.

Following the suggestion of Reviewer Dimitris Vavylonis, we modified Figure 2D to include more nodes with higher intensities that also satisfied the two conditions for unitary nodes. A Gaussian fit to the new distribution showed a slightly longer tail at high intensities and a small peak that may corresponds two closely spaced, unresolved nodes in the ROI. Despite this slightly long tail the overall distribution followed the normal distribution and passed the D'Agostino Pearson’s normality test (P = 0.1) (Figure 2).

References:

Akamatsu, M., Berro, J., Pu, K. M., Tebbs, I. R., and Pollard, T. D. (2014). Cytokinetic nodes in fission yeast arise from two distinct types of nodes that merge during interphase. *Journal of Cell Biology*, *204*(6), 977–988. https://doi.org/10.1083/jcb.201307174

Akamatsu, M., Lin, Y., Bewersdorf, J., and Pollard, T. D. (2017). Analysis of interphase node proteins in fission yeast by quantitative and superresolution fluorescence microscopy. *Molecular Biology of the Cell*, *28*(23), 3203–3214. https://doi.org/10.1091/mbc.e16-07-0522

Laplante, C., Huang, F., Tebbs, I. R., Bewersdorf, J., and Pollard, T. D. (2016). Molecular organization of cytokinesis nodes and contractile rings by super-resolution fluorescence microscopy of live fission yeast. Proceedings of the National Academy of Sciences,

*113*(40), E5876–E5885. https://doi.org/10.1073/pnas.1608252113

Pan, K. Z., Saunders, T. E., Flor-Parra, I., Howard, M., and Chang, F. (2014). Cortical regulation of cell size by a sizer cdr2p. *ELife*, *2014*(3), 1–24. https://doi.org/10.7554/*eLife*.02040

Wang, S., and O’Shaughnessy, B. (2019). Anchoring of actin to the plasma membrane enables tension production in the fission yeast cytokinetic ring. *Molecular Biology of the Cell*,

*30*(16), 2053–2064. https://doi.org/10.1091/mbc.E19-03-0173

Wu, J. Q., Kuhn, J. R., Kovar, D. R., and Pollard, T. D. (2003). Spatial and temporal pathway for assembly and constriction of the contractile ring in fission yeast cytokinesis. *Developmental Cell*, *5*(5), 723–734. https://doi.org/10.1016/S1534-5807(03)00324-1

[Editors’ note: what follows is the authors’ response to the third round of review.]

Essential revisions:1. Please note that addressing the point # 2 raised by Reviewer 3 is essential for the strength of the conclusions and should be performed.

We thank reviewer 3 for the constructive suggestion. As described below, we made the recommended measurements, which support our main conclusions.

2. Please also address points # 2-7 by Reviewer 2. Although they seem numerous, I have gone through the list. Most of these pertain to giving more information, reanalysis of data and the use of the most appropriate statistical treatments.

Below, we reply to point #1 in addition to the other 6 points, which are based on the issue in point 1. We note that the critique from reviewer #2 has expanded for a third time to include not just questions about the paper under consideration but also published work from our lab and other groups who showed that the number of interphase nodes scales with cell size.

Reviewer # 1:The revised analysis in Figure 2D, which now includes nodes of all intensities, provides additional support, and is consistent with prior evidence, for unitary nodes, as explained in the response letter. This addresses the main concern I had.I have a comment regarding the "dim specks" (that could possibly be cytoplasmic oligomers) mentioned in Response to Reviewer 3. The authors mention that these specks were not confused with the brighter unitary nodes. It was not clear to me if this is true because the dim specks did not satisfy the two node criteria (fluorescence within 12 pixels and 7 or fewer consecutive Z-slices with a peak in the middle slice), or else if their intensity was smaller than the first bin in Figure 2D, or both.

We thank reviewer #1 for their suggestion to analyze brighter nodes and for agreeing that this new information strengthened our paper. We also thank the reviewer for their new question about possible confusion between unitary nodes and “dim specks,” which may be Blt1 oligomers. These dim specks are distributed throughout the cell including the middle part where cytokinesis nodes are located. As indicated by the reviewer, they did not meet our criteria for unitary nodes, because their fluorescence appeared in less than 3-4 Z slices and in less than a 7-pixel diameter circle. Stimulated by this question, we measured the total fluorescence intensities of samples of “dim specks” and calculated their contribution to the total intensity of broad band. After subtraction of the cytoplasmic background fluorescence, the average fluorescence of a dim speck was ~170 arbitrary units, which is just 7% of the average intensity of unitary nodes (~ 2300). We found about 100 dim specks in the cortex of broad band region so their total contribution is 17,000 AU, whereas the total fluorescence of the broad band after background subtraction is ~450,000 AU. Therefore, the dim specks contribute only 4% of the fluorescence and have a minimal impact on the calculated number of nodes.

Being spread throughout a cell, these dim specks do contribute to the cytoplasmic background, which we subtracted from the total fluorescence of the broad band as explained in the Methods section.

Reviewer #2:In the previous rounds, we pointed out various problems in data analysis and the limited impact of the work. Unfortunately, these were not substantially improved in this version. As detailed below, it is unclear whether the scaling of the node numbers with cell size is real. Insights into the mechanism of scaling or into the functional role of this phenomenon on cytokinesis are still missing. Please refer to slides in my 'problems.pdf', shared here: https://drive.google.com/file/d/1CdsqS3PgZNmfpZvgD_p4ZL6Z5UL1VZJf/view?usp=sharing.

We thank the reviewer for taking the time to write a five page review with 21 pages of supplemental figures. Our additional work and revisions of the paper in response to this critique have strengthened the revised paper. We respond to each of the reviewer’s questions. In several cases, we carried out new measurements or analyses before revising the paper. In other cases, editing the text sufficed. The reviewer’s concerns had merit, but after investigation and suitable revisions, none of them impacted the conclusions of our paper.

The reviewer also explains their belief that cytokinesis nodes are “amorphous aggregates of proteins” rather than discrete structures. However, the work in this paper relies on the well-supported, fundamental assumption that nodes are discrete structures with stoichiometric ratios of proteins. Unfortunately, disagreement on this fundamental feature of the system means that the reviewer is unlikely to be satisfied with our responses.

Issue 1. Methods used to measure the fluorescence intensity of nodes. The authors' response to the Referee 1 (second round, I was Referee 3) prompted me to look into the background and the method of measuring the fluorescence intensity of the 'unitary nodes' more carefully. The fluorescence intensity of a small node was measured by the intensity in a circular ROI of 12-pixel diameter on the sum stack of 7 slices (3 below and above the target node). The background was corrected by subtracting the mean of the equivalent ROIs arbitrary set near the cell tip on the same sum stack.

Concern about the regions used to measure background came up multiple times, so we aggregated the reviewer’s comments and respond to them collectively.

Issue 1A, arbitrary selection of region to measure background:

Rather than being arbitrary, we chose areas to measure the cytoplasmic background fluorescence near the tips of the cells to avoid large organelles like the nucleus that exclude Blt1 and highly fluorescent nodes tagged with Blt1.

Issue 1B, impact of nonuniform diameter of cells on background measurements: The fact that both the cell diameter and the cytoplasmic background are not uniform along the cell axis make the background correction highly unreliable. The cell diameter is not uniform along the cell axis, being narrower at the tips, which must be considered in correcting for background fluorescence. According to the trace of the cell periphery in "background_RoiSet.zip", the diameter of the cell near the cell tips, where the background was measured, is smaller (~2.7 µm) than that at the equator (~3.7 µm). A circular ROI of 12-pixel diameter in the sum stack of 7 slices corresponds to a cylinder of ~0.5 µm diameter and ~1.2 µm height. For a node at the top of the cell, the corresponding background volumes near the tips are largely outside of the cytoplasm (page 1 of 'problems.pdf Geometrical problem 1). This underestimates the background intensity. These effects are not negligible because the raw background intensity varies between 1000 to 2000 depending on the depth of the z-slices while the peak raw intensity of the nodes is 3000~4000 (page 5 of 'problems.pdf').” On the other hand, for a node on the midplane of the cell, the corresponding background volume is entirely inside the cytoplasm (page2 Geometrical problem 2). This overestimates the background intensity.From another part of the review: The intensity of the cylindrical ROI put on the empty space outside of the cell (z=1~7) is 770 +/- 30 (mean +/- SD) (page 6 of 'problems.pdf'). In the central 7 slices (z=23~29), the intensity of the background ROIs set by the authors (green) is 2010 +/- 360. This increases to 2740 +/- 480 closer to the equatorial (magenta, still outside of the broad equatorial band) and 3050 +/- 610 at the periphery of the nucleus (yellow). These are not negligible in comparison with the raw intensity of the nodes 4170 +/- 690 (min. 3280 ~ max. 5510).The authors reasoned usage of the ROIs near the cell tip as the regions without brighter spots. However, this is not an ideal choice because the intensity for background correction should be taken from the region with a similar level of the background. The authors' method will systematically underestimate the background signal. If the background was measured near the equator avoiding the cortical signals such as the ROIs in the perinuclear region (yellow on page 6 of 'problems.pdf'), we would get the WT node intensity, in the worst case, ~50% smaller than those reported in the current manuscript. Using the background ROIs near the cell tips also introduces unnecessary uncertainty that might vary depending on the cells and on the positioning of the ROIs.From another part of the review: “How stable the current results are against the selection of the background ROIs closer to the equator in the wild-type or at the same distance from the equator for inter-strain comparison?”

Response regarding the location chosen to measure background fluorescence: We thank the reviewer for this important question. We agree that measuring the background fluorescence at the tips of cells in the same plane as nodes in the middle of the cells is problematic because the diameters of the cells are smaller at the tips than in the middle. Therefore, we remeasured the fluorescence background intensities of our entire collection of cells in two regions flanking the broad band of nodes and used these new values to correct the intensity of each node, calculate the average intensity per node, and calculate the total number of nodes (Author response image 1). The background fluorescence was 9% higher next to the broad band than at the tips of the cells, but subtracting this higher value reduced the average intensities per node by only ~5% and the total intensity of the node broad band by only 13%. Therefore, the calculation of the total number of nodes from the ratio of these two values was nearly the same as when using the tips of the cell to measure the cytoplasmic background. We remade all of the relevant figures (main text Figure 2D, Figures 3G-I, Figure 5G, and Figure 6E). An example of X-Y-Z positions of background ROIs flanking broad band of nodes is also included in the Source Code1 file.

**Author response image 1. sa2fig1:** Comparison of two regions to measure cytoplasmic background fluorescence for correcting the fluorescence intensities of unitary nodes, the total fluorescence in the broad band, and the number of cytokinesis nodes. (A) Schematic diagram of cells to show the position of ROIs for measuring background at the cell tips (left) and adjacent to the node broad band (right). Blue circles are ROIs to measure the background for unitary node intensities and the black polygons are ROIs to measure the background for total node broad band intensities. The histograms show the distributions of unitary node fluorescence intensities in wild type cells obtained after subtracting the background measured at cell tips (left) and adjacent to the node broad band (right). Each distribution satisfies the Shapiro-Wilk normality test with P> 0.05. (B) Violin plots of the numbers of nodes calculated from the ratio of total broad band fluorescence to the average fluorescence of a unitary node when the background (Bg) ROIs were near cell tips or adjacent to node broad band. An unpaired t-test shows that these two distributions did not differ significantly with P = 0.3.

Issue 2. Variability of the cytoplasmic signal and measurement of the node intensity and the node number.Issue 2A, nonuniformity of the background fluorescence: The cytoplasmic background is not uniform along the cell axis, which must be considered in correcting for background fluorescence. Apart from the granularity, the cytoplasmic Blt1-GFP signals in the cell in "NodeBroadband.czi" marked with "background_RoiSet.zip" and "Node_RoiSet.zip" show significant variation along the cell axis. This can be demonstrated both by reslicing the cell volume and measuring a circular ROI at the center of the cell (page 3 of 'problems.pdf') or by line-profiling in the sum projection of the central 7 slices (z=23~29, page 4 of 'problems.pdf'). In addition to the large fluctuation due to the cytoplasmic spots, there is a clear trend that the cytoplasmic signal increases towards the cell equator.

We thank the reviewer for measuring the background fluorescence in a slice along the cell axis and showing that the intensity of a rectangular section of seven slices along the length of the cell varies between 15 and 30 units. The cytoplasmic fluorescence varies owing to organelles such as the nucleus and vacuoles that exclude Blt1. The reviewer’s graph clearly shows lower fluorescence in the volume occupied by the nucleus. The intensities of the cytoplasmic spots are actually low (see the response to issue 3) so they do not cause “large fluctuations.”

Author response image 2 shows that the background fluorescence is low relative to the fluorescence of nodes. For this analysis, we used the same sum projection image of the central 7 slices (z slices 23 to 29) as the reviewer used on page 4 of ‘problem.pdf’ file. First, we subtracted the intensity per pixel outside cells to deduct the contribution from the extracellular fluorescence. Then we selected two ROIs by drawing two lines with widths of 12 pixels (~0.5 µm): a blue line along the length of the cell in the middle of the cytoplasm; and a green line along the edge of the cell where the broad band of nodes is located (Author response image 2). Author response image 2 shows the corresponding line profiles.

**Author response image 2. sa2fig2:** Comparison of the fluorescence intensity of Blt1-mEGFP in the cytoplasmic background and the broad band. (A) SUM projected image of 7 central Z- slices (23-29). The yellow line outlines the boundary of the cell. Blue and green shaded areas are 12 pixels wide. (B) Line profiles of the fluorescence intensities of the (blue line) background and (green line) node broad band are both corrected for background outside the cell.

The average intensity per pixel of the cytoplasmic background along the length of the midsection of the cell is 14 AU with a SD of 5 AU, which is less than 27% of the average intensity per pixel of the region near the plasma membrane containing nodes (51 AU with a SD of 21). Our measurement of background fluorescence (blue curve, in Author response image 2) differed from the reviewer’s analysis in two ways: (1) They did not subtract the background outside cells from the image, so the background intensity is higher in their analysis; and (2) their ROI extended outside the tips of the cell, where the fluorescence is lower. Because of this low fluorescence at the cell tips, the cytoplasmic background fluorescence in the middle of the cell was relatively higher compared with the cell tips. Our measurements show that the modest variation in the cytoplasmic background (Author response image 2) has little impact on measurements of the node fluorescence or the calculation of the number of nodes.

Issue 2C, Exclusion of Blt1-mEGFP from the nucleus lowers the background in the middle of the cell. To make the situation more complicated, the intensity drops in the region corresponding to the nucleus around the equator. An additional twist is the drop of the cytoplasmic signal at the equator due to exclusion by the nucleus. Although this effect might be less significant than the cell size-dependent drop of the background measured near the cell tips, its influence on different measurements should be checked. The same issue came up in item 2.3 “Check the influence of the drop of Blt1-GFP signal at the nucleus on various measurements.”

We thank the reviewer for pointing out that the nucleus lowers the total fluorescence intensity of the broad band, because it excludes Blt1-mEGFP. We used an alternative method to measure the impact of the nucleus on the total intensity of broad band after background subtraction (Author response image 3).

**Author response image 3. sa2fig3:** Measuring the total broad band fluorescence intensity with hollow cylinders. We measured the total fluorescence of a hollow cylinder with a wall thickness extending from the cell surface to the surface of the nucleus and a height equal to the width of the broad band. First, we SUM projected the region of the cell containing nodes in the X-direction (red arrow in panel B) and subtracted the total intensity of the cytoplasm inside the inner white dotted line in panel C. The remaining disk has the total intensity of the node broad band but without background correction (t). To calculate the background, we used the region of the cell near the node broad band shown by the purple disk and average projected it in the direction of the purple arrow shown in panel B to obtain the image in panel A. We calculated the intensity per pixel (i) within the two dotted lines in panel A, multiplied it by the total number of pixels (n) in the hollow cylinder of the node broad band, and subtracted it from the total intensity of broad band (t) to obtain the total intensity of broad band without contributions from the background and nucleus. T = t – (I X n).

The value of total broad band fluorescence after background subtraction (T = 399,552) was similar to that calculated (358,735) by subtracting the background adjacent to broad band from the total fluorescence of the broad band. Therefore, the lower background due to the presence of the nucleus resulted in an overcorrection of 40k AU. This is equivalent to 17 nodes, a small difference of 9 % compared to 190 nodes in these wild type cells.

Issue 2.1A: What is the variety of the background in the same strain and across the strains?

We used the method in Author response image 1 to measure the cytoplasmic background at the cell tips and adjacent to the broad band in WT and *cdc25-22* mutant cells (Author response image 4).

**Author response image 4. sa2fig4:** Violin plots comparing the background fluorescence in WT and *cdc25-22* mutant cells at the cell tips and adjacent to the broad band (BB).

The background fluorescent intensity adjacent to the broad band in WT cells was significantly higher than the background fluorescent intensity at the tips of WT cells or at both locations in *cdc25-22* mutant cells (paired/unpaired t-test, p < 0.008). The background fluorescent intensity was similar at both locations in *cdc25-22* mutant cells (paired t-test, P = 0.27). Nevertheless, this small difference in the cytoplasmic background has no impact on the measurement of node numbers, since these background values are subtracted from both the fluorescence intensities of individual nodes and the total fluorescence intensity of the broad band.

Issue 2.1B, Provide the raw data.

We are happy to share our data, but as the reviewer should understand, we are not allowed another round of review.

Issue 2.1C. Possible differences in background fluorescence within and between cells from the various strains. This problem becomes more serious when we compare the strains with different cell sizes. Consistent with the above observation, Blt1-mCherry in Figure 4B outside the equatorial band shows a rather constant decrease towards the poles. The gradient seems to be common across different strains (page 7 of ‘problems.pdf’). This suggests that the bigger the cell size, the lower the background signal measured near the poles. This would result in a larger over-estimation of the total intensity of Blt1-GFP in the broad equatorial band in larger cells. Dividing this with the intensity per node, which is largely constant across the strains (Figure 3G), would result in the size-dependent over-estimation of the node number. In other words, the scaling of the node number might just be an artifact of the cell size-dependent under-estimation of the background.

The reviewer’s speculation that “the scaling of the node number might just be an artifact of the cell size-dependent under-estimation of the background” is not true. Figure R4 above shows that the background fluorescence is only slightly lower in the largest cells in our sample (the *cdc25-22 strain)* than in wild type cells. Furthermore, the background is similar near the tips and the centers of both strains. We subtracted the background values for each strain from the fluorescence intensities of both the whole broad band and individual nodes, so they were taken into account when calculating the number of nodes from the ratio of these two fluorescence values.

On the other hand, the experiment in Figure 4B in the paper addresses other matters and does not provide information about the cytoplasmic background fluorescence. The graphs in Figure 4B show the distributions of both Pom1 and Blt1 along the length of five strains of cells. As explained clearly in the figure legend, for each cell, we measured a line profile the width of the cell. We aligned the scans of 10-12 cells of each strain at the centers of the cells and calculated the average intensities and SDs. The measured intensities fall off away from the cell center because the cell tips are hemispheres rather than cylinders and the cells are narrower at the ends. Therefore, these scans do not reflect the background intensities*.* To better illustrate the variability, we replaced the average values with the raw data in the main text Figure 4B and reworded lines #776-783.

Issue 3. Dimmer spots on the cell surfaceIssue 3A: Criteria for the 'unitary node' by the authors are "To qualify as a unitary node, a spot had to satisfy two conditions: (1) the fluorescence of a spot was present within a circle 12 pixels (~0.5 μm) in diameter; and (2) the fluorescence of a spot was present in 7 or fewer consecutive Z-slices with a peak in the middle slice and diminishing towards both sides." However, these are not sufficiently specific for unanimously identifying the 'unitary nodes'.

We thank the reviewer for asking about the dimmer spots (called “dim specks” by reviewer 1). These dim specks differ in three ways from unitary nodes. (1) Their fluorescence intensities are much lower than unitary nodes. We measured the average total fluorescence of a sample of 50 of these dim specks to be ~170 AU, which is roughly 15 times less than the average intensity of unitary nodes (~2300). (2) The distribution of the fluorescence from these dim specks was less than 7 pixels in diameter and confined to 3-4 Z slices. (3) Rather than being concentrated on the inner surface of the plasma membrane in the middle of cells like unitary nodes, dim specks are dispersed in the cytoplasm, so they were included in the background subtraction.

To evaluate the contribution of dim specks to the measurement of node numbers, we counted the dim spots in the volume of the broad band of wild type cells and found about 100. Thus they contribute about 17,000 AU of fluorescence to the total of the broad band compared with about 450,000 AU of fluorescence from 190 unitary nodes with an average fluorescence of 2300 AU. Thus the fluorescence of the dim spots is equal to the fluorescence of about 6 unitary nodes, which is 4% of the total 190 nodes in wild type cells. Therefore, the impact of these dim specks on our counts of cytokinesis nodes is negligible.

Near the spots hand-picked as 'unitary nodes' by the authors (or at an equivalent cortical position), there are spots that are significantly dimmer but satisfy these criteria (pages 8 to 13 of 'problems.pdf', 'Dimmer cortical spots, example' 1~6). The majority of them are brighter than the background signals (at least those in the volumes defined by the authors near the cell tips) (pages 14 and 15 of 'problems.pdf').

The reviewer selected and commented on eight spots in the file named ‘darker_spots.zip’. Our interpretation of the structures in some of these ROIs differs from the interpretation by the reviewer. The following explains why we did not choose some of them as unitary nodes. Page numbers refer to the ‘problems.pdf’ file provided by the reviewer.

Spot 1 looked like two spots together in both the X-Y and X-Z-slices, so we did not count it as a node. Page #8, center image.

Spot 2 is very small and the fluorescence is limited to 2-3 Z-slices, so we did not count it as a node. Page #8.

Spot 3 is very small and the fluorescence is limited to 2-3 Z-slices, so we did not count it as a node. Page #8

Spot 4 can be considered as a node but its fluorescence is extended in the Z slice in #28. Page #11 (spot 4, last image).

Spot 5 is okay to consider as a unitary node.

Spot 6 is small and the fluorescence is limited to 2-3 Z-slices, so we did not count it as a node. Page #9.

Spot 7 is very small and the fluorescence is limited to 2-3 Z-slices, so we did not count it as a node. Page #12.

Spot 8 is very small and the fluorescence is limited to 2-3 Z-slices, so we did not count it as a node. Page #10.

We have missed the Spot 5 for inclusion in the distribution of unitary nodes, but it was included in the total intensity of node broad band to estimate the number of unitary nodes. Please note that we measured more than 20 cells from the wild type strain to get average unitary node intensities of more than 200 nodes and the X-Y-Z positions (ROIs) provided to the reviewer with a Z-stack image of a WT cell were a representative sample of isolated unitary nodes that fulfilled our criteria.

Issue 3.1 Why were they (dim spots) excluded from the node measurement? :

Spots satisfying our conditions are included in the total broad band intensity, but the dim specks did not meet our criteria for unitary nodes, because their fluorescence appeared in less than 7-pixel diameter circle and in less than 3-4 Z slices. Furthermore, the dim specks were spread throughout the cell rather than being confined to the middle of the cell where nodes were present.

Issue 3.2 Analysis in Figure 2 should be done, including these dimmer spots on the cell surface.

The dim spots are everywhere in the cytoplasm, not confined to the cell surface like cytokinesis nodes. As explained above they did not contribute significantly to the total intensity of broad band or impact the total number of nodes, so we did not include them in the Analysis in Figure 2.

Issues 3.3 and 3.4: 3.3, Whether the node counting based on the intensity of 'unitary node' is still valid needs to be reconsidered. 3.4 If the authors insist on excluding the dimmer spots, more strict (but reasonable and not arbitrary) criteria for the 'unitary nodes' should be provided (although, to be honest, I doubt the concept of 'unitary node', see below).

Our new analysis (see the response to issue 3A) shows that the contribution of the dim spots to the total fluorescence intensity of the broad band is negligible and therefore has only a small impact on the calculated numbers of nodes*.*

Issue 4. Statistical test for normal distributionThe D'Agostino-Pearson normality test uses the skewness and kurtosis of the distribution (https://en.wikipedia.org/wiki/D%27Agostino%27s_K-squared_test). It is not suitable to examine whether the samples are from a normal distribution or a sum of multiple normal distributions. As shown on page 16 of 'problems.pdf', with the simulated samples from combined two normal distributions (not normal as a whole), the D'Agostino-Pearson test fails to reject the null hypothesis of normality while the Shapiro-Wilk test properly detected the deviation from the normality and rejected the null hypothesis (R script in page 17 of 'problems.pdf').The Shapiro-Wilk test and the Kolmogorov-Smirnov test were explicitly suggested in my first review (based on the above simulation, the Shapiro-Wilk test seems to be more appropriate for our case). I suspect that the authors might have obtained p-values < 0.05 with these tests, indicating that the distribution is not likely normal. This would provide another line of reason to doubt the existence of the 'unitary node'.

We reanalyzed our entire data using background ROIs located adjacent to node broad band instead at cell tips as suggested by the reviewer. The Shapiro-Wilk test showed that the data in text Figure 2D do not have a normal distribution. However, we know that the sample includes the fluorescence from both unitary nodes and a small number (< 2%) of unresolved pairs of closely spaced nodes. The peak in the distribution of fluorescence intensities of unitary nodes was ~2300, so the peak fluorescence from unresolved pairs of nodes should be ~4600. To see if the fluorescence intensities of unitary nodes are normally distributed, we considered the fluorescence ≤4600 and replotted the histogram. This distribution satisfies the Shapiro-Wilk test with P = 0.1. Please refer to Figure R1 above.

Issue 4.1 Provide the raw data behind Figure 2D and 2E including both the raw node intensity and the background measurements so that other people can examine them

As the reviewer should understand, we are not allowed another round of review.

Issue 5. Distributions of fluorescence intensities of small nodes in mutant strains Although "The distributions of intensities Blt1 fluorescence of cytokinesis nodes were similar (around 2500 AU) in wild-type cells (Figure. 2C) and three of the experimental strains (Figure. 3G)." (line 292-4), none of the histograms in Figure 3G look like a normal distribution.Issue 5.1 Why aren't they normally distributed?

The distributions of intensities of Blt1 fluorescence of cytokinesis nodes of all strains obtained in the new analysis with the background ROIs adjacent to node broad band satisfied the Shapiro-Wilk normality test. Please see new text Figures 2D and 3G and their legends.

Issue 5.2 What would they look like if the dimmer spots on the cell surface were included?

The negligible fluorescence of the dim spots in WT cells explained above did not make a substantial contribution to the total intensity of the broad band. Therefore, the dim spots are unlikely to change the node counts in the other strains.

Issue 5.3 Provide the raw data behind Figure 3G-K so that other people can examine them.

We are not allowed another round of review, as the reviewer should understand.

Issue 6. Unitarity of the 'unitary nodes'The spots on the cell surface dimmer than the 'unitary nodes' hand-picked by the authors were indeed observed also in the FPALM image in Figure S2 of Laplante (2017). There were smaller clusters of emissions marked by arrows (pages 18 of 'problems.pdf'), which were excluded from the analysis without any reasoning.

The reviewer is correct that the FPALM image in Figure S2 of Laplante (2016) has not only unitary cytokinesis nodes but also some smaller clusters of localizations. Laplante et al. did not analyze the smallest clusters of localizations highlighted by the reviewer, because they are distributed across the entire cell surface rather than being confined to the broad band of cytokinesis nodes in the center of the cell. Furthermore, the methods section of Laplante et al. 2016 explained that they only analyzed nodes in the relatively flat area at the top and bottom of surfaces or in the midplane of the cylindrical cells to avoid artifacts associated with the curved surface of the cells. Some of the reviewer’s arrows point to small spots on the curved surface of the cell, which would never be used for quantitative purposes.

This is another comment on Laplante (2016): Moreover, even within the data of the clusters that the authors believed to be 'unitary nodes', there is a clear sign of heterogeneity. The distributions of the numbers of emissions per cluster show a long tail, which contains ~30% or more clusters. This indicates that, in addition to the dimmer clusters that were not counted as 'nodes', there exist significant heterogeneities even among the hand-picked 'nodes'. It is not clear what "Nodes in the tail were seen only in long reconstructions (25 s), not in short reconstructions (5 s)." (page E5880 Laplante (2016)) means without the real data for the short reconstructions.

Do not confuse the tails in our Figure 2D with the tails in the counts of localized emitters in Figure S6 of Laplante et al. (2016). Taking into account the principles that govern single-molecule localization microscopy (SMLM), stochasticity in the emissions of the fluorescent probe is most likely responsible for the tails in Figure S6. The stochastic nature of the fluorophore emission and the fact that the distributions are necessarily bound at zero means that some nodes will have more localizations than other manually selected nodes.

This is a criticism on Akamatsu et al. (2017): The concept of 'unitary node' might be supported if the brighter spots were able to be explained as integer multiples of unit intensity. In Akamatsu (2017), the authors argued that the distributions of fluoresce intensity of the interface (sic “interphase”) nodes can be better fitted by multiple Gaussian distributions than by continuous log-normal distributions. However, the fitting was done by setting the amplitudes, SDs, and means for each of the Gaussian distributions as free parameters (9 or 12 parameters for 3 or 4 Gaussians, respectively). This is simply wrong since there must be restrictions both on the means and SDs. The mean and SD of the n'th peak (m_n and s_n, respectively) should follow m_n = m_1 x n and sd_n = sd_1 x square root of n. A simple comparison of the goodness of fit with the other distribution that has only a limited degree of freedom (the log-normal distribution has only 2 parameters) is non-sense. The comparison between the statistical models should have been done based on an appropriate estimator such as the Akaike information criterion (AIC) (https://en.wikipedia.org/wiki/Akaike_information_criterion). With a simulation https://www.dropbox.com/s/suti2gz9ovgmfyw/histogram.pdf?dl=0, Referee 2 raised a valid example of artificial appearance of multimodality from a log-normal distribution.

Simple visual inspection of Figure 2 in Akamatsu et al. (2017) reveals that the distributions of intensities of nodes marked with various proteins all had a major low intensity peak and a range of nodes with higher intensities including peaks with intensities that were multiples of the low intensity peak. Akamatsu considered various ways of fitting this data and used the default starting points and constraints section of the Matlab package, where the only necessary constraint is that the value used must be greater than 0. See the information at the following link:

https://www.mathworks.com/help/curvefit/parametric-fitting.html#bszh0sy-13. The reviewer suggests alternative fitting methods, which may be better, but they do not alter the obvious conclusion from the raw data that the observed intensities of nodes are multiples of the low intensity peak, so this criticism of a published paper should not be used to argue against the publication of the paper under review.

Considering all the points discussed so far, i.e.,– Unexplained exclusion of smaller clusters in the analysis of FPALM data (Laplante (2017))

This is refuted above.

– The long tail towards the bigger clusters even among the selected '(unitary) nodes' (Laplante (2017))

This is explained above.

– Inappropriate statistical analysis of the distribution of the intensity of the nodes with a long tail (Akamatsu (2017)).

The statistical analysis of the data in the published paper is not relevant to the work in this paper, as explained above.

– The intensity of the spots picked by the authors as 'unitary nodes' from the Airyscan images doesn't seem to follow a normal distribution (Figure 2D and Figure 3G),

This is explained above.

The evidence for the 'unitary node' is pretty weak. We can't exclude a possibility that there is no such thing as 'unitary node' but the nodes are rather amorphous aggregates of proteins as illustrated on page 20 of 'problems.pdf'. A critical re-examination of the author's own past data must be done.

This reviewer simply does not believe in unitary nodes, so they have looked for every possible inconsistency that supports their belief that they are amorphous aggregates while ignoring the very strong data that support the concept of unitary nodes with stoichiometric ratios of proteins.

To support their belief that nodes are amorphous aggregates, the reviewer proposed an entirely imaginary hypothesis that nodes are composed of a wide range of stoichiometries of a fundamental unit (yellow dot) to give nodes with a continuous range of molecules (page 20 of their supplemental materials). This diagram is inconsistent with the evidence of Laplante et al. (2016) and newer evidence, that is inconsistent with the amorphous aggregate hypothesis:

1. Independently-measured numbers of localizations of several node components give stoichiometric ratios of these protein molecules rather than continuous distributions (Bellingham-Johnstun et al., 2021; Laplante et al., 2016).

2. Splitting the N-Myo2 data wouldn’t have given the same CDF (Laplante et al., 2016).

3. Bellingham-Johnstun et al. (Bellingham-Johnstun et al., 2021) measured the same radii for the localizations of Myo2 heads as Laplante et al. (2016).

4. There would be no predictive power between the constriction rate and the radius of the N-Myo2 (Bellingham-Johnstun et al. 2021).

5. Other arguments include the stoichiometric ratio of node components measured by quantitative confocal microscopy.

Most importantly, the reviewer’s diagram implies that the number of emitters per node determines the size and shape of the node. On the contrary, the data of Laplante et al. (2016) show that the radius and RDD are independent of the number of emitters per node. Datasets with high and low numbers of emitters give the same calculated radii and radial density distributions. This can be appreciated by comparing the radii and numbers of localizations of Cdc12-mEos3.2 and mEos3.2-Cdc15 in nodes (Laplante et al. 2016, Figure 2A). The radii calculated from this data (27 vs 26 nm) are the same even though the mEos3.2-Cdc15 nodes have twice the numbers of localizations as the Cdc12-mEos3.2 nodes.

To appreciate the power of SMLM one must understand the principles of the method. SMLM provides the numbers and positions of single molecule emissions over time. From the numbers of localizations at a particular physical location, one can calculate the radial density distribution of the emitters (distribution of the sources of the emissions within the protein cluster) and the radius of the protein cluster (the size of the protein cluster). These measurements give estimates of the sizes and probabilities of the emitter distributions within these zones. Laplante et al. (2016) reported these distributions are remarkably accurate by showing that they give the right physical distance (165 nm) between mEos3.2 on the two ends of the myosin-II molecule.

Issue 7. Nature of the dotty cytoplasmic signalThe authors responded, "The processing of Airyscan images increases the contrast of these dim inhomogeneities". It remains as a black box what image processing was done as the only information provided is "To increase signal-to-noise ratio and resolution, acquired images were processed by 3D Airyscan filter strength 7.0 with Zen Black software."7.1 Clarify the principle of the filter with an appropriate reference and specify the parameter settings (Is 'strength 7.0' sufficient? What does this mean?) :

Airyscan uses an array of 32 channel detectors each with the size of 0.2 Airy Units. The ZEN Black software applies deconvolution to each individual image collected by each detector element. Once each image has been deconvolved separately, a sum image is reconstructed using weighting to acknowledge the positional information and unique optical transfer function (OTF) of each detector. This process improves the resolution in all three spatial directions by a factor of 1.7. The processing includes a Wiener filter deconvolution with options of either 2D or 3D, which are collectively called the Airyscan Filter strength (AF) (Korobchevskaya et al., 2017).

We used AF = 7 to reconstruct Airyscan images based on the improved resolution while limiting artifacts seen in the images using higher strength AF = 8 or AF = 9. Author response image 5 has images of 0.170 µm green fluorescence beads processed with a range of Airyscan filtering strengths to illustrate the best value to be used to reconstruct fission yeast cells images. Please see Korobchevskaya et al. (Korobchevskaya et al., 2017) for details.

**Author response image 5. sa2fig5:** Comparison of different Airyscan filtering strengths. (A) Airyscan images of two 0.17 µm green fluorescence beads processed with a range of Airyscan filtering strengths. (B) Corresponding line profiles along the white dotted lines in each micrograph in panel (A). The images obtained using AF >7 show artifacts.

7.2 Does this processing preserve the linearity of the fluorescence signal?

As the processing first works on the individual images obtained in 32 channel detectors and then sums them according to the position of the detectors it is expected to preserve the linearity of the fluorescence signal. The details can be found here:

(https://asset-downloads.zeiss.com/catalogs/download/mic/104cc06d-f997-4cc0-b679-b2cf93bfc863/EN_wp_LSM-880_Basic-Principle-Airyscan.pdf)

7.3 Why are the grayscale levels so small? Only a range 0 to <~30 in 0-65535 grayscale levels was used.

The limited range of grayscale values resulted from using a low excitation intensity to limit photobleaching and a small pinhole of ~ 0.2 Airy unit for each channel on the Airyscan detector to reject most of the out-of-focus light.

Reviewer #3:The revised manuscript demonstrates a correlation between cell size and the number of cytokinetic node precursor structures. This conclusion supports and extends previous studies in the field. The strengths of the work are balanced by some limitations that have been expressed in earlier reviews and remain largely unresolved:(1) The overall impact of the work is limited because it reaches the same conclusion as previous work. As the authors have pointed out, their current work moves beyond previous studies in two ways: (a) they examine these node structures during cytokinesis whereas previous studies examined them during interphase using different markers; and (b) authors have used cell size mutants to extend the range of sizes examined. However, the overall conclusion is the same as earlier work. It is also worth noting that several of the size mutants (wee1-50, rga4D, and cdc25-22) do not follow similar size-scaling properties, but these potentially interesting differences remain unexamined.

We would like to emphasize again that previous studies considered interphase nodes, while we investigated cytokinesis nodes that have different compositions (including proteins that will form the contractile ring) than interphase nodes. It is interesting that cell size influences the numbers of both types of nodes, but our work is on different nodes at a different time in the cell cycle rather than being simply confirmatory of the previous work on interphase nodes.

(2) Essential to address The major conclusions of the paper rely on the number of cytokinetic nodes present in different cells, but the authors have not actually counted the number of nodes. Their methods for estimating the number of nodes rely on several tenuous assumptions and have raised concerns from all of the reviewers at each stage of review.

Our assumptions are on much stronger ground than suggested by the reviewer’s use of ’tenuous’. The extensive critiques from Reviewer # 2 may have raised questions in the mind of reviewer #3. Although reviewer #2 raised some plausible concerns and invested much time analyzing our data in the current and previous rounds of review, our responses to reviewer #2 given above explain that none of these issues actually had a substantial impact on the interpretation and conclusions of our experiments. We encourage the reviewer to read our responses to reviewer #2.

It is not definitively known if Blt1 nodes are all the same size (referred to as 'unitary nodes' in the paper).

Previous work (Akamatsu et al., 2017; Laplante et al., 2016) presented multiple lines of evidence that cytokinesis nodes are regular structures with stoichiometric ratios of proteins. Additional evidence for a uniform population is in Figure 2D of the revised manuscript, which shows that the fluorescence intensities of individual nodes resolved by Airyscan microscopy have a normal distribution. We hope that counting the numbers of molecules in individual nodes using single molecule localization microscopy will provide a definitive answer, but we still working on that separate project, which is well beyond the scope of the current study.

It is also not known if all plasma membrane-bound Blt1 in the cell middle is contained in nodes, or alternatively if there is membrane-bound Blt1 diffusively localized in this region outside of nodes. Both 'unknowns' leave me concerned about the accuracy of the numbers presented.

Other than the low level of diffuse background fluorescence, the only other Blt1-mEGFP fluorescence near the plasma membrane comes from dim spots/specks, which are also distributed throughout the cytoplasm. New measurements made in response to reviewers #1 and #2 during this round of review showed that the fluorescence from the dim spots/specks in the broad band region of the cells is only 4% of the total fluorescence from the nodes. Since these dim spots are spread throughout the cytoplasm, they were included in the background fluorescence and subtracted from the total intensity of the node broad band.

I would like to propose a potential solution to this problem. The authors discuss the improved ability of airyscan imaging to resolve unitary nodes within the broad band. They could use their high-resolution images to count Blt1 nodes from the same images, and then present these numbers as a 'proof-of-principle' that node number scales with cell size by 2 separate analyses (counting versus their current estimations). The counting method would underestimate due to areas with multiple, unresolved unitary nodes; but similar scaling should be observed. It would not be necessary to reanalyze all strains/images this way, but perhaps just wild type and some cdc25-22 cells to establish a similar scaling trend by both methods. The fluorescence intensity of a small node was measured by the intensity in a circular ROI of 12-pixel diameter on the sum stack of 7 slices.

We thank the reviewer for this constructive suggestion and made the proposed measurements. We counted directly the nodes resolved in 3D reconstructions of Airyscan images and satisfying our two conditions for unitary nodes, their fluorescence is in a 12-pixel diameter circle and spread within 7 Z-slices. The numbers were higher in *cdc25-22* cells than wild type cells (Author response image 6). In both strains, the numbers of nodes were about one-fourth of those measured by dividing total broad band fluorescence by the fluorescence of a unitary node. Individual nodes were not resolved in larger fluorescent spots and were not counted. Thus, the reviewer’s experiment illustrates the problem that we solved by counting nodes as the ratio of total broad band fluorescence (corrected for background) to the average fluorescence of a unitary node.

**Author response image 6. sa2fig6:** The number of nodes in the broad bands of two strains with different volumes: (black) wild type (n = 11); and (orange) *cdc25-22* mutant cells (n = 8). Nodes were counted by two methods: (circles) calculation from the ratio of total broad band Blt1-mEGFP fluorescence (corrected for background) to the average fluorescence intensity of a unitary node (corrected for background) as explained in the Methods section; and (triangles) direct counting of unitary nodes resolved in 3D reconstructions of Airyscan images. Vertical and horizontal bars are ±1 SD. The vertical SD is smaller (5) than the data point for the direct count of WT cells.

References:

Akamatsu, M., Lin, Y., Bewersdorf, J., and Pollard, T. D. (2017). Analysis of interphase node proteins in fission yeast by quantitative and superresolution fluorescence microscopy. *Molecular Biology of the Cell*, *28*(23), 3203–3214. https://doi.org/10.1091/mbc.e16-07-0522

Bellingham-Johnstun, K., Anders, E. C., Ravi, J., Bruinsma, C., and Laplante, C. (2021). Molecular organization of cytokinesis node predicts the constriction rate of the contractile ring. *Journal of Cell Biology*, *220*(3). https://doi.org/10.1083/JCB.202008032

Korobchevskaya, K., Lagerholm, B. C., Colin-York, H., and Fritzsche, M. (2017). Exploring the potential of Airyscan microscopy for live cell imaging. *Photonics*, *4*(3). https://doi.org/10.3390/photonics4030041

Laplante, C., Huang, F., Tebbs, I. R., Bewersdorf, J., and Pollard, T. D. (2016). Molecular organization of cytokinesis nodes and contractile rings by super-resolution fluorescence microscopy of live fission yeast. Proceedings of the National Academy of Sciences of the United States of America, 113(40), E5876–E5885. https://doi.org/10.1073/pnas.1608252113

Pan, K. Z., Saunders, T. E., Flor-Parra, I., Howard, M., and Chang, F. (2014). Cortical regulation of cell size by a sizer cdr2p. *ELife*, *2014*(3), 1–24. https://doi.org/10.7554/*eLife*.02040

[Editors’ note: further revisions were suggested prior to acceptance, as described below.]

In this version, there are two different Figure 2D histograms, one on page 9 and the other on page 55. They are different in bins centered on 3000 and 3500 (arrows in page 3 of problems2.pdf) in addition to those centered on 5000, 5500, 6000, and 6500. This difference must influence the Gaussian fitting with the nodes whose intensity was less than 4600 A.U. However, the two curves of Gaussian fitting seem to be identical (both annotated as "2300{plus minus}900"). Something is wrong here.Some figures have multiple versions within the current manuscript. Two embedded within the text seem to be one from the previous versions and the new version. Confusingly, this new version in the text is different from the high-resolution version at the end of the manuscript. Even more confusingly, in some cases, the new one in the text seems to be the one consistent with the text and the legend (Figure 2E) while the high-resolution one seems to be more appropriate in other cases (Figure 2D).

The editors requested the revised manuscript with “track changes,” so it shows both the deleted figure from the previous version and the revised figure. For example, the Figure 2D histogram on page 9 is the previous version shown as deleted by “track changes” and the revised figure is on page 10. We also provided high-resolution figures at the end of the manuscript. We apologize for uploading the wrong revised Figure 2 in the text and in high resolution. We also noticed that the data in Figure 3H–K shifted during the conversion from.TIF to.pdf. We corrected both problems.

2. Variability of the cytoplasmic signal and measurement of the node intensity and the node number.The reviewer pointed out the difficulties in handling the background signal (1) large fluctuation and (2) systematic decrease towards the cell tips. The regions closer to the equatorial zone used in this version are better than the regions near tips, addressing the point 2 to some extent although, as was previously mentioned, the cytoplasmic signals are even higher in the perinuclear region next to the equatorial cortex.

In the most recent submission, we followed the reviewer’s advice and reanalyzed the data using an area near the broad band of nodes to measure the cytoplasmic background instead of the cell tips. Our previous response to this Issue explained that the perinuclear region is inappropriate to measure cytoplasmic background due to the presence of high-intensity nodes as well as the nucleus.

To draw a conclusion that the fluctuation is small, the line scans were shown in Figure R2. The authors claimed that the signal along the cell edge (green) containing the nodes is much higher (51{plus minus}21 AU, mean{plus minus}sd) than the cytoplasm (14{plus minus}5 AU). This itself is correct. However, this doesn't support that the background fluctuation is small relative to the intensity of the 'unitary nodes' since the major peaks (60~90 AU) on the green profile in Figure R2B correspond to the clusters of the many 'unitary nodes' (please refer to page1 of problems2.pdf). The spot measured as a 'unitary node' (ROI 0025-0270-0044, arrow) corresponds to a minor peak of ~30 AU, which is comparable to the fluctuations of the blue profile. They brought (up) the intensity of the clusters of 'unitary node' when discussing the intensity of individual 'unitary nodes' relative to the background fluctuation.

The plots in the previous response are line profiles, average intensities per pixel across lines 12 pixels wide plotted against the length of the cell. Since it is an average intensity per pixel the numbers are in tens (15–90 A.U.) and the minor peak of ~30 A.U. corresponding to a unitary node (single node) referred by the reviewer on page1 of problems2.pdf is an average intensity of a pixel in lines across the node and not the intensity of the whole node at (ROI 0025-0270-0044).

The fluctuation in the average intensity of a line profile depends upon the width of the line (the number of parallel pixels averaged), so averaging more pixels across wider lines will smooth out the fluctuations. Therefore, to emphasize the background fluctuations we SUM projected Z slices 22–28 and used lines one pixel wide to measure the intensities at one-pixel intervals of the cytoplasmic background in the middle of the cell and of the broad band of nodes along the long axis of the cell (Author response image 7).

**Author response image 7. sa2fig7:** Comparison of the fluorescence intensity of Blt1-mEGFP in the cytoplasmic background and along the one edge of the cell containing a single node and clusters of nodes. (A) SUM projected image of 7 central Z-slices (22–28). The white dotted line outlines the boundary of the cell. Blue and green lines are one pixel wide. (B) Line profiles of the fluorescence intensities of the (blue) cytoplasmic background and (green) the broad band of nodes, including a single node are both corrected for background outside the cell.

The spot measured as a 'unitary/single node' (ROI 0025-0270-0044, arrow) has a peak intensity of ~47 A.U. above the background average (14 A. U.), while the peaks of the fluctuations in the background above the average is ~25 A.U., so the intensity of the node is well above the fluctuations in the background.

Furthermore, the violin plots in Author response image 8 show that the average total intensity of 231 single nodes in wild-type cells (3568 A.U. without background correction; Node, green) is significantly higher by an unpaired t-test (p < 0.0001) than the average intensity (1238 A.U.; AvgBg, red) of the cytoplasmic background measured in equivalent volumes adjacent to the node broad bands.

**Author response image 8. sa2fig8:** Violin plots of the intensity of single nodes without background correction (Node, green) and the average cytoplasmic background (AvgBg).

The authors tried to explain the difference between their claim and the reviewer's analysis "Our measurement of background fluorescence (blue curve, in Figure R2B) differed from the reviewer's analysis in two ways: (1) They (the reviewer) did not subtract the background outside cells from the image, so the background intensity is higher in their analysis; and (2) their (the reviewer’s) ROI extended outside the tips of the cell, where the fluorescence is lower." In addition to the above issue, both the remarks miss the point. As to (1), firstly, subtraction of the common background (outside of the cell) doesn't influence the comparison between the peak of the 'unitary node' and the fluctuation of the cytoplasmic background.

We agree that subtraction of the common background (outside of the cell) does not influence the comparison of the peak of the 'unitary node' and the fluctuation of the cytoplasmic background. But our above remark (1) on the reviewer's analysis is true if we were to compare the background values with the node intensity.

In the previous review, when discussing the mean levels of the cytoplasmic background, they were standardized with the background outside the cell set to 0 and the mean node intensity set to 1 and pointed out that the cytoplasmic background levels vary 33% to 64% of the mean intensity of the 'unitary nodes' depending on the location with large spreads. As to (2), the line ROI was simply set from one end to another end of the outline of the cell defined by the authors, which seems to be overlapping with the other cell on the righthand side. "The background fluorescence was 9% higher next to the broad band than at the tips of the cells, but subtracting this higher value reduced the average intensities per node by only ~5% and the total intensity of the node broad band by only 13%." (page 4 of the rebuttal). This is quite different from what the reviewer measured and presented to the authors in the previous round. One would need to look at the raw data.

The reviewer measured the background in a SUM projected image of Z-slices 23–29 (page 6, problems.pdf file). This would be appropriate for subtracting the background from nodes located in Zslice 26, but we did not measure nodes in slice 26. The manuscript explains that we measured the fluorescence of nodes and the background in the same SUM projected image of 7 Z slices. Our previous response to *Issue 2.1A* included a graph comparing the background values in wild-type cells measured at both the cell tips and adjacent to the node broad band. We attached to this response the numerical data from all the wild-type cells that we analyzed.

Was the image data of a cell presented by the authors an exception?

The image data of the cell presented by the authors is not an exception.

The numbers of the various cell types analyzed for this version are different from those in the previous versions (WT: 24 reduced to 21, wee1:24 to18, rga2: 23 to ?, cdc25: 32 to 31, rga4: 18 to 17).From another part of the review (page 3): Different Figure 2E versions on page 12 and page 55. Sample size is different from the previous version.From another part of the review. Issue 4, the reason for the change in the number of cells needs to be clarified.From another part of the review. However, the number of the cells analyzed is quite different from the previous versions (24 cells to 18 cells in an extreme case). Why were some data silently omitted? Can we get the same results with the original set of cells?From another part of the review (page 4): Figure 3G. Drastically different sample sizes from the previous versions.

We apologize for not explaining clearly the change in the number of cells examined in the last submission. For that revision, we reanalyzed the fluorescence intensities of nodes in the same cells using a region near the broad band of nodes to correct for background fluorescence, as recommended by the reviewer, rather than the cell tips. Our two previous criteria for single nodes were “fluorescence had to be confined in the X-Y plane within a circle 12-pixels (0.5 µm) in diameter and in the Z direction to 7 zslices of 170 nm (~1.19 µm).” During the reanalysis for the most recent submission, we excluded dim spots with fluorescence in 3 or less consecutive Z-slices, spots with fluorescence in more than 7 z-slices, and ambiguous spots, as explained in *Issue 3* in the previous response to review. These new criteria eliminated 77 out of 312 nodes in WT cells, 73 out of 226 in *rga2Δ* cells, 3 out of 310 in *cdc25-22* cells, 46 out of 184 in *rga4Δ* cells, 42 out of 195 in *rga4Δ* + HU cells and 64 out of 255 in *wee1-50* cells, but changed the mean fluorescence very little.

The reanalysis for the most recent submission used the same images and cells as the original analysis in the first submission in July 2021. The cells were numbered consecutively on the dates they were imaged. The original analysis used all of the cells. The reanalysis in the most recent submission began with the first cell in each series and stopped after measuring nodes in enough cells to satisfy the reviewer’s concerns, omitting 3 of 24 WT cells, 3 of 23 *rga2Δ* cells, 1 of 32 *cdc25-22* cells, 1 of 18 *rga4Δ* cells, 0 of 18 *rga4Δ* + HU cells and 3 of 24 *wee1-50* cells*.* These omitted cells were random members of the data sets, so they should not bias the results.

4. Differences in the distributions in Figure 3G between the original and revised manuscripts.There was no mention or explanation of this significant change while the new look of the distributions in Figure 3G was attributed to the change in the background correction.

As shown on page 4 of problems3.pdf, the distributions of node intensities in Figure 3G differ in the most recent revision of the manuscript, because we used (as explained above) more stringent criteria for picking single nodes, we sampled somewhat fewer cells, and we changed the background correction method as recommended by the reviewer. Nevertheless, the values of the mean intensities were similar (ratio new/original = 1.03, 0.96, 1.0, 0.99, 0.98). These differences between the original analysis and the reanalysis for the most recent submission are small and do not affect the analysis of node numbers or the conclusions of the paper.

5. Statistical tests.(Figure R1B legend) "An unpaired t-test shows that these two distributions did not differ significantly with P = 0.3" This interpretation is simply wrong. T-test assumes that the two groups are normally distributed and its null hypothesis is that their means are equal. It does NOT examine whether the two groups follow the same statistical distribution. For example, t-test between two distributions, (A) a normal distribution with mean 15, standard deviation sqrt(8) (N=100), and (B) a superposition of two normal distributions with mean 10 and mean 20 (both with standard deviation 2, total N=100) almost always returns P>0.05. If we follow the authors' way of interpretation, this would mean that there is no difference between the distributions of A and B. However, this is a clear error.

We agree with the reviewer’s interpretation. Thank you. These statistical statements were worded correctly in the most recently submitted text but not in the previous response to review.

Issue of the shape of the distribution of node intensities in wild type cellsIssue 4 Statistical test for normal distribution. It was admitted that the proper test rejected that the distribution of the intensity of the ‘unitary nodes’ follows a Gaussian distribution unless an additional selection is introduced. This is consistent with the uncertainty and subjectiveness of the criteria for the ‘unitary nodes’ discussed above.Pages 1 and 2 Figure 2D: 2300{plus minus}900 is OK, but the curve on the histogram is different from this (2180{plus minus}980).Shapiro-Wilk test doesn't support the normal distribution of the node intensities (<4600).

This issue of non-normality was addressed in the second revision and accordingly the text in the manuscript was changed in the last submission. Please see lines #211–215. Consistent with the wellestablished difficulty of resolving closely spaced nodes, 2% of nodes in wild-type cells had intensities more than twice the peak value. Our reasonable assumption is that these high-intensity spots are two rather than one single nodes. This assumption, based on the biology, provides a reasonable rationale to exclude these high-intensity nodes from the test for the Gaussian distribution of the intensities of nodes.

We replotted the node intensities of wild-type cells as a histogram. To estimate the peak value of the data, we used the Solver add-in from the Analysis Toolpak of Excel to fit a Gaussian curve to all of the data by minimizing the chi-squared values and by adjusting the amplitude, position, and FWHM of the function. A small number of nodes with high intensities fell outside the curve, as expected for pairs of unresolved nodes. We used the same method to fit a Gaussian distribution to the data excluding values greater than twice the peak value (Figure 2D).

As noted by the reviewer, the p-value from a Shapiro-Wilk analysis rejected the null hypothesis of a normal distribution. On the other hand, the Shapiro-Wilk W test statistic (W) was 0.9854, which does not reject the null hypothesis of normality (King and Eckersley, 2019; Royston, 1992). The Shapiro-Wilk normality test is recommended for sample sizes less than 50 but is less reliable for samples like ours between 50 and 300, because it cannot adjust the standard error (Ghasemi and Zahediasl, 2012). Since we have a medium-sized sample of 231 values, we used descriptive statistics to calculate a Z-score to check for normality. A Z score is calculated by dividing the skewness value or excess kurtosis value by their standard errors. For medium-sized samples of 50–300, an absolute Z-value of ± 3.29 is consistent with a normal distribution (Kim, 2013). In our case, the ratio of the skewness (0.27) and the standard error ((6/n)^1/2^ = 0.16) gave a value of 1.6, which is <3.95, so the test does not reject the null hypothesis that the data has a normal distribution.

It is difficult to understand why the authors insist on the normal distribution. Akamatsu (2017) showed a distribution of Blt1 with a similar spread, which was fit with multiple Gaussian. Figure 2D can easily be fit with multiple Gaussians as well (page 2 of the Reviewer problems2.pdf)

The revised histogram in Figure R2B clearly has a single peak with a small tail of high values, as expected from the difficulty of resolving two closely-spaced nodes. As explained in earlier responses, the unitary node concept came from FPALM super-resolution data (Laplante et al., 2016). Akamatsu et al. (2017) probably recorded more multiple nodes by confocal microscopy, because the resolution is 1.7 times lower than the Airyscan microscope. The shape of our distribution of node intensities in wild-type cells does not justify fitting with multiple Gaussian distributions.

Issue of the shape of the distribution of node intensities in mutant cellsIssue 5 Distributions of fluorescence intensities of small nodes in mutant strains. The authors claim that the distributions in the current Figure 3G follow a normal distribution based on the Shapiro-Wilk test, attributing this change to the new analysis with the background ROIs adjacent to the node broad band.From another part of the review (page 4). Figure 3G. Shapiro-Wilk test doesn't support the normal distribution of the node intensities (<4600) (without extreme correction for multiple comparisons).From another part of the review (Issue 4). Does the original dataset pass the normality test after the same manipulation (cutting the long tail)?

We agree with the referee that the distributions of node intensities in mutant cells (excluding values larger than twice the peak value) failed the Shapiro-Wilk test for normality based on the p-values. However, the more appropriate Shapiro-Wilk W test for sample size >50 had W statistic values ≥ 0.97 for all the mutant strains, which does not reject the null hypothesis of normal distributions (King and Eckersley, 2019; Royston, 1992). Analysis of the mutant cell data by descriptive statistics gave Z scores <3.95, which did not reject the null hypothesis that the distributions are from normal distributions.

6. Dimmer spots on the cell surfaceA concern was raised about the criteria to spot the 'unitary nodes' by picking up (we do not understand “picking up” but will assume that the reviewer meant here “ignoring”) just slightly dimmer spots observed next to the 'unitary nodes' on the equatorial cortex, which were not counted by the authors. The authors responded to this by measuring the dim specks that are found everywhere and are easily distinguishable from the 'unitary nodes' (~170 AU, ~15 times dimmer). However, the key point was how one can define criteria that distinguish those just slightly dimmer spots the reviewer picked up from the authors' 'unitary nodes'. This was because, if it was difficult, (1) this would question the concept of the 'unitary node' and (2) this would affect the average intensity of the 'unitary nodes' and hence the node counts. The authors' argument using much dimmer cytoplasmic specks is out of focus. The authors' argument based on the minimal contribution of the dim specks to the overall signal in the broad band doesn't address these problems.They also made comments on the spots the reviewer picked up. 5 of 8 spots were rejected because "Spot x is small and the fluorescence is limited to 2–3 Z-slices" However, these are ad-hoc and highly subjective criteria as we still find in the Methods section "To qualify as a unitary node, a spot had to satisfy two conditions: (1) the fluorescence of a spot was present within a circle 12 pixels (~0.5 μm) in diameter; (2) the fluorescence of a spot was present in 7 or fewer consecutive Z-slices. with a peak in the middle slice and diminishing towards both sides.". Note the terminologies, "within" and "7 or fewer consecutive Z-slices".

We agree that our original criteria did not exclude dim specks, so when we reanalyzed the data for the most recent submission, we used the background correction recommended by the reviewer, and we also modified our second condition for inclusion as follows: (2) the fluorescence of a spot was present in more than 3 and up to 7 consecutive Z-slices, with a peak in the middle slice/s and diminishing towards both sides. We also modified lines 763–764 in the manuscript accordingly.

They also commented on Spot 1 "Spot 1 looked like two spots together in both the X-Y and X-Z-slices, so we did not count it as a node." However, this is also the case for the ROI 0016-0244-0082 on the same zslices, ROI 0018-0306-0104 and 0019-0249-0116, which have irregular, non-circular appearances (pages 8–10 of problem.pdf shared previously).

As required by the second condition for a spot to be a unitary node, Spot 1 does not show high fluorescence in the middle slice like the other ROIs mentioned by the reviewer. In addition, the Airyscan imaging system also reassigns pixels and sums the signal from the 32-channel detector to enhance the resolution. Even if not resolved, two closely adjacent nodes can distort the shape of the fluorescent spot, providing grounds for rejection as explained for Spot 1 above.

The node is a structure with a diameter of less than 100 nm. The observed spots are the convolution of the geometry of fluorescent proteins and the point spread function of the microscope, followed by deconvolution. The major factor that determines the size of a spot is the intensity of the fluorescence. The exclusion based on the size of the spot is almost equivalent to the exclusion by the fluorescence intensity, which is proportional to the number of the tagged molecules.

The reviewer is correct about the mechanism of image formation and the influence of intensity on spot size. These facts do not impact our analysis.

7. Unitarity of the 'unitary nodes'The authors accepted that, by FPALM, they had detected clusters smaller (or less number of localized emissions) than the 'unitary nodes' but excluded them from their analysis. They rationalize the omission of these smaller clusters by arguing (1) they are distributed across the entire surface and (2) some of them are not on the flat surface. However, these arguments are weak. As for (1), it is true that there are small clusters across the entire cell surface (Laplante 2016 Figure 1A, an enhanced image on page 4 of problems2.pdf). However, they are much smaller/dimmer than those the reviewer previously spotted in Figure S2E, and the similar ones in Figure 1A found in the equatorial zone with roughly about a half the density of those marked as the 'unitary nodes' (indicated by arrows on page 4 of problems2.pdf). Much smaller clusters everywhere are not a good reason to exclude the slightly smaller clusters (with about a half the number of localized emissions of the 'unitary node') found next to the 'unitary node' in the equatorial cell cortex. As for (2), the dimmer clusters spotted are within 500 nm from those scored as nodes containing one or two 'unitary nodes'. Is the curvature of the cell surface so steep?

The reviewer asks about the PNAS paper by Laplante et al. (2016), not the paper that is under review. We counted 35% more nodes than Laplante (2016). We included in the revision the reviewer’s point that Laplante 2016 may have missed some nodes. One reason is that the image processing of the raw FPALM data excludes localizations outside an optical section of about ~400 nm, so localizations are lost progressively at the lateral edges of face views as the surface curves away from the top of the cell where the microscope was focused.

The authors' strategy to support the unitarity of the 'unitary nodes' is to distinguish the slightly dim spots from the 'unitary nodes' and combine them with the much dimmer cytoplasmic specks. However, as detailed below, their arguments are not sufficiently strong and appear subjective.

We disagree with the observation of the reviewer. In the previous response to review we showed that the average fluorescence of a dim speck was ~170 arbitrary units, which is just 7% of the average intensity of unitary nodes (~ 2300). We included all the spots in our analysis which satisfied our conditions for unitary nodes.

Please note that a newly cited, recent paper, Bellingham-Johnstun et al. (2021), whose corresponding author is the same person as the first author of Laplante et al. (2016), carefully and reasonably avoided the term 'unitary node'.

Bellingham-Johnstun et al. (2021) did not study the scaling of node numbers so they did not need to mention the term unitary node.